# 🐤 PAC Bench: Do Foundation Models Understand Prerequisites for Executing Manipulation Policies?

**Atharva Gundawar**,* **Som Sagar**,* **Ransalu Senanayake**
School of Computing and Augmented Intelligence
Arizona State University, USA
{agundawa, ssagar6, ransalu}@asu.edu

## Abstract

Vision-Language Models (VLMs) are increasingly pivotal for generalist robot manipulation, enabling tasks such as physical reasoning, policy generation, and failure detection. However, their proficiency in these high-level applications often assumes a deep understanding of low-level physical prerequisites, a capability that is largely unverified. To perform actions reliably, robots must comprehend intrinsic object properties (e.g., material, weight), action affordances (e.g., graspable, stackable), and physical constraints (e.g., stability, reachability, or an object's state like being closed). Despite their ubiquitous use in manipulation, we argue that off-the-shelf VLMs may lack this granular, physically-grounded understanding, as these specific prerequisites are often overlooked during training. Addressing this critical gap, we introduce **PAC Bench**, a comprehensive benchmark designed to systematically evaluate VLMs on their understanding of these core **P**roperties, **A**ffordances, and **C**onstraints (PAC) from a task executability perspective. PAC Bench features a diverse dataset with more than 30,000 annotations, comprising 673 real-world images (115 object classes, 15 property types, 1–3 affordances defined per object class), 100 real-world humanoid view scenarios, and 120 unique simulated constraint scenarios across four tasks. Our evaluations reveal significant gaps in the ability of VLMs to grasp fundamental physical concepts, underscoring their current limitations for reliable robot manipulation and pointing to key areas that require targeted research. PAC Bench also serves as a standardized benchmark for rigorously evaluating the physical reasoning capabilities of VLMs guiding the development of more robust and physically grounded models for robot manipulation. Hugging Face : https://huggingface.co/datasets/lens-lab/pacbench.

## 1 Introduction

The quest for generalist robots capable of intelligently and safely interacting with the complexities of the physical world represents a grand challenge in artificial intelligence. Recent breakthroughs in Large Language Models (LLMs) and Vision-Language Models (VLMs) have catalyzed remarkable progress, particularly enabling the development of versatile Vision-Language-Action (VLA) models [1, 2, 3]. These systems leverage the powerful representational capabilities of pre-trained models to interpret multimodal sensory input, generate language-grounded plans, and execute a diverse range of manipulation tasks, showcasing impressive generalization. However, their impressive capabilities often mask a critical, yet largely unverified, assumption: that the underlying foundation models possess a sufficiently deep and physically grounded understanding of the fundamental prerequisites for safe, effective, and truly generalizable manipulation.

---

*Equal Contribution

39th Conference on Neural Information Processing Systems (NeurIPS 2025) Track on Datasets and Benchmarks.

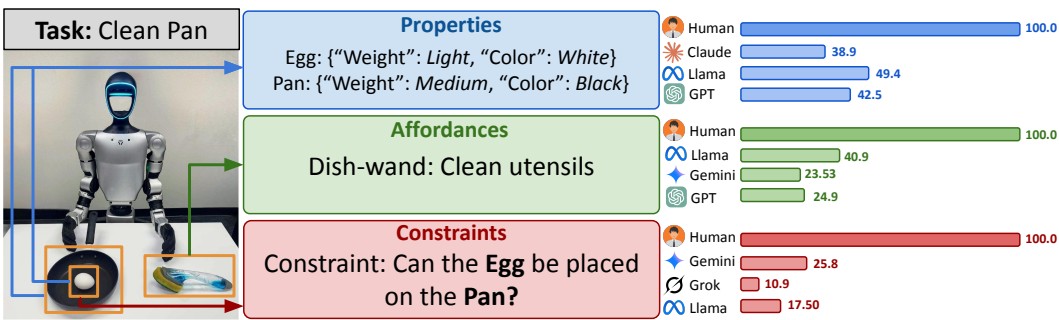

Figure 1: Evaluating foundation models' understanding of Properties, Affordances, and Constraints (PAC) for robotic manipulation. (Left) PAC Bench uses scenarios requiring nuanced physical understanding. (Right) We present example performance of leading VLMs (e.g., GPT-4o, Llama, Claude, Deepseek) on tasks related to Properties (blue), Affordances (green), and Constraints (red), indicating varied strengths and weaknesses across these fundamental reasoning skills.

This assumption demands rigorous scrutiny. Foundation models, despite their exposure to vast quantities of text and video, often lack explicit grounding in the fine-grained physical interplay of objects, actions, and their environmental context knowledge that is intuitive to humans and essential for robust robotic interaction. Consequently, high performance on standard vision-language benchmarks (e.g., VQA [4]) does not reliably translate to the nuanced physical reasoning required to anticipate action outcomes or adapt to novel physical scenarios. Before a robot can confidently execute any manipulation, it must implicitly or explicitly reason about the world: assessing intrinsic object **P**roperties (e.g., Is it heavy? Is it fragile?), discerning valid action **A**ffordances (e.g., Can this be stacked?), and recognizing critical physical **C**onstraints (e.g., Is the target reachable without collision?). Relying on superficial correlations learned from web-scale data, without a robust grasp of these Properties, Affordances, and Constraints (PAC), can lead to unpredictable failures, unsafe operations, and a fundamental brittleness that severely limits their deployment in safety-critical or economically vital open-world applications. As these powerful models are increasingly positioned at the core of autonomous systems, rectifying these gaps in physical understanding is not merely an academic pursuit but a prerequisite for trustworthy and scalable robotic intelligence.

Despite the critical importance of this granular physical understanding, existing benchmarks predominantly focus on end-to-end task performance [5], broad physical knowledge question-answering [6, 7], or other aspects of model behavior like trustworthiness [8] or safety from a policy perspective [9]. A targeted evaluation framework to specifically dissect and measure foundation models' comprehension of the core *prerequisites* for manipulation has been notably absent. This absence hinders targeted improvements, as developers lack precise diagnostics to identify *why* end-to-end policies fail or *which specific aspects* of physical reasoning are underdeveloped in their foundation models.

To bridge this crucial diagnostic gap, we introduce **PAC Bench** (Figure 1): the first benchmark meticulously engineered to evaluate foundation models' understanding of **Properties**, **Affordances**, and **Constraints** essential for robotic manipulation. PAC Bench moves beyond holistic task success by decomposing physical reasoning into these three core, queryable components. Through a diverse suite of targeted evaluations across both simulated and real-world scenarios, our benchmark enables researchers to pinpoint specific deficiencies in models' internal representations of the physical world. We envision PAC Bench not just as an evaluation tool, but as a catalyst for a new wave of research into building more robustly and verifiably grounded foundation models. This detailed diagnostic capability is vital for accelerating the development of VLA systems that can reason causally about their actions, adapt to unforeseen circumstances, and ultimately operate with greater safety and efficacy, advancing the frontier of general-purpose robotics. Our primary contributions are as follows.

1. A benchmark featuring over 30,000 annotations of real scenarios targeting the essential Properties, Affordances, and Constraints for robotic manipulation.
2. A comprehensive suite of tasks and metrics for fine-grained assessment of VLM physical understanding across the three PAC dimensions.
3. Extensive empirical results highlighting current VLM capabilities and critical limitations in PAC reasoning, offering a clear path for advancing physically grounded AI.

Table 1: Comparison of benchmarks evaluating physical properties (P), affordances (A), constraints (C), or related concepts. Manip: Manipulation focus. Sim/Real/Human: Data sources. Parentheses indicate *implicit or partial* coverage of that concept rather than explicit, task-level evaluation. (†) PhysBench includes limited physical dynamics implying some constraint understanding but lacks explicit executability evaluation. (‡) UniAff focuses narrowly on tool-use and 3D motion constraints, covering a subset of affordances but not general manipulation PAC evaluation.

| Benchmark | Concepts Evaluated | | | Focus | Data Source | | | Access | Size | |
| | P | A | C | Manip | Sim | Real | Human | Open Data | Size (GB) | (# Points) |
| --- | --- | --- | --- | --- | --- | --- | --- | --- | --- | --- |
| **PAC Bench (Ours)** | ✓ | ✓ | ✓ | ✓ | ✓ | ✓ | ✓ | ✓ | ∼10 | 30,529 images-text |
| PhysBench [7] | ✓ | (✓) | (✓)† | ✗ | ✓ | ✓ | ✓ | ✓ | ∼10 | 10,002 video-image-text |
| ActAffordance [14] | ✗ | ✓ | ✗ | ✓ | ✗ | ✓ | ✓ | ✓ | 25–40 | 278,000 images |
| EQA-phys [15] | ✗ | ✗ | ✓ | ✓ | ✓ | ✓ | ✗ | ✓ | <1 | 1,300 Q&A |
| Physion [16] | ✗ | ✗ | ✓ | ✗ | ✓ | ✗ | ✗ | ✓ | ∼5 | 1,200 examples |
| ManipVQA [17] | ✓ | ✓ | ✗ | ✓ | ✗ | ✓ | ✓ | ✓ | ∼20 | 84,000 examples |
| UniAff [18] | (✓) | ✓ | (✓)‡ | ✓ | ✓ | ✓ | ✓ | ✓ | 3–5 | 1,500 objects |
| NrVLM [19] | ✗ | ✓ | ✗ | ✓ | ✓ | ✗ | ✗ | ✓ | 5–10 | 4,500 episodes |
| PHYBench [6] | ✓ | ✗ | ✓ | ✗ | ✓ | ✗ | ✗ | ✓ | <1 | ∼500 problems |

## 2 Related Work

The rapid evolution of LLMs and VLMs has spurred a critical need for comprehensive evaluation methodologies. General frameworks like HELM [10] and its visual counterpart VHELM [11] provide holistic assessments across a wide array of tasks and capabilities. Complementing these, numerous benchmarks target specific facets of foundation models, such as trustworthiness with DecodingTrust [8], safety through regulatory lenses with Air-Bench [9], domain-specific reliability in medicine with CARES [12], and agentic capabilities in scientific discovery with MLAgentBench [13]. Public leaderboards further track ongoing performance on various safety and ethical dimensions[2]. While these efforts are crucial for understanding the broader landscape of model behavior, they do not typically delve into the nuanced, granular physical common sense specifically required as prerequisites for robust robotic manipulation.

Closer to the domain of robotics and physical interaction, several benchmarks have begun to probe foundation models' understanding of the physical world. Some focus on general physics knowledge or predictive capabilities. For instance, PHYBench [6] primarily uses text-based scenarios to assess LLMs on formal physics problems, while Physion [16] evaluates visual physical prediction, implicitly testing understanding of object properties and physical constraints governing dynamics. PhysBench [7] offers a broader multimodal evaluation of VLMs, covering aspects like explicit object properties, object relationships, scene understanding, and rudimentary physical dynamics, thus touching upon elements of properties, affordances (via relationships), and constraints (via dynamics).

Other research lines target more specific components of physical understanding relevant to manipulation. For affordances, ManipVQA [17] injects affordance knowledge into VLMs alongside property understanding, ActAffordance [14] focuses on learning bimanual affordances from human videos, and NrVLM [19] develops benchmarks for affordance-guided manipulation based on fine-grained language instructions. UniAff [18] proposes a unified representation for affordances, especially for tool use, and importantly, also incorporates the reasoning of 3D motion constraints and object properties within its framework. For constraints, EQA-phys [15] specifically targets VLM understanding of robotic physical reachability. Distinct from these, benchmarks like The Colosseum [5] are vital for assessing the generalization of end-to-end robotic manipulation policies to various environmental perturbations, rather than the underlying conceptual understanding of physical prerequisites.

Despite this valuable landscape (summarized and compared in Table 1), a critical gap remains: a dedicated, fine-grained benchmark that systematically evaluates whether foundation models comprehend the fundamental and *interconnected prerequisites* for executing manipulation actions, specifically framed through object properties, action affordances, and physical constraints. While works like PhysBench [7] and UniAff [18] evaluate aspects across P, A, and C (as indicated in Table 1) and more recent efforts such as ManipBench [20] explore complementary *low-level* visuomotor reasoning for robotic manipulation through key-point and trajectory prediction, PAC Bench distinguishes

---

[2]https://huggingface.co/spaces/AI-Secure/llm-trustworthy-leaderboard

itself through several key dimensions. First, it focuses on the explicit, understanding of these three components as *preconditions* for action, rather than evaluating them solely through downstream task performance. Second, PAC Bench is designed to assess these PAC dimensions with a granularity specifically tailored to common manipulation scenarios, supported by a dataset that combines diverse real-world images (for properties and affordances) with both simulated and novel real-world humanoid-view scenarios (for constraints). While existing benchmarks may test general physics knowledge, dynamic prediction, or policy generalization, PAC Bench fundamentally probes whether VLMs can reason about the specific P, A, and C conditions that make a manipulation task executable in the first place, a crucial step towards building more robust, and safe VLA systems.

## 3 The PAC Bench Dataset

PAC Bench evaluates a VLM's understanding of three fundamental, interdependent components crucial for determining the executability of robotic manipulation actions:

1. **Properties:** These are the inherent physical or material characteristics of objects, as well as their states, that dictate how they behave and can be interacted with. In PAC Bench, we focus on a comprehensive suite of 12 distinct physical and material attributes, including, for instance, an object's inferred *Weight* (e.g., light, medium, heavy), its *Material* (e.g., wood, metal, plastic), its *Containment State* (e.g., lidded, open, sealed). Accurately perceiving these properties is the first step towards effective physical reasoning.
2. **Affordances:** Affordances [21, 22] describe the potential for action that an object offers to an agent, or that an agent can enact upon an object [23, 24], given its properties and the broader environmental context. These are specifically tailored to manipulation, covering common interactions such as *is-graspable* (by a standard gripper), *is-containable-in* (for placing objects), and *is-stackable-on* (another object). Understanding affordances bridges the gap between object perception and actionable knowledge.
3. **Constraints:** These are the physical, geometric, or environmental limitations and conditions that govern whether an intended action can be successfully executed given a task. Failure to recognize constraints often leads to task failure or unsafe robot behavior. PAC Bench evaluates understanding of constraints such as *stability limits* (e.g., predicting if stacking a specific object will cause a topple), *containment failure* (e.g., contents spilling if an open container is moved inappropriately), and *reachability issues for a robotic arm*.

A grounded understanding of these three pillars – Properties, Affordances, and Constraints – is paramount for any robotic system intended to operate robustly in the complexities of the real world. Without it, even sophisticated policies are prone to errors stemming from a superficial interpretation of the scene. For instance, attempting to lift an object perceived as light (misjudged Property) might fail if it is actually heavy. Similarly, trying to stack an object that appears stackable (misjudged Affordance) might lead to collapse if its instability (unrecognized Constraint) is not considered. PAC Bench is therefore designed to specifically probe these interconnected concepts, offering a more targeted benchmarks focusing on broader physics knowledge. By focusing on PAC, we aim to evaluate the foundational understanding that enables models to predict action feasibility *before* execution, a critical component for building more reliable and adaptable robotic agents. Note that *we only evaluate pre-trained VLMs' capabilities without having access to a specific robot or the environment it operates in* as these generic pre-trained VLMs serve as the backbone for the development of VLAs [1], failure detection models [25], etc.

### 3.1 Data Acquisition and Curation

The PAC Bench dataset is constructed through a multi-faceted approach, aggregating data from diverse real-world image sources and meticulously designed scenarios from both simulated and real-world robot interactions (Fig 2). This hybrid strategy ensures broad visual diversity for property and affordance, complemented by targeted and varied constraint evaluations from multiple views.

**Data for Properties and Affordances: Diverse Real-World and Simulated Imagery.** To ground our property and affordance assessments in varied visual data, PAC Bench aggregates images from four key sources (2 real, 2 simulation): the extensive *OpenImages Dataset V7 and Extensions* [26], novel real-world captures from multiple perspectives (agent and side views) of a *Unitree G1 humanoid robot*, multi-angle(24) capture of 45 unique objects from the *RoboCasa framework* [27], which

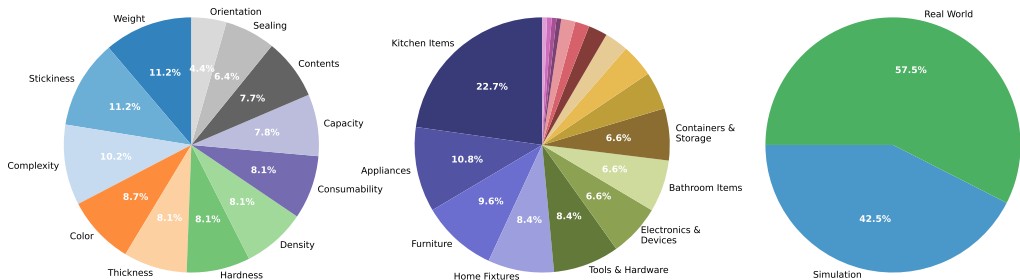

Figure 2: Distribution of annotations in PAC Bench across three dimensions: (Left) physical properties annotated in the dataset, showing the relative frequency of each property; (Center) affordance categories, with slices below 5% omitted for clarity; (Right) constraint domains, contrasting simulation (blue shades) and real-world (green shades) scenarios.

leverages the MuJoCo physics engine for structured household environments. Across these sources, we target **115 unique object classes** (e.g., *Container, Towel, Chair, Apple, Knife*), selected for their prevalence and relevance to household manipulation. These are organized into **18 primary categories** (e.g., Appliances, Furniture, Kitchen Items; see Appendix B.2 for full taxonomy). We utilized the provided human-annotated bounding boxes for annotations. This ensures precise localization for our subsequent PAC annotations. For the VLM evaluations detailed in this paper, we curated **977** images from OpenImages and our Unitree G1 captures. The RoboCasa image data (1080 unique images), while part of the full PAC Bench dataset release to support broader research, is not included in the current VLM evaluation set due to computational costs.

*Property Annotation:* For each of the 977 curated images, we annotated a comprehensive suite of intrinsic and extrinsic physical properties. We defined a set of **12 distinct property types** relevant to manipulation, including: *Stickiness, Thickness, Density, Sealing, Contents, Capacity, Complexity of Parts, Consumability, Orientation, Hardness, Color*, and *Weight*. This resulted in a total of **27,674 property annotations** across the dataset. (Detailed definitions are shown in Appendix B.1.) The property annotation process was designed for high quality. Each image instance, along with a specific property query (e.g., "What is the material of the object in the bounding box?"), was presented to annotators with a set of predefined, mutually exclusive answer choices. To ensure reliability, every image instance was independently annotated for each property by **two human annotators**. The final ground-truth label for a given property was determined by consensus, requiring agreement between both annotators. Disagreements were resolved by a senior annotator or discarded if no consensus could be reached. This rigorous process yielded a high-quality set of property labels. We utilized *LabelBox* as our annotation platform, with a team of over **10 annotators** contributing to this effort (Appendix E.4).

*Affordance Annotation:* For each of the 115 selected object classes, we also collected affordance labels. The process involved manually identifying and listing the **top three most common action affordances** associated with each object class, ranked in order of typicality or importance. For example, for the object class *Chair*, the annotated affordances include (1) *is-sittable*, (2) *is-climbable*, and (3) *can-place-objects-on*. This initial phase of affordance annotation was conducted by assigning each object class to a primary annotator. This initial phase of affordance annotation involved a primary annotator per object class. While this provides a foundational set of common affordances, we acknowledge that future work will involve expanding this with multiple annotators to establish inter-annotator agreement and a consensus-based label set.

**Data for Constraints: Simulated and Real-World Humanoid Scenarios.** To evaluate the understanding of physical constraints often involving complex or dynamic interactions PAC Bench incorporates data from both simulated environments and the real-world humanoid robot perspectives. This hybrid strategy allows for scalable, controlled generation of diverse constraint types in simulation, ideal for iterative VLM testing and aligning with common policy training paradigms. These are complemented by authentic real-world humanoid scenarios that ground evaluations in genuine physical complexities and robot-centric perspectives, offering a crucial testbed for sim-to-real transfer of constraint comprehension. (Detailed specifications for all constraint domains are in Appendix B.3).

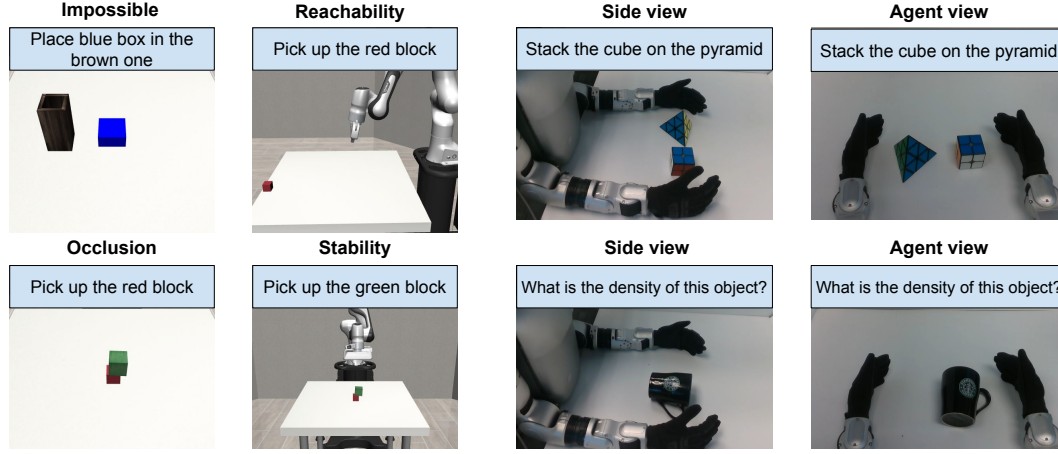

Figure 3: Examples from PAC Bench. (Left 4 Images) Scenarios designed to evaluate understanding of various physical **C**onstraints: Impossible Placement, Occlusion, Reachabilityand Stability. (Right 4 images) Example of a **P**roperty query presented with a real-world robot view from PAC Bench.

*Simulated Constraint Scenarios:* We leveraged the MuJoCo physics engine [28] to generate synthetic scenarios depicting various constraints (Fig 3 (left)) relevant to robotic manipulation. We designed four primary constraint domains:

1. **Impossible Placement:** Scenarios where an object cannot be stably placed on another due to factors like shape, size mismatch, or unstable support.
2. **Occlusion/Support Issues:** Challenges related to accessing an object, such as attempting to pick up a target block that is currently supporting another block.
3. **Stability Constraints:** Situations involving picking up an object that is itself part of an unstable assembly.
4. **Reachability and Access Constraints:** Scenarios where an object is present but difficult or impossible to reach due to its position or surrounding obstacles.

For each simulated constraint domain, we procedurally generated **10 distinct environment instantiations** by introducing randomization in object positions, orientations, and/or distractor elements. Each instantiation was rendered from **three different camera viewpoints** (front, agent, and side view) to provide visual diversity and assess view-invariance. This resulted in a total of **120 unique simulated constraint scenarios**.

*Real-World Humanoid Constraint Scenarios:* These scenarios involve a dual-arm Unitree G1 humanoid robot attempting simple manipulation tasks in tabletop environments with everyday objects. For each scenario, we captured synchronized images from two camera views. A question was then formulated about a potential action and the physical constraints that might prevent its successful execution (see Appendix D.1 for an example prompts). The ground-truth answer provides an explanation of the relevant constraint(s). This real-world component currently comprises **2727** unique question-answer scenarios, focusing on constraints such as (Question: Can you keep the food on the plate? Expected Answer: No the plate is inverted.). (Appendix B.3 provides further examples.)

## 4 Experimental Results and Analysis

In this section, we present the empirical evaluation of several state-of-the-art foundation models [29, 30, 31, 32, 33, 34, 35, 36, 37] on PAC Bench. We detail our experimental setup, followed by an analysis of model performance on understanding object properties, action affordances, and physical constraints.

### 4.1 Experimental Setup

**Models Evaluated:** We evaluated a diverse suite of publicly available and proprietary VLMs to assess their PAC understanding capabilities. For some models, we also explored different prompting strategies (e.g., direct querying vs. chain-of-thought prompting in Appendix D).

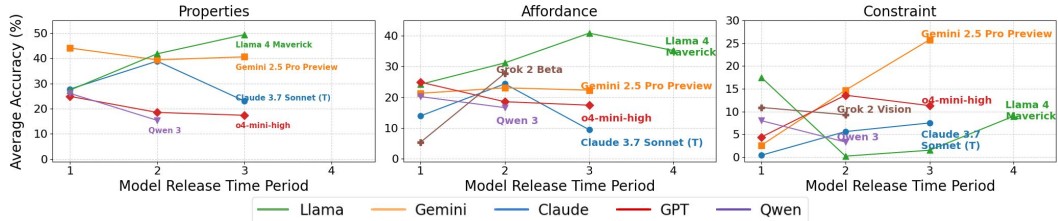

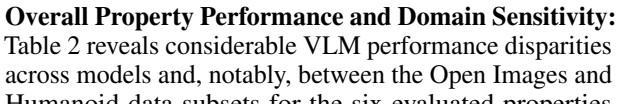

Figure 4: Comparative PAC understanding profiles of selected VLMs. The x-axis indicates nominal model release time periods (1=earlier to 4=most recent among those shown). The diverse performance signatures suggest varied developmental trajectories in acquiring physical common sense.

**Evaluation Protocol:** For each task in PAC Bench, VLMs were provided with images from a scenario and a textual prompt (Appendix D.1) that queries a specific property, affordability, or constraint. Model responses were evaluated against ground-truth annotations derived from our dataset.

1. *Property* questions were multiple-choice (typically [Number, e.g., 3-5] options) targeting one of 12 predefined attributes for a specified object (e.g., "What is the density of the object in the box? A) High, B) Low...").
2. *Affordance* questions required models to provide all applicable affordances for a given object class (e.g., "What are the affordances of [object]? A) Can carry items, B) is-stackable...").
3. *Constraint* questions asked models to determine the feasibility of an action or identify the most constraining pre-condition to successfully complete a task. (e.g., "Can the robot stack X on Y? If no, why?").

Table 2: Property accuracies (%) for Open Images (PAC Bench) vs. Humanoid benchmarks. Properties P1–P6 are: Color, Contents, Weight, Density, Sealing, Hardness.

| | **Open Images** | | | | | | **Humanoid** | | | | | | **Avg** |
|---|---|---|---|---|---|---|---|---|---|---|---|---|---|
| **Model** | **P1** | **P2** | **P3** | **P4** | **P5** | **P6** | **P1** | **P2** | **P3** | **P4** | **P5** | **P6** | |
| Claude 3.5 Sonnet | 0.0 | 31.9 | 0.0 | 0.0 | 2.7 | 42.3 | 50.2 | 28.9 | **50.7** | 52.7 | 19.4 | 55.2 | **27.8** |
| Claude 3.7 Sonnet | 20.2 | 23.5 | 32.6 | 36.7 | 66.4 | 37.0 | 47.8 | 30.3 | 48.3 | 55.7 | 13.2 | 55.7 | **38.9** |
| Claude 3.7 Sonnet (T) | 6.7 | 22.3 | 15.0 | 9.0 | 50.9 | 23.4 | 24.9 | 11.9 | 28.5 | 36.3 | 8.3 | 39.8 | **23.1** |
| Gemini 2.0 Flash 001 | 19.7 | **35.3** | **40.8** | **58.0** | 56.1 | **43.9** | **55.2** | 39.8 | 40.3 | 46.8 | 38.2 | 54.7 | **44.1** |
| Gemini 2.5 Flash P | 26.9 | 28.8 | 27.1 | 40.1 | 58.9 | 31.1 | 53.2 | 27.9 | 33.8 | 40.3 | 41.8 | 63.2 | **39.4** |
| Gemini 2.5 Pro Pre (T) | 27.0 | 34.1 | 31.2 | 43.2 | 57.2 | 16.7 | 13.0 | 42.7 | 49.5 | 55.7 | 53.5 | 64.0 | **40.6** |
| GPT-4.1 Mini | 26.6 | 28.4 | 24.1 | 43.2 | 64.0 | 18.1 | 36.3 | 36.3 | 26.9 | 40.3 | 15.3 | 60.2 | **35.0** |
| GPT-4.1 | 13.8 | 29.0 | 4.4 | 25.9 | **91.0** | 27.8 | 51.2 | 55.7 | 43.3 | **58.2** | **43.8** | **64.2** | **42.4** |
| o4-mini-high (T) | 17.1 | 0.2 | 4.7 | 26.4 | 72.7 | 26.2 | 20.4 | 36.6 | 31.5 | 52.7 | 43.1 | 63.8 | **33.0** |
| Llama 3.2 90B Vision I | 13.1 | 14.8 | 4.2 | 25.0 | 30.2 | 12.8 | 37.3 | 51.2 | 31.3 | 44.8 | 27.1 | 37.3 | **27.4** |
| Llama 4 Scout | 30.4 | 0.6 | 36.4 | 51.1 | 84.9 | 18.6 | 51.2 | 60.2 | 37.8 | 43.3 | 36.1 | 51.2 | **41.8** |
| Llama 4 Maverick | **36.2** | 34.9 | 37.6 | 47.0 | 90.0 | 14.6 | 43.8 | **77.1** | 59.2 | 57.7 | 40.3 | 54.2 | **49.4** |
| Qwen 3 | 18.7 | 22.7 | 9.9 | 20.1 | 85.2 | 28.6 | 0.0 | 0.0 | 0.0 | 0.0 | 0.0 | 0.0 | **15.4** |
| Qwen 2.5 VL | 21.9 | 20.7 | 18.7 | 9.6 | 61.8 | 42.3 | 47.8 | 22.9 | 24.9 | 4.5 | 2.8 | 35.8 | **26.1** |

### 4.2 Analyzing Property Awareness: Do VLMs Discern Fundamental Object Features?

This subsection presents a detailed evaluation of how well contemporary VLMs are grounded in these essential attributes. We assessed model performance across twelve distinct property categories critical for robotic manipulation: P1 (Capacity), P2 (Color), P3 (Complexity), P4 (Consumability), P5 (Contents), P6 (Density), P7 (Hardness), P8 (Orientation), P9 (Sealing), P10 (Stickiness), P11 (Thickness), and P12 (Weight). The comprehensive results are detailed in Table 7.

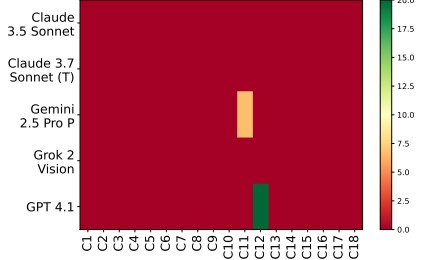

Figure 5: All affordance subset heatmap. Full heatmap in Appendix D.3

**Overall Property Performance and Domain Sensitivity:** Table 2 reveals considerable VLM performance disparities across models and, notably, between the Open Images and Humanoid data subsets for the six evaluated properties. No single model masters all properties across both domains, highlighting varied strengths and significant domain sensitivity. Many models,

such as 'Claude 3.5 Sonnet' and 'GPT-4.1', demonstrate decent accuracy on properties like 'Color (P1)', 'Weight (P3)' when evaluated on Humanoid views compared to the more varied Open Images data. Conversely, properties like 'Sealing (P5)' frequently see higher scores on Open Images (e.g., 'Llama 4 Maverick': 90.0% vs. 40.3%). Some models show extreme domain dependence; for instance, 'Qwen 3' performs reasonably on Open Images but scores 0.0% across all six properties on the Humanoid dataset. These findings underscore that VLM property understanding is not yet consistently robust across different visual contexts, even for fundamental attributes, pointing to challenges in generalization. For detailed results across all 12 evaluated properties from our primary dataset, see Appendix D.2.

### 4.3 Evaluating Affordance Understanding: Can VLMs Discern Possible Interactions?

Recognizing potential actions, or affordances, that an object offers is fundamental for goal-oriented manipulation. In this subsection, we assess VLM performance on identifying common affordances for 115 object classes, primarily grouped into 14 primary categories derived from web-scale images (A1-A14). Table 3 presents results for the metric of identifying *at least one* correct affordance, and importantly, also includes overall accuracies from our distinct Humanoid dataset evaluations and an aggregated average. The stricter metric, requiring identification of *all* ground-truth affordances, is detailed in Table 10 (further visualized in Figure 5). Additional results for multi-category and per-object evaluations are in Appendix D.3.

**Partial Affordance Recognition and Humanoid Insights:** As shown in Table 3, VLMs exhibit highly varied success in recognizing at least one correct affordance. Based on the overall average scores (Avg), which combine web-image category and humanoid task performance, models like 'Llama 4 Scout' (40.9%) and 'Llama 4 Maverick' (35.2%) demonstrate broader, albeit still moderate, capabilities. Performance peaks on specific web-image categories are notable, for instance, 'GPT 4.1 Mini' and 'Qwen 2.5 VL' achieve 100% for A10, and several Llama models along with 'Claude 3.7 Sonnet' show perfect scores for A1. However, many categories, such as A2.The Humanoid dataset scores (H1-H3) reveal further nuances; for example, 'Gemini 2.0 Flash 001' performs decent across H1-H3 (avg. 60%), while 'Gemini 2.5 Pro P' scores 0% on all Humanoid tasks despite reasonable performance on A1-A14. This suggests that affordance understanding from diverse web images may not readily transfer to specific robot-centric views or tasks without further adaptation, with models like 'Qwen VP' (0.0% Avg) struggling broadly.

**Comprehensive Affordance Recognition Remains Elusive:** The capacity of VLMs to identify the *full set* of an object's affordances is far more limited. As starkly illustrated in Table 10 (and the heatmap in Figure 5), when requiring models to recognize all ground-truth affordances, performance plummets to near-zero across almost all models and categories. The rare non-zero scores (e.g., 'GPT 4.1' at 20.0% for Home Fixtures. This significant drop from the "at least one" metric highlights that while VLMs might identify a primary or common affordance, they generally lack the comprehensive functional understanding critical for versatile and truly intelligent robotic interaction.

### 4.4 Assessing Constraint Comprehension: Can VLMs Understand Physical Limits?

PAC Bench evaluates constraints by presenting VLMs with scenarios where proposed actions might be infeasible due to underlying physical limitations. Our evaluation spans four distinct constraint domains. Furthermore, we introduce a novel set of real-world constraint scenarios captured from a humanoid robot's perspective, which will be analyzed subsequently. The performance of VLMs on the simulated constraint tasks is detailed in Table 4 (More in Appendix D.4).

**Constraint Understanding: A Profound Challenge Across Simulated and Real-World Scenarios.** The results presented in Table 4 underscore that reasoning about physical constraints remains a profound challenge for current VLMs, with overall average (Avg) accuracies being exceptionally low for most models. Many prominent VLMs, including Claude 3.5 Sonnet (0.4% Avg), Llama 3.2 90B Vision I (0.2% Avg), and Llama 4 Scout (1.5% Avg), register near-zero performance across the majority of both simulated and real-world tasks. This pervasive failure highlights a fundamental difficulty in inferring basic stability, support, occlusion, and reachability limits from visual input.

In the **Simulated Domains**, "Impossible Placement" scenarios almost universally failed. The "Occlusion" domain saw slightly more success, particularly from Gemini 2.5 Pro Preview (up to 90.0%) and GPT-4.1 (up to 70.0%). "Stability" and "Reachability" tasks in simulation also proved

Table 3: Affordance Accuracy (%) of VLMs on recognizing at least one correct affordance for objects grouped by primary categories (Single-Category Mapping) in PAC Bench, plus overall accuracy in the humanoid dataset scores H1–H3. Categories A1–A18 are: A1 (Adhesives), A2 (Appliances), A3 (Luggage), A4 (Bathroom Items), A5 (Cleaning), A6 (Clothing), A7 (Storage), A8 (Decor), A9 (Electronics), A10 (Food & Beverage), A11 (Furniture), A12 (Home Fixtures), A13 (Kitchen Items), A14 (Instruments), H1 (Humanoid Front View), H2 (Side View), H3 (Both Views)

| Model | A1 | A2 | A3 | A4 | A5 | A6 | A7 | A8 | A9 | A10 | A11 | A12 | A13 | A14 | H1 | H2 | H3 | Avg |
|---|---|---|---|---|---|---|---|---|---|---|---|---|---|---|---|---|---|---|
| Claude 3.5 Sonnet | 0.0 | 0.0 | 0.0 | 0.0 | 0.0 | 0.0 | 16.7 | **25.0** | 0.0 | 66.7 | 13.3 | 40.0 | 9.1 | 0.0 | 2.9 | 47.1 | 14.7 | **13.9** |
| Claude 3.7 Sonnet | **100.0** | 0.0 | 0.0 | 20.0 | 0.0 | 0.0 | 0.0 | 0.0 | 11.1 | 66.7 | 0.0 | 20.0 | 22.7 | **100.0** | 2.9 | 58.8 | 11.8 | **24.4** |
| Claude 3.7 Sonnet (T) | 0.0 | 5.6 | 0.0 | 30.0 | 0.0 | 0.0 | 0.0 | 0.0 | 11.1 | 0.0 | 6.7 | 20.0 | 18.2 | 0.0 | 2.9 | 54.4 | 10.3 | **9.4** |
| Gemini 2.0 Flash 001 | 0.0 | 0.0 | 0.0 | 40.0 | 0.0 | 0.0 | 16.7 | 0.0 | 0.0 | 66.7 | 0.0 | 40.0 | 13.6 | 0.0 | **54.4** | 66.2 | 64.7 | **21.3** |
| Gemini 2.5 Flash P | 0.0 | 5.6 | 0.0 | 20.0 | 0.0 | 50.0 | 0.0 | 0.0 | 11.1 | 66.7 | 13.3 | 40.0 | 18.2 | 0.0 | 52.9 | 55.9 | 57.4 | **23.0** |
| Gemini 2.5 Pro P | 0.0 | 16.7 | 66.7 | 30.0 | 0.0 | 0.0 | 33.3 | **25.0** | 22.2 | 66.7 | 26.7 | 60.0 | 31.8 | 0.0 | 0.0 | 0.0 | 0.0 | **22.3** |
| Llama 3.2 11B Vision I | **100.0** | 22.2 | 0.0 | 30.0 | 0.0 | 50.0 | 33.3 | 0.0 | 22.2 | 66.7 | 0.0 | 0.0 | 13.6 | 0.0 | 20.5 | 27.9 | 25.0 | **24.2** |
| Llama 3.2 90B Vision I | **100.0** | 11.1 | 33.3 | 10.0 | 0.0 | 50.0 | **50.0** | **25.0** | 22.2 | 66.7 | 26.7 | 60.0 | 9.1 | 0.0 | 22.1 | 44.1 | 0.0 | **31.2** |
| Llama 4 Scout | 0.0 | 11.1 | 66.7 | 50.0 | 0.0 | 50.0 | **50.0** | **25.0** | 33.3 | 66.7 | 53.3 | 60.0 | **54.6** | **100.0** | 20.6 | 27.9 | 26.5 | **40.9** |
| Llama 4 Maverick | 0.0 | 22.2 | 33.3 | 50.0 | 0.0 | **100.0** | **50.0** | 0.0 | 33.3 | 66.7 | 26.7 | **100.0** | 31.8 | 0.0 | 20.6 | 39.7 | 23.5 | **35.2** |
| GPT 4.1 Mini | 0.0 | 5.6 | 0.0 | 30.0 | 0.0 | 0.0 | **50.0** | **25.0** | 0.0 | **100.0** | 13.3 | 60.0 | 36.4 | 0.0 | 20.6 | 57.4 | 25.0 | **24.9** |
| GPT 4.1 | 0.0 | 5.6 | 0.0 | 20.0 | 0.0 | 0.0 | 16.7 | **25.0** | 0.0 | 0.0 | 6.7 | 60.0 | 18.2 | 0.0 | 48.5 | **67.6** | 45.6 | **18.5** |
| o4-mini-high (T) | 0.0 | 16.7 | 0.0 | 20.0 | 0.0 | 0.0 | 16.7 | **25.0** | 11.1 | 33.3 | 33.3 | 20.0 | 22.7 | 0.0 | 16.2 | 45.6 | 35.3 | **17.4** |
| Qwen 2.5 VL | 0.0 | 0.0 | 0.0 | 30.0 | 0.0 | 0.0 | 33.3 | 0.0 | 0.0 | **100.0** | 6.7 | 80.0 | 9.1 | 0.0 | 14.7 | 48.5 | 20.6 | **20.2** |
| Qwen 3 | 0.0 | 5.5 | 0.0 | 30.0 | 0.0 | 0.0 | 33.3 | **25.0** | 0.0 | **100.0** | 0.0 | 60.0 | 13.6 | 0.0 | 4.4 | 1.4 | 8.8 | **16.6** |
| Grok Vision Beta | 0.0 | 5.6 | 0.0 | 10.0 | 0.0 | 0.0 | 0.0 | 0.0 | 11.1 | 0.0 | 13.3 | 20.0 | 4.6 | 0.0 | 8.8 | 8.8 | 7.4 | **5.3** |
| Grok 2 Vision | 0.0 | 5.6 | 33.3 | 50.0 | 0.0 | 0.0 | 0.0 | 0.0 | 11.1 | **100.0** | 6.7 | 20.0 | 13.6 | **100.0** | 44.1 | 47.1 | 41.2 | **27.8** |

Table 4: Constraint Accuracy (%) of VLMs on understanding physical constraints in PAC Bench across four simulated domains, three views (**F**: front-view, **A**: agent-view, **S**: side-view), and a real-world **Humanoid** split ( **Both=A+S**).

| Model | Simulation | | | | | | | | | | | | Real World | | | Avg |
|---|---|---|---|---|---|---|---|---|---|---|---|---|---|---|---|---|
| | Impossible Place (↑) | | | Occlusion (↑) | | | Stability (↑) | | | Reachability (↑) | | | Humanoid (↑) | | | |
| | F | A | S | F | A | S | F | A | S | F | A | S | A | S | Both | |
| Claude 3.5 Sonnet | 0.0 | 0.0 | 0.0 | 0.0 | 0.0 | 0.0 | 0.0 | 0.0 | 0.0 | 0.0 | 0.0 | 0.0 | 1.8 | 0.0 | 3.7 | **0.4** |
| Claude 3.7 Sonnet | 0.0 | 0.0 | 0.0 | 40.0 | 10.0 | 30.0 | 0.0 | 0.0 | 0.0 | 0.0 | 0.0 | 0.0 | 1.8 | 0.0 | 1.8 | **5.6** |
| Claude 3.7 Sonnet (T) | 0.0 | 0.0 | 0.0 | 20.0 | 20.0 | 30.0 | 0.0 | 0.0 | 10.0 | 10.0 | 0.0 | 0.0 | 0.0 | 0.0 | 3.7 | **7.5** |
| Gemini 2.0 Flash 001 | 0.0 | 0.0 | 0.0 | 10.0 | 10.0 | 0.0 | 0.0 | 0.0 | 0.0 | 0.0 | 0.0 | 0.0 | 5.6 | 3.7 | 9.4 | **2.6** |
| Gemini 2.5 Flash P | 0.0 | 0.0 | 0.0 | 50.0 | 20.0 | 40.0 | 10.0 | **40.0** | 20.0 | 0.0 | 20.0 | 0.0 | 9.4 | 9.4 | 1.8 | **14.7** |
| Gemini 2.5 Pro P | 10.0 | **20.0** | 10.0 | **90.0** | 30.0 | 60.0 | 0.0 | **40.0** | 0.0 | **30.0** | 0.0 | **20.0** | 11.3 | 18.8 | 9.4 | **25.8** |
| Llama 3.2 11B Vision I | **20.0** | 10.0 | 0.0 | 30.0 | 30.0 | 20.0 | 20.0 | 20.0 | 20.0 | 10.0 | **30.0** | 0.0 | 0.0 | 1.8 | 0.0 | **17.5** |
| Llama 3.2 90B Vision I | 0.0 | 0.0 | 0.0 | 0.0 | 0.0 | 0.0 | 0.0 | 0.0 | 0.0 | 0.0 | 0.0 | 0.0 | 3.7 | 0.0 | 0.0 | **0.2** |
| Llama 4 Scout | 0.0 | 0.0 | 0.0 | 0.0 | 0.0 | 20.0 | 0.0 | 0.0 | 0.0 | 0.0 | 0.0 | 0.0 | 0.0 | 0.0 | 1.8 | **1.5** |
| Llama 4 Maverick | 0.0 | 0.0 | 0.0 | 10.0 | 0.0 | 50.0 | 30.0 | 10.0 | 10.0 | 0.0 | 0.0 | 0.0 | 9.4 | 7.5 | 7.5 | **9.0** |
| GPT-4.1 | 0.0 | 0.0 | 0.0 | 50.0 | **70.0** | 50.0 | 0.0 | 0.0 | 0.0 | 0.0 | 0.0 | 0.0 | 11.3 | 13.2 | 9.4 | **13.6** |
| GPT-4.1 Mini | 0.0 | 0.0 | 0.0 | 0.0 | 0.0 | 0.0 | 0.0 | 0.0 | 0.0 | 0.0 | 0.0 | 0.0 | 18.8 | 24.5 | 22.6 | **4.4** |
| o4-mini-high (T) | 0.0 | 0.0 | 0.0 | 60.0 | 40.0 | 50.0 | 0.0 | 0.0 | 20.0 | 0.0 | 0.0 | 0.0 | 11.3 | 13.2 | 11.3 | **11.3** |
| Qwen 2.5 VL | 0.0 | 0.0 | 0.0 | 20.0 | 20.0 | 10.0 | 10.0 | 20.0 | 20.0 | 20.0 | 0.0 | 0.0 | 0.0 | 0.0 | 0.0 | **8.0** |
| Qwen 3 | 10.0 | 0.0 | 0.0 | 60.0 | 20.0 | **70.0** | 30.0 | 80.0 | 80.0 | 10.0 | 10.0 | 0.0 | 3.7 | 0.0 | 0.0 | **3.3** |
| Grok Vision Beta | 0.0 | 0.0 | 0.0 | 33.3 | 50.0 | 0.0 | 25.0 | 0.0 | **22.2** | 11.1 | 0.0 | 11.1 | 0.0 | 0.0 | 0.0 | **10.9** |
| Grok 2 Vision | 0.0 | 0.0 | 0.0 | 20.0 | 50.0 | 40.0 | 10.0 | 0.0 | 10.0 | 10.0 | 0.0 | 0.0 | 0.0 | 0.0 | 0.0 | **9.3** |

very difficult, with only sporadic, low scores from most models, though Gemini 2.5 Pro P and Llama 3.2 11B Vision Instruct showed some capability in specific views for Reachability. Viewpoint (F, A, S) within simulation influenced scores inconsistently (e.g., Gemini 2.5 Pro P on Sim-Occlusion: F:90.0%, A:30.0%, S:60.0%), indicating a lack of robust view-invariance.

In **Real-World Humanoid** scenarios performance is generally low, though some models show interesting divergences. GPT-4.1 Mini, despite near-zero performance in simulation, achieves comparatively better scores on the Humanoid tasks (18.8% Agent, 24.5% Side, 22.6% Both), although its overall average remains low (4.4%). Conversely, Gemini 2.5 Pro, the strongest performer in simulation (25.8% Avg), shows more modest results on the Humanoid tasks (11.3% Agent, 18.8% Side, 9.4% Both). This suggests that performance in simulated constraint scenarios does not directly translate to real-world robot-centric views, pointing to a significant sim-to-real gap in constraint understanding. Reasoning models, as seen with "(T)," provided only marginal and inconsistent benefits in these highly challenging constraint tasks. The overall poor performance across constraint evaluations clearly marks constraint comprehension as a critical area requiring substantial advancement for reliable VLM-driven robotics.

# 5 Key Findings

> **Key Findings from PAC Bench**
>
> - VLMs perform moderately on **object properties** and **basic affordances**.
> - They fail significantly on understanding **complex affordances** and **physical constraints**.
> - Performance varies drastically across **domains** and **viewpoints**.

We expect PAC Bench to help the development of physical AI agents in the following ways:

1. Targeted diagnosis of failure modes:
   Training large VLAs is still a largely heuristic, trial-and-error process that demands substantial computational resources. PAC Bench helps determine whether failures arise from poor physical understanding in the foundation VLM itself (e.g., lack of ability to understand a particular constraint) vs. issues introduced during architectural choices or fine-tuning process.

2. Improved robustness and transferability:
   PAC Bench exposes the sensitivity of VLMs to domain, viewpoint, and other shifts, highlighting the sim-to-real and view-invariance challenges critical to real-world robotics. This enables more systematic adaptation of VLA systems to diverse environments, improving both reliability and generalization.

3. Cognitive modular testing and verification:
   By decomposing the pre-requisites for manipulation into Properties, Affordances, and Constraints, PAC Bench allows each component of a robot's reasoning pipeline to be individually tested and empirically verified. This *cognitive modularity*, inspired by the *core knowledge systems* [38] identified in developmental psychology, stands in contrast to traditional data-flow modularity by enabling the development of interpretable and verifiable manipulation policies in the era of foundation models, thereby helping ensure that learned behaviors align with real-world safety and reliability requirements prior to deployment.

# 6 Limitations and Conclusion

Although PAC Bench offers a significant step forward with its diverse hybrid dataset for evaluating VLM understanding of **P**roperties, **A**ffordances, and **C**onstraints (PAC), we acknowledge current limitations. These include the initial single-annotator pass for affordances, the exclusion of the RoboCasa subset from current VLM evaluations due to cost all of which suggest avenues for future expansion and refinement. Despite these, we introduced **PAC Bench** to address the critical, often unverified, assumption of deep physical grounding in VLMs for robotic manipulation. Our extensive evaluations of state-of-the-art models starkly reveal widespread deficiencies: while partial success is observed in property and basic affordance recognition, VLMs profoundly struggle with comprehensive affordance understanding and nearly all aspects of constraint reasoning in both simulated and real-world tests. These findings underscore that current VLM sophistication does not yet equate to robust physical grounding. PAC Bench thus provides the community with a crucial diagnostic tool and a structured methodology to systematically measure these foundational skills, pinpoint key weaknesses (such as poor constraint generalization or difficulty with compositional affordances), and catalyze the development of more physically intelligent, reliable, and ultimately, safer VLMs for real-world robotic interaction.

# Acknowledgment

We would like to thank the members of the LENS Lab[3] at ASU and Yifan Zhou for continual feedback.

---

[3]`https://ransml.github.io/lens-lab/`

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

# NeurIPS Paper Checklist

The checklist is designed to encourage best practices for responsible machine learning research, addressing issues of reproducibility, transparency, research ethics, and societal impact. Do not remove the checklist: **The papers not including the checklist will be desk rejected.** The checklist should follow the references and follow the (optional) supplemental material. The checklist does NOT count towards the page limit.

Please read the checklist guidelines carefully for information on how to answer these questions. For each question in the checklist:

- You should answer [Yes] , [No] , or [NA] .
- [NA] means either that the question is Not Applicable for that particular paper or the relevant information is Not Available.
- Please provide a short (1–2 sentence) justification right after your answer (even for NA).

**The checklist answers are an integral part of your paper submission.** They are visible to the reviewers, area chairs, senior area chairs, and ethics reviewers. You will be asked to also include it (after eventual revisions) with the final version of your paper, and its final version will be published with the paper.

The reviewers of your paper will be asked to use the checklist as one of the factors in their evaluation. While "[Yes] " is generally preferable to "[No] ", it is perfectly acceptable to answer "[No] " provided a proper justification is given (e.g., "error bars are not reported because it would be too computationally expensive" or "we were unable to find the license for the dataset we used"). In general, answering "[No] " or "[NA] " is not grounds for rejection. While the questions are phrased in a binary way, we acknowledge that the true answer is often more nuanced, so please just use your best judgment and write a justification to elaborate. All supporting evidence can appear either in the main paper or the supplemental material, provided in appendix. If you answer [Yes] to a question, in the justification please point to the section(s) where related material for the question can be found.

IMPORTANT, please:

- **Delete this instruction block, but keep the section heading "NeurIPS Paper Checklist",**
- **Keep the checklist subsection headings, questions/answers and guidelines below.**
- **Do not modify the questions and only use the provided macros for your answers**.


# A  Broader impacts

The primary goal of PAC Bench is to catalyze the development of more capable, reliable, and physically grounded VLMs and their fine-tuned variants, often called VLAs for real-world robotic applications. Because VLA fine-tuning typically relies on low-level trajectory data rather than higher level reasoning, probing the underlying VLM's understanding of object Properties, action Affordances, and physical Constraints (PAC) gives us a grounded lens into the capabilities that downstream robotic policies will inherit. By diagnosing PAC weaknesses in the base model, researchers can distinguish whether a VLA's performance stems from genuine physical common sense or simply memorized motion patterns, and thus guide targeted improvements in model architectures, training methodologies, and dataset curation. In doing so, PAC Bench helps ensure that robotic systems become more predictable, less prone to errors from a lack of physical understanding, and better equipped for safe, effective collaboration in complex, everyday environments.

By providing a fine-grained diagnostic tool, PAC Bench can help researchers and developers identify specific weaknesses in current models, thereby guiding targeted improvements in model architectures, training methodologies, and dataset curation. This, in turn, can lead to robotic systems that are more predictable, less prone to errors stemming from a lack of physical common sense, and better able to perform a wide range of useful tasks. The open release of our benchmark and its diverse data sources (including web-scale images, real-world humanoid captures, and simulated scenarios) is intended to foster broad community engagement and accelerate progress in this crucial area of AI.

While any advancement in AI capabilities warrants ongoing consideration of its societal implications, our work focuses on enhancing the fundamental understanding and robustness of AI systems, which we see as a positive step towards more responsible AI development. We encourage the community to leverage PAC Bench to build systems that not only demonstrate impressive capabilities but also operate with a clear and verifiable understanding of their physical environment, ultimately contributing to the beneficial integration of AI into society.

# B  Dataset Statistics

Carey and Spelke, in their work on *core knowledge* notes that "the perceptual and action capacities of humans result not from one general-purpose system for perceiving or acting, but from the orchestration of distinct, specialized systems for perceiving different kinds of environmental properties (e.g., color, depth, melodies, etc.) and for engaging in different patterns of activity (e.g., reaching, grasping, locomoting, scanning a scene)" [38].

> In the first few months of life, infants begin by understanding basic object **properties**; they then explore what actions those objects **afford**, and only later, through trial and error, do they learn the physical **constraints** that govern whether those actions can be successfully executed to achive a task. We test VLMs on these three distinct cognitive capabilities required to complete a robot manipulation task.

## B.1  Properties

**Real Robo**

The *Real Robo* properties subset contains 785 annotations spread across 67 unique scenario image–pairs, giving a mean of 11.7 annotated properties per scenario (the schema expects 12).

**Property–name frequency.**  Every property except SEALING appears exactly 67 times, corresponding to 8.54 % of all annotations each. SEALING appears 48 times (6.11 %).

**Category distribution (overall).**  Non-consumable 67 (8.54 %), Medium thickness 63 (8.03 %), Non-sticky 55 (7.01 %), Contains 50 (6.37 %), Non-containable 38 (4.84 %), Horizontal 38 (4.84 %), Hard 36 (4.59 %), Simple 36 (4.59 %), High-density 34 (4.33 %), Light 33 (4.20 %), Multicolored 33 (4.20 %), Low-density 33 (4.20 %), Soft 31 (3.95 %), Multi-object 31 (3.95 %), Sealed 29 (3.69 %), Containable 29 (3.69 %), Monochromatic 29 (3.69 %), Unsealed 19 (2.42 %), Vertical 19 (2.42

%), Empty 17 (2.17 %), Thick 16 (2.04 %), Thin 16 (2.04 %), Multi-directional 10 (1.27 %), Sticky 7 (0.89 %), Heavy 6 (0.76 %), Metallic 5 (0.64 %), Variable 5 (0.64 %).

**Category distribution per property.** CAPACITY: Non-containable 38 (56.7 %), Containable 29 (43.3 %). COLOR: Multicolored 33 (49.3 %), Monochromatic 29 (43.3 %), Metallic 5 (7.5 %). COMPLEXITY: Simple 36 (53.7 %), Multi-object 31 (46.3 %). CONSUMABILITY: Non-consumable 67 (100 %). CONTENTS: Contains 50 (74.6 %), Empty 17 (25.4 %). DENSITY: High-density 34 (50.8 %), Low-density 33 (49.2 %). HARDNESS: Hard 36 (53.7 %), Soft 31 (46.3 %). ORIENTATION: Horizontal 38 (56.7 %), Vertical 19 (28.4 %), Multi-directional 10 (14.9 %). SEALING: Sealed 29 (60.4 %), Unsealed 19 (39.6 %). STICKINESS: Non-sticky 55 (82.1 %), Sticky 7 (10.4 %), Variable 5 (7.5 %). THICKNESS: Medium 35 (52.2 %), Thick 16 (23.9 %), Thin 16 (23.9 %). WEIGHT: Light 33 (49.3 %), Medium 28 (41.8 %), Heavy 6 (9.0 %).

**Descriptor distribution (overall).** Solid 74 (4.71 %); Reusable 67, Permanent 67 (4.27 % each); Lightweight 66 (4.20 %); Balanced 63 (4.01 %); Smooth 55, Slippery 55 (3.50 % each); Filled 50, Occupied 50 (3.18 % each); Dense 40 (2.55 %); Flat 38, Reclined 38, Unperforated 38 (2.42 % each); Rigid 36, Single-unit 36, Monolithic 36 (2.29 % each); Standard Thickness 35 (2.23 %); Compact 34 (2.17 %); Gradient 33, Striped 33, Featherweight 33, Buoyant 33 (2.10 % each); Assembled 31, Interconnected 31, Plush 31, Flexible 31 (1.97 % each); Airtight 29, Watertight 29, Single Color 29, Neutral 29, Hollow 29, Enclosable 29 (1.85 % each); Moderate 28 (1.78 %); Bulky 22 (1.40 %); Upright 19, Standing 19, Open 19, Can-leak 19 (1.21 % each); Vacant 17, Void 17 (1.08 % each); Sturdy 16, Slim 16, Minimal Thickness 16 (1.02 % each); Rotational 10, Adjustable 10 (0.64 % each); Adhesive 7, Tacky 7 (0.45 % each); Glossy 5, Shiny 5, Temporary Stickiness 5, Conditional Adhesion 5 (0.32 % each).

**Descriptor distribution per property & category.** Each category listed above is characterised by exactly two descriptors, each accounting for half of the annotations in that category—for example, Containable objects are equally annotated as *Hollow* and *Enclosable*, Metallic objects as *Glossy* and *Shiny*, Hard objects as *Solid* and *Rigid*, and so on across all 27 property–category pairs.

**Robocasa**

The *Robocasa* synthetic-properties subset comprises 424 property annotations describing 41 distinct household objects (≈10.3 properties per object).

**Property–name frequency.** Eight properties (WEIGHT, COLOR, HARDNESS, CONSUMABILITY, COMPLEXITY, THICKNESS, DENSITY, STICKINESS) appear once for every object (41 annotations each, 9.67 % apiece). CAPACITY appears 39 times (9.20 %), CONTENTS 38 (8.96 %), and SEALING 19 (4.48 %).

**Category distribution (overall).** Medium 36 (8.49 %), Non-sticky 36 (8.49 %), Contains 33 (7.78 %), Simple 31 (7.31 %), Non-containable 24 (5.66 %), Consumable 24 (5.66 %), Monochromatic 23 (5.42 %), High-density 21 (4.95 %), Low-density 20 (4.72 %), Hard 20 (4.72 %), Multicolored 18 (4.25 %), Light 18 (4.25 %), Soft 17 (4.01 %), Non-consumable 17 (4.01 %), Containable 15 (3.54 %), Sealed 13 (3.07 %), Thick 10 (2.36 %), Multi-object 10 (2.36 %), Heavy 10 (2.36 %), Thin 8 (1.89 %), Unsealed 6 (1.42 %), Empty 5 (1.18 %), Brittle 4 (0.94 %), Sticky 3 (0.71 %), Variable 2 (0.47 %).

**Category distribution per property.** CAPACITY: Non-containable 24 (61.5 %), Containable 15 (38.5 %). COLOR: Monochromatic 23 (56.1 %), Multicolored 18 (43.9 %). COMPLEXITY: Simple 31 (75.6 %), Multi-object 10 (24.4 %). CONSUMABILITY: Consumable 24 (58.5 %), Non-consumable 17 (41.5 %). CONTENTS: Contains 33 (86.8 %), Empty 5 (13.2 %). DENSITY: High-density 21 (51.2 %), Low-density 20 (48.8 %). HARDNESS: Hard 20 (48.8 %), Soft 17 (41.5 %), Brittle 4 (9.8 %). SEALING: Sealed 13 (68.4 %), Unsealed 6 (31.6 %). STICKINESS: Non-sticky 36 (87.8 %), Sticky 3 (7.3 %), Variable 2 (4.9 %). THICKNESS: Medium 23 (56.1 %), Thick 10 (24.4 %), Thin 8 (19.5 %). WEIGHT: Light 18 (43.9 %), Medium 13 (31.7 %), Heavy 10 (24.4 %).

**Descriptor distribution (overall).** Solid 44 (5.05 %); Lightweight 38 (4.36 %); Balanced 36, Smooth 36, Slippery 36 (4.13 % each); Filled 33, Occupied 33 (3.78 % each); Single-unit 31,

Monolithic 31, Dense 31 (3.56 % each); Edible 24, Burnable 24, Disposable 24, Unperforated 24 (2.75 % each); Single Color 23, Neutral 23, Standard Thickness 23 (2.64 % each); Compact 21 (2.41 %); Buoyant 20, Bulky 20, Rigid 20 (2.29 % each); Featherweight 18, Gradient 18, Striped 18 (2.06 % each); Plush 17, Flexible 17, Reusable 17, Permanent 17 (1.95 % each); Hollow 15, Enclosable 15 (1.72 % each); Moderate 13, Airtight 13, Watertight 13 (1.49 % each); Sturdy 10, Assembled 10, Interconnected 10 (1.15 % each); Slim 8, Minimal Thickness 8 (0.92 % each); Open 6, Can-leak 6 (0.69 % each); Vacant 5, Void 5 (0.57 % each); Fragile 4, Breakable 4 (0.46 % each); Adhesive 3, Tacky 3 (0.34 % each); Temporary Stickiness 2, Conditional Adhesion 2 (0.23 % each).

**Descriptor distribution per property & category.** The synthetic generator enforces symmetric pairings: every category co-occurs with exactly two descriptors that split its count evenly—for instance, Containable objects are half *Hollow* and half *Enclosable*; Consumable items distribute equally among *Edible*, *Burnable*, and *Disposable*; Brittle objects are evenly *Fragile* and *Breakable*; analogous 50 % pairings hold across all remaining property–category combinations.

## OpenImages

The *OpenImages* split aggregates 10 506 property annotations covering 679 everyday-object images for each of the 12 properties, i.e. 8 148 image–property pairs in total. Annotator effort is uneven but broad: Annot.,7 contributed 2 037 labels (19.4 %), Annot.,4 — 1 928 (18.4 %), Annot.,11 — 1 821 (17.3 %), Annot.,1 and 9 — 1 358 each (12.9 % ea.), Annot.,10 — 694 (6.6 %), Annot.,5 — 585 (5.6 %), Annot.,3 — 319 (3.0 %), Annot.,8 — 214 (2.0 %), Annot.,6 — 192 (1.8 %).

**Capacity.** All 679 images carry a CAPACITY label: Containable 321 (47.28 %), Non-containable 317 (46.69 %), Don't-know 35 (5.15 %), Not-applicable 6 (0.88 %). Descriptors cluster in two symmetrical pairs—*Hollow*/*Enclosable* (321 each, 25.16 % apiece) and *Solid*/*Unperforated* (317 each, 24.84 %).

**Color.** 818 colour judgements (often double-annotated) span the same image set. Categories: Multicolored 300 (36.67 %), Metallic 260 (31.78 %), Monochromatic 188 (22.98 %), Matte 59 (7.21 %), Don't-know 10 (1.22 %), Not-applicable 1. Descriptors: *Gradient* and *Striped* 300 each (18.59 %), *Glossy* and *Shiny* 260 each (16.11 %), *Single Color* 188 (11.65 %).

**Complexity.** 1 140 annotations—Multi-object 883 (77.46 %), Simple 242 (21.23 %), Don't-know 10, Invalid-format 5. Descriptors: *Assembled / Interconnected* 883 each (39.24 %), *Single-unit / Monolithic* 242 each (10.76 %).

**Consumability.** Every image is labelled once: Non-consumable 633 (93.23 %), Consumable 41 (6.04 %), Invalid-format 4, Not-applicable 1. Descriptors split into reusable pairs—*Reusable*/*Permanent* 633 each (45.57 %) versus *Edible*/*Burnable*/*Disposable* 41 each (2.95 %).

**Contents.** 679 labels: Contains 249 (36.67 %), Empty 149 (21.94 %), Not-applicable 149 (21.94 %), Don't-know 130 (19.15 %), Invalid-format 2. Descriptors: *Filled*/*Occupied* 249 each (31.28 %); *Vacant*/*Void* 149 each (18.72 %).

**Density.** High-density 412 (60.68 %), Low-density 248 (36.52 %), Not-applicable 12, Don't-know 6, Variable 1. Descriptors mirror the split—*Dense*/*Compact* 412 each (31.16 %) versus *Lightweight*/*Buoyant* 248 each (18.76 %); one image is uniquely *Adjustable*.

**Hardness.** Hard 297 (43.74 %), Brittle 160 (23.56 %), Don't-know 126 (18.56 %), Soft 86 (12.67 %), Not-applicable 10. Descriptor pairs: *Solid*/*Rigid* 297 each (27.35 %), *Fragile*/*Breakable* 160 each (14.73 %), *Plush* 86 (7.92 %).

**Orientation.** Vertical 496 (55.92 %), Horizontal 241 (27.17 %), Multi-directional 70 (7.89 %) plus 70 identical Invalid-format rows, Don't-know 8, Not-applicable 2. Descriptors: *Upright*/*Standing* 496 each (30.73 %), *Flat*/*Reclined* 241 each (14.93 %), *Rotational* 70 (4.34 %).

**Sealing.** Unsealed 495 (56.83 %), Sealed 351 (40.30 %), Don't-know 16 (1.84 %), Not-applicable 5 (0.57 %), Invalid-format 4 (0.46 %). Descriptors partition cleanly: *Open*/*Can leak* 495 each (29.26 %), *Airtight*/*Watertight* 351 each (20.74 %).

**Stickness.** 1 358 labels (two annotators × all images): Non-sticky 1 097 (80.78 %), Sticky 244 (17.97 %), Don't-know 15, Variable 2. Descriptors: *Smooth*/*Slippery* 1 097 each (40.84 %), *Adhesive*/*Tacky* 244 each (9.08 %), *Temporary Stickiness* 2 (0.07 %).

**Thickness.** Thick 258 (38.00 %), Medium 220 (32.40 %), Thin 163 (24.01 %), Not-applicable 27 (3.98 %), Don't-know 11 (1.62 %). Descriptors: *Sturdy*/*Bulky* 258 each (20.12 %), *Standard Thickness*/*Balanced* 220 each (17.16 %), *Slim* 163 (12.71 %).

**Weight.** Heavy 482 (35.49 %), Light 443 (32.62 %), Medium 426 (31.37 %), Not-applicable 3, Don't-know 2, Dynamic 2. Descriptors: *Bulky*/*Dense* 482 each (17.81 %), *Featherweight*/*Lightweight* 443 each (16.37 %), *Moderate* 426 (15.74 %).

## B.2 Affordance

### OpenImages

Across 116 objects every image is annotated once, giving 116 affordance rows produced by seven annotators. Most images list three affordances (61 entries, 52.6 %), 50 list two (43.1 %), four list one (3.5 %) and one lists none. The ten most frequent affordances are: *Hold* 36 (12.5 %), *Holding* 11 (3.8 %), *Open/Close* 9 (3.1 %), *Cook* 9 (3.1 %), *Turn on/off* 8 (2.8 %), *Hold items* 6 (2.1 %), *Pour* 6 (2.1 %), *Fill* 5 (1.7 %), *Manipulating controls* 4 (1.4 %) and *Hold food* 4 (1.4 %).

### Real Robot

Sixty-eight scenario pairs each have one affordance row, totalling 68 sets. Half of the scenarios list three affordances (34, 50 %), 29 list two (42.7 %) and five list one (7.4 %). Across all 170 recorded affordance slots the most common actions are: *act as weight* 29 (17.6 %), *Contain things* 12 (7.3 %), *scrape things* 10 (6.1 %), *stick things* 7 (4.2 %), *add thickness* 7 (4.2 %), *act as cushion* 7 (4.2 %), followed by fifteen further affordances occurring five or six times each. Slot-wise patterns highlight typical triplets such as *Contain things / act as cushion / act as weight* (7 cases, 10.3 %), and frequent pairs like *stick things / add thickness* or *break things / act as weight*. Slot 3 is often left blank (34 empty entries, 50 %).

### Robocasa

The synthetic set covers 41 household objects. Eight objects list a single affordance (19.5 %), eighteen list two (43.9 %) and fifteen list three (36.6 %), giving 89 affordance mentions overall. *edible* dominates slot 1 (23 occurrences, 56.1 %) and is the single most frequent affordance overall (24, 27 %). Other common actions are *cookable* 10 (11.2 %), *garnish* and *can be used to stir things* 4 each (4.5 %), *can contain things*, *can be used to pour things*, *stackable* and *can be contain things* 3 each (3.4 %). All remaining 26 affordances appear once or twice ($\leq$2.5 % each), illustrating the long-tail synthetically injected diversity. The most common triplet is *edible / cookable / $\emptyset$* (10 objects, 24.4 %), followed by *edible / $\emptyset$ / $\emptyset$* (8, 19.5 %).

## B.3 Constraints

### Real Robo

This constraint Dataset contains 53 question–answer pairs, one per scenario. The nine distinct questions appear with the following frequencies: "Can we keep the ball inside the penstand?" 13 (24.53 %); "Can we keep the pen inside the penstand?" 10 (18.87 %); "Can you keep the food on the plate?" 8 (15.09 %); "Can you reverse the stacking of the objects?" 8 (15.09 %); "Can you write on the notepad using the marker?" 6 (11.32 %); "Can the robot stack the object near the right hand on the object near the left hand?" 4 (7.55 %); "Can the robot stack the object near the left hand on the object near the right hand?" 2 (3.77 %); "Can the robot stack the object away from it on the object near it?" 1 (1.89 %); and the lower-case duplicate "can you keep the food on the plate?" 1 (1.89 %).

All responses are negative and distributed across nineteen phrasings: "No the cube won't balance on the pyramid." 14 (26.42 %); "No the penstand is inverted." 11 (20.75 %); "No the the penstand is inverted." 4 (7.55 %); "No the penstand is not upright." 4 (7.55 %); "No the box is on the plate." 2 (3.77 %); "No the the penstand is not upright." 2 (3.77 %); "No the plate is inverted." 2 (3.77 %); "No the marker is closed." 2 (3.77 %); "No the notepad is inside the cup." 2 (3.77 %); plus nine single-occurrence answers covering cube–pyramid balance, covered openings, inverted or closed objects, and misplaced items.

Keyword extraction highlights the chief obstacles: "penstand" 23 mentions, "inverted" 20, and the instability trio "cube/balance/pyramid" 15 each, followed by "box" 9, "plate" 8, "closed" 7, "upright" 6, and sporadic references to notepad, cup, marker, inside, covered openings, under-placement and table contact.

Mapping these words to constraint types shows that inverted-orientation issues account for 20 cases (37.74 %); balance on a pyramid for 15 (28.30 %); object closure for 7 (13.21 %); non-upright alignment for 6 (11.32 %); containment failures ("inside", "covered", "under", "on table") and other special cases each represent ≤4 % of the set. Overall, tasks are blocked chiefly because penstands or plates are upside-down, cubes cannot balance on pyramids, or target objects are sealed or mis-aligned.

**Mujoco**

The Mujoco constraint Dataset contains 4 sub domains wach with 3 camera views. For each view we sample 10 different scenes configurations.

## C  Experimental Setup

This appendix provides further details on the experimental setup used for collecting data and for evaluating VLMs on PAC Bench, complementing Section 4.1 of the main paper.

### C.1  Models Evaluated and Access

The VLM evaluations reported in this paper (Section 4) encompass a diverse suite of models. All models were accessed via their respective APIs available through the OpenRouter service[4] between April 2024 and May 2024. The specific models evaluated are detailed below, along with their OpenRouter paths:

1. **Claude 3.7 Sonnet:** `https://openrouter.ai/anthropic/claude-3.7-sonnet`
2. **Claude 3.7 Sonnet (T):** `https://openrouter.ai/anthropic/claude-3.7-sonnet:thinking` *(This denotes Chain-of-Thought prompting applied to the Claude 3.7 Sonnet model.)*
3. **Claude 3.5 Sonnet:** `https://openrouter.ai/anthropic/claude-3.5-sonnet`
4. **Gemini 2.0 Flash 001:** `https://openrouter.ai/google/gemini-2.0-flash-001`
5. **Gemini 2.5 Flash P:** `https://openrouter.ai/google/gemini-2.5-flash-preview`
6. **Gemini 2.5 Pro P:** `https://openrouter.ai/google/gemini-2.5-pro-preview-03-25`
7. **GPT-4.1:** `https://openrouter.ai/openai/gpt-4.1`
8. **o4-mini-high:** `https://openrouter.ai/openai/o4-mini-high` *(Note: The "(T)" for this model in some tables also indicates Chain-of-Thought prompting.)*
9. **GPT-4.1 Mini:** `https://openrouter.ai/openai/gpt-4.1-mini`
10. **Llama 4 Maverick:** `https://openrouter.ai/meta-llama/llama-4-maverick`
11. **Llama 4 Scout:** `https://openrouter.ai/meta-llama/llama-4-scout`
12. **Llama 3.2 90B VI:** `https://openrouter.ai/meta-llama/llama-3.2-90b-vision-instruct` *(VI denotes Vision Instruct. Your tables may use Llama 3.2 90B Vision I)*

---

[4] `https://openrouter.ai/`

13. **Grok 2 Vision:** `https://openrouter.ai/x-ai/grok-2-vision-1212`

14. **Grok Vision Beta:** `https://openrouter.ai/x-ai/grok-vision-beta`

15. **Qwen2.5 VL:** `https://openrouter.ai/qwen/qwen2.5-vl-72b-instruct` *(VL denotes Vision Language.)*

16. **Qwen VL Plus:** `https://openrouter.ai/qwen/qwen-vl-plus`

17. **Qwen 3 (235B):** `https://openrouter.ai/qwen/qwen3-235b-a22b` *(This appears as "Qwen 3".)*

## C.2 Simulated Constraint Scenario Generation

To generate a diverse and controllable set of scenarios for evaluating VLM understanding of physical constraints, we developed a simulation-based pipeline using the MuJoCo physics engine. This approach allows for the systematic creation of situations where specific physical limitations are the primary factor determining task feasibility. Our design focused on four primary constraint domains critical for robotic manipulation:

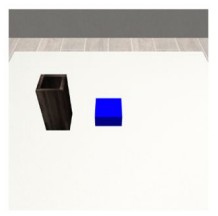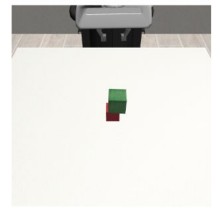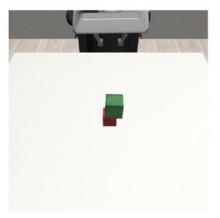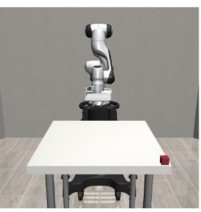

Figure 6: Example scenes corresponding to each constraint domain (left to right): (a) *Impossible Placement*: attempting to fit the blue block into the brown box; (b) *Occlusion/Support Issues*: picking up the red block beneath another; (c) *Stability Constraints*: lifting the unstable green block from the top of a stack; (d) *Reachability and Access Constraints*: grasping a block placed at the very edge of the workspace.

- **Impossible Placement:** These scenarios test the understanding of basic stability and geometric fit. Examples include attempting to place a larger object into a smaller container or balancing an object on an inadequately small or unstable surface. Objects with varying shapes (e.g., cubes, spheres, custom meshes) and sizes were utilized.

  *Prompt:* "I want to place the blue object inside the brown box, are there any constraints that prevent me from doing so or is it possible?"
  *Verification Prompt:* "Does the following text say that the task is impossible? Answer only with 'true' or 'false'."

- **Occlusion/Support Issues:** These scenarios evaluate reasoning about object interdependencies and access. A typical setup involves a target object being either partially or fully occluded by another object, or where the target object is currently supporting another, making its removal problematic (e.g., preventing a stack from toppling if a lower block is removed).

  *Prompt:* "I want to pick up the red object from the bottom of the stack, are there any constraints that prevent me from doing so or is it possible?"
  *Verification Prompt:* "Does the following text say that object on the top might fall due to it's placement? Answer only with 'true' or 'false'."

- **Stability Constraints:** These focus on the inherent stability of an object or an assembly if an action is performed. Examples include attempting to pick a block from an unstable stack where the act of picking itself or the removal of the object leads to the collapse of the remaining structure, or attempting to place an object such that the resulting configuration is unstable.

  *Prompt:* "I want to pick up the green object from the top of the stack, are there any constraints that prevent me from doing so or is it possible?"
  *Verification Prompt:* "Does the following text say that object on the top might fall due to it's placement? Answer only with 'true' or 'false'."

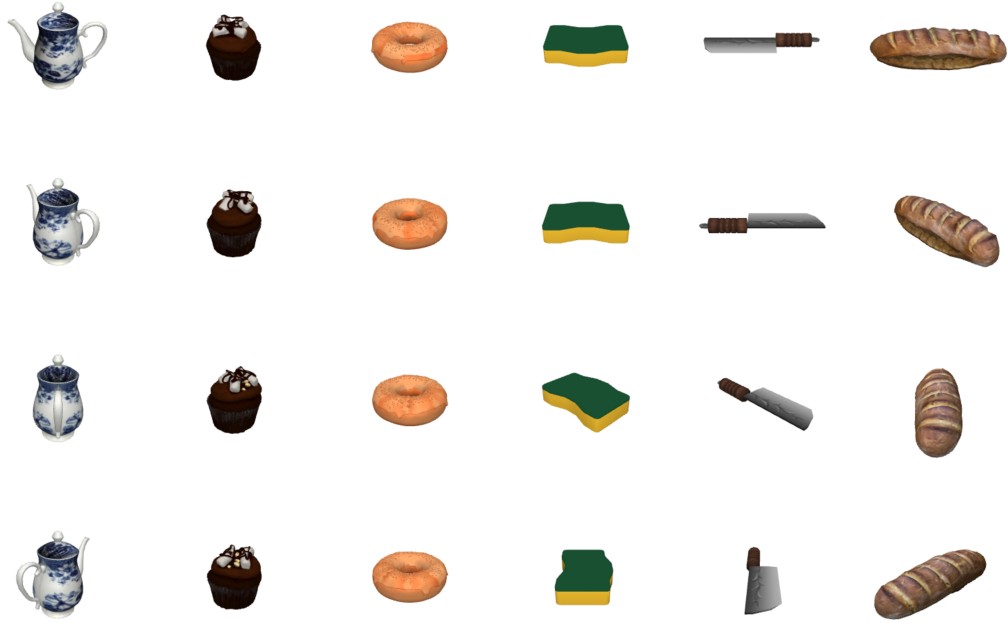

Figure 7: Samples from robocasa datapoint in PACBench

- **Reachability and Access Constraints:** These scenarios test understanding of spatial and kinematic limitations. Objects might be placed at the edge of a workspace, behind obstacles, or in orientations that make them difficult or impossible for a standard robotic gripper to access without collision or exceeding plausible joint limits.

    *Prompt:* "I want to pick up the red object from the edge of the stack, are there any constraints that prevent me from doing so or is it possible?"
    *Verification Prompt:* "Does the following text say that object is out of reach? Answer only with 'true' or 'false'."

For each of these four domains, we procedurally generated **10 distinct environment instantiations**. Randomization was applied to object properties (e.g., slight variations in size and mass where relevant for dynamics), initial positions and orientations, as well as the placement of minor distractor objects to increase visual diversity while ensuring the core constraint remained salient.

Figure 6 provides a visual summary of one example from each sub-domain.

### C.3   Synthetic Object-Centric Dataset from RoboCasa Assets

To support fine-grained object reasoning evaluations, we constructed a synthetic image dataset by curating a subset of authentic 3D meshes from the RoboCasa simulation framework. While RoboCasa provides a rich large-scale kitchen environment with hundreds of AI-generated and hand-modeled assets, we selected only the 45 objects that had artist-modeled meshes (i.e., excluding purely AI-generated models). Each object is paired with high-resolution renders, manual affordance annotations, and detailed physical/property labels.

- **Asset Selection:** We chose 45 common kitchen and tabletop items, spanning food-stuffs, containers, utensils, and small appliances. The full set is: apple, baguette, beer, bottled_water, bowl, boxed_food, broccoli, candle, cereal, cheese, chocolate, corn, croissant, cucumber, cupcake, cutting_board, donut, egg, eggplant, jug, ketchup, kettle_non_electric, knife, lime, liquor, milk, onion, orange, pan, peach, pot, potato, shaker,

```
spatula, sponge, spoon, spray, sweet_potato, tangerine, teapot,
tomato, tray, waffle, wine, yogurt.
```

- **Viewpoint Sampling:** Each object was rendered from **24** distinct viewpoints by rotating the camera around the object's vertical axis (Z) at three elevations $(-30°, 0°, +30°)$ and eight azimuths $(0°, 45°, \ldots, 315°)$. Filenames follow the pattern:

$$\texttt{elev<elevation>\_azim<azimuth>.png}$$

for example `elev-30_azim135.png`, yielding $45 \times 24 = 1080$ high-resolution images.

- **Affordance Annotation and Evaluation:** We hand-annotated **41** of the 45 objects with one or more affordances (e.g., *edible*, *pourable*, *stackable*). To probe model understanding, we used the prompt:

```
List all the possible affordances of a <object_name>.
An affordance is what an object can be used for or what
actions can be performed with it.  List them in a clear,
comma-separated format.
```

We then computed two strict metrics:

1. **All-correct:** Does the LLM output contain *all* ground-truth affordances?
2. **At-least-one:** Does the LLM output contain *at least one* ground-truth affordance?

Verification prompts were:

```
Given the following ground truth affordances for a
<object_name>:  <list>
And the following LLM response:  <llm_response>
Does the LLM response contain all the ground truth
affordances?  Answer only with 'true' or 'false'.

Given the following ground truth affordances for a
<object_name>:  <list>
And the following LLM response:  <llm_response>
Does the LLM response contain at least one of the ground
truth affordances?  Answer only with 'true' or 'false'.
```

- **Property Annotation and Evaluation:** We manually labeled each object with up to 11 physical and functional properties: COLOR, COMPLEXITY, CONSUMABILITY, DENSITY, HARDNESS, STICKINESS, THICKNESS, WEIGHT, CAPACITY, CONTENTS, and SEALING. Table 5 summarizes the number of objects annotated per property. For example, *yogurt* was annotated as:

```
yogurt|WEIGHT|Medium|Moderate, Balanced
yogurt|COLOR|Multicolored|Gradient, Striped
yogurt|HARDNESS|Hard|Solid, Rigid
...
```

Each property uses a predefined set of discrete options and synonyms. We defined:

```
WEIGHT_options = """
Light: Featherweight, Lightweight
Medium: Moderate, Balanced
Heavy: Bulky, Dense
Dynamic: Fluctuating, Variable
"""

COLOR_options = """
Monochromatic: Single Color, Neutral
Multicolored: Gradient, Striped
Metallic: Glossy, Shiny
Matte: Flat, Dull
"""

HARDNESS_options = """
```

```
Hard: Solid, Rigid
Soft: Plush, Flexible
Brittle: Fragile, Breakable
"""

ORIENTATION_options = """
Vertical: Upright, Standing
Horizontal: Flat, Reclined
Multi-directional: Rotational, Adjustable
"""

CONSUMABILITY_options = """
Consumable: Edible, Burnable, Disposable
Non-consumable: Reusable, Permanent
"""

COMPLEXITY_options = """
Simple: Single-unit, Monolithic
Multi-object: Assembled, Interconnected
"""

CAPACITY_options = """
Containable: Hollow, Enclosable
Non-containable: Solid, Unperforated
"""

CONTENTS_options = """
Contains: Filled, Occupied
Empty: Vacant, Void
"""

SEALING_options = """
Sealed: Airtight, Watertight
Unsealed: Open, can leak
"""

DENSITY_options = """
High-density: Dense, Compact
Low-density: Lightweight, Buoyant
Variable: Adjustable, Fluid
"""

THICKNESS_options = """
Thin: Slim, Minimal Thickness
Medium: Standard Thickness, Balanced
Thick: Sturdy, Bulky
"""

STICKINESS_options = """
Sticky: Adhesive, Tacky
Non-sticky: Smooth, Slippery
Variable: Temporary Stickiness, Conditional Adhesion
"""
```

Models were queried with the following template:

```
Evaluate the {property} of the object(s) enclosed within the
red bounding box in the image.
Respond with only one of the following options:
```

```
{options}
 Provide no additional text, explanations, or numbers.
```

| Property | # Objects Annotated |
|---|---|
| COLOR | 41 |
| COMPLEXITY | 41 |
| CONSUMABILITY | 41 |
| DENSITY | 41 |
| HARDNESS | 41 |
| STICKINESS | 41 |
| THICKNESS | 41 |
| WEIGHT | 41 |
| CAPACITY | 39 |
| CONTENTS | 38 |
| SEALING | 19 |

Table 5: Number of objects annotated per property.

Overall, this dataset comprises **1080** images of 45 objects, enriched with manual affordance and property labels, enabling comprehensive evaluation of VLM performance on view-invariant recognition, affordance inference, and property classification tasks.

### C.4   Open Images V7 Subset for Object-Centric Affordance and Property Evaluation

Open Images V7 is a comprehensive, real-world image corpus of approximately 1.9 million images spanning 600 object classes, annotated with image-level labels, bounding boxes, segmentation masks, visual relationships, and localized narratives. From this large-scale dataset, we selected 116 object classes for which single-instance examples could be clearly isolated and annotated. For each class, we sampled between four and eight representative images, yielding a total of 679 unique frames. Filenames conform to the pattern `<object_id>_<image_id>.jpg` (e.g. `012w5l_226957c99fab6ddf.jpg`), where the first token denotes the Open Images class identifier and the second is the image hash. In every image, exactly one instance of the target object is marked with a yellow bounding box (see Fig. 8).

To probe visual-language models' understanding of object affordances, we gathered human annotations at the class level, specifying between one and three affordances per object (for example, "Sit", "Pour", or "Cut"). These annotations were recorded in CSV form as `object,affordance1,affordance2,affordance3`, resulting in over 300 total affordance entries across the 116 classes. Model outputs are evaluated under two strict criteria: (1) whether all ground-truth affordances appear in the response ("all-correct"), and (2) whether at least one ground-truth affordance appears ("at-least-one"). Verification is automated via prompts that present the ground-truth list alongside the model's response and request a single answer of "true" or "false."

In addition to affordances, we annotated each image for up to 15 physical and functional properties (COLOR, COMPLEXITY, CONSUMABILITY, DENSITY, HARDNESS, STICKINESS, THICKNESS, WEIGHT, CAPACITY, CONTENTS, SEALING, ORIENTATION, plus four domain-specific traits). Over 12,421 annotation entries were collected, corresponding to 10,506 unique (image, property) pairs—some images received multiple annotations for the same property. The distribution of annotations per property file is summarized below:

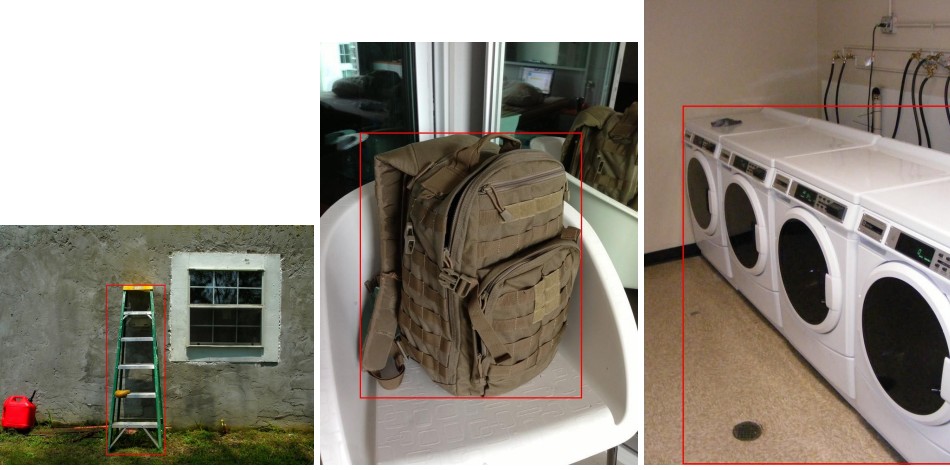

Figure 8: Example from our Open Images subset: a single object annotated with a red bounding box.

| Property File | Lines |
|---|---|
| property_CAPACITY_.csv | 679 |
| property_COLOR_.csv | 818 |
| property_COMPLEXITY_.csv | 1140 |
| property_CONSUMABILITY_.csv | 679 |
| property_CONTENTS_.csv | 679 |
| property_DENSITY_.csv | 679 |
| property_HARDNESS_.csv | 679 |
| property_ORIENTATION_.csv | 887 |
| property_SEALING_.csv | 871 |
| property_STICKINESS_.csv | 1358 |
| property_THICKNESS_.csv | 679 |
| property_WEIGHT_.csv | 1358 |

Models are queried with the template:

```
Evaluate the {property} of the object(s) enclosed within the
red bounding box in the image.
Respond with only one of the following options:  {options}
Provide no additional text, explanations, or numbers.
```

Because Open Images V7 comprises 600 classes and nearly two million images, this protocol can be extended seamlessly to new categories and additional examples. Once class-level affordance and property labels are established, any further images sampled under the same class identifier inherit those annotations, enabling scalable evaluation of view-invariant recognition, affordance inference, and physical attribute classification.

### C.5 Embodied Robot Capture: Unitree G1 Dual-Arm Dataset

To complement our web-sourced and simulated resources with truly embodied visual data, we collected a fresh corpus of interactions using a dual-arm Unitree G1 humanoid operating in an indoor laboratory. The robot was tele-operated or executed short, pre-programmed primitives at a standing workstation filled with diverse household objects that were *not* present in either our RoboCasa or Open-Images subsets, thereby increasing inter-dataset heterogeneity. Each scene was photographed simultaneously from two calibrated perspectives: an egocentric camera rigidly attached to the robot's head (1280 × 720 at 30 Hz) and a side-mounted static camera that offered a wider allocentric view of the workspace. The resulting paired images allow Vision–Language Models (VLMs) to be probed under both first- and third-person viewpoints—conditions that often lead to markedly different perceptual challenges in robotics.

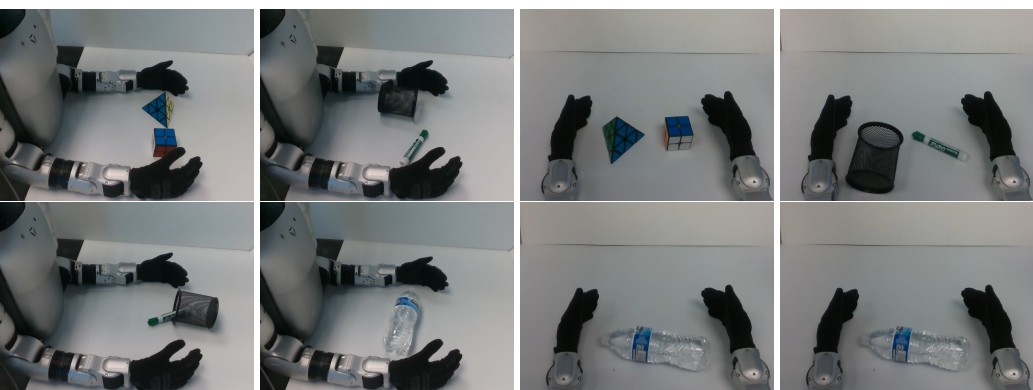

Figure 9: Samples from Unitree G1 humanoid from PacBench

**Property annotations.** For every object-centric tabletop configuration we recorded up to twelve physical and functional properties using the controlled vocabulary introduced in previous sections (e.g., WEIGHT, COLOR, SEALING). A total of 785 property rows were produced across 67 unique image pairs, giving an average of roughly twelve properties per scenario. All properties except SEALING are exhaustively annotated for every scene; SEALING appears in 48 of the 67 cases, reflecting either inapplicability or annotator uncertainty for the remaining scenes. Distributions are well balanced: for example, the WEIGHT axis splits into *Light* (49 %), *Medium* (42 %), and *Heavy* (9 %), while COLOR is almost evenly divided between *Monochromatic* and *Multicolored* with a small metallic tail. Descriptor-level statistics show that every categorical choice is accompanied by its canonical pair of synonyms (e.g., *Dense, Compact* whenever *High-density* is selected), a consequence of the structured drop-down interface used during labelling.

**Affordance annotations.** Sixty-eight scenarios were further enriched with up to three free-form affordances per object, resulting in 181 individual affordance strings. Half of the scenes list a full triplet, roughly 43 % include two entries, and only seven per cent contain a single affordance. The vocabulary is intentionally open; nevertheless several patterns emerge—"act as weight" accounts for 18 % of all mentions, followed by "contain things" and "scrape things." Frequent combinations such as *(contain things, act as cushion, act as weight)* illustrate that annotators naturally link physical support, compliance, and mass when reasoning about everyday artefacts. Evaluation uses the same strict "all-correct" and "at-least-one" metrics adopted for our other datasets, coupled with the verification prompts described earlier.

**Constraint annotations.** Finally, 53 of the scenarios include a natural-language question about the feasibility of a specific robot action together with a short justification when the answer is negative. These queries test spatial reasoning (e.g., balancing a cube on a pyramid), containment under orientation changes (placing items inside an inverted pen-stand), and accessibility issues (writing when a marker cap is closed). Recurrent keywords such as *inverted*, *balance*, *upright*, and *closed* reveal the dominant failure modes considered. Although the majority of responses start with a terse "No," the accompanying explanations provide fine-grained cues that are invaluable for evaluating whether a VLM can pinpoint the exact limiting factor.

**Cross-modal linking and usage.** Because every record—whether property, affordance, or constraint—references the same cam0_file / cam1_file pair, researchers can seamlessly join the three ground-truth tables to obtain a fully articulated description of each physical scene. This makes it possible to explore, for instance, how an object's annotated orientation (Vertical, Horizontal, Multi-directional) influences both its perceived affordances and the constraints imposed on manipulation tasks. The corpus therefore serves as a high-fidelity test-bed for embodied VLM evaluation, filling the gap between purely synthetic renders and images scraped from the web. In total, the Unitree G1 set delivers 67–68 richly annotated scenarios, amounting to hundreds of individual labels that capture the intertwined facets of Properties, Affordances, and Constraints from a truly robot-centric vantage point.

# D   Additional Model Evaluation Results

## D.1   Prompt Design

Notations like "(T)" or "CoT" in the result tables (e.g., for Claude 3.7 Sonnet (T), o4-mini-high (T)) indicate the application of a Chain-of-Thought prompting strategy, where models were explicitly instructed to "think step by step" or provide reasoning before their final answer. The syntax for prompts are shown in Section  C.2 C.3 C.4

## D.2   Properties Evaluations

Beyond direct querying, we investigated the influence of prompting strategies, specifically Chain-of-Thought (CoT), on the performance of VLMs in understanding object properties. Table 8 presents the accuracies for various models when employing CoT prompting, which can be compared against their direct query performance shown in Table 7 (our main property results table with new data).

Table 6: Properties accuracy (%) of leading VLMs across twelve distinct object property categories

| Model | P1 | P2 | P3 | P4 | P5 | P6 | P7 | P8 | P9 | P10 | P11 | P12 |
|---|---|---|---|---|---|---|---|---|---|---|---|---|
| Claude 3.5 Sonnet | 17.8 | 0.0 | 0.4 | 0.3 | 31.9 | 0.0 | 42.3 | 15.8 | 2.7 | 0.0 | 52.0 | 0.0 |
| Claude 3.7 Sonnet | 88.1 | 20.2 | 34.0 | 91.4 | 23.5 | 36.7 | 37.0 | 48.7 | 66.4 | 96.6 | 59.2 | 32.6 |
| Claude 3.7 Sonnet (T) | 81.3 | 6.7 | 38.4 | 93.8 | 22.3 | 9.0 | 23.4 | 24.0 | 50.9 | 73.8 | 46.2 | 15.0 |
| Gemini 2.0 Flash 001 | 59.4 | 19.7 | 84.8 | 7.0 | 35.3 | 58.0 | 43.9 | 57.6 | 56.1 | 38.2 | 24.3 | 40.8 |
| Gemini 2.5 Flash P | 54.9 | 26.9 | 47.3 | 11.0 | 28.8 | 40.1 | 31.1 | 41.1 | 58.9 | 74.5 | 29.2 | 27.1 |
| Gemini 2.5 Pro P** | 48.9 | 27.0 | 47.4 | 23.7 | 34.1 | 43.2 | 16.7 | 33.1 | 57.2 | 23.2 | 32.6 | 31.2 |
| Llama 3.2 90B Vision I | 35.6 | 13.1 | 33.3 | 1.3 | 14.8 | 25.0 | 12.8 | 47.5 | 30.2 | 23.1 | 26.8 | 4.2 |
| Llama 4 Maverick | 53.0 | 36.2 | 52.5 | 69.6 | 34.9 | 47.0 | 14.6 | 53.9 | 90.0 | 93.6 | 37.9 | 37.6 |
| Llama 4 Scout | 43.3 | 30.4 | 12.6 | 0.2 | 0.6 | 51.1 | 18.6 | 31.7 | 84.9 | 9.5 | 28.3 | 36.4 |
| GPT-4.1 Mini | 70.1 | 26.6 | 85.0 | 59.9 | 28.4 | 43.2 | 18.1 | 45.6 | 64.0 | 91.9 | 52.3 | 24.1 |
| GPT-4.1 | 10.9 | 13.8 | 38.1 | 5.3 | 29.0 | 25.9 | 27.8 | 42.3 | 91.0 | 35.3 | 37.0 | 4.4 |
| o4-mini-high (T) | 1.2 | 17.1 | 62.7 | 15.6 | 0.2 | 26.4 | 26.2 | 35.2 | 72.7 | 60.6 | 23.6 | 4.7 |
| Qwen VL Plus | 50.0 | 25.0 | 66.7 | 0.0 | 0.0 | 50.0 | 0.0 | 0.0 | 50.0 | 0.0 | 0.0 | 66.7 |
| Qwen2.5 VL | 53.2 | 21.9 | 34.2 | 9.0 | 20.7 | 9.6 | 42.3 | 57.1 | 61.8 | 70.7 | 66.6 | 18.7 |

**Chain-of-Thought Efficacy: A Mixed Bag for Property Recognition.** Our analysis reveals that the impact of CoT prompting on property recognition is model-dependent and not uniformly beneficial across all properties or models. For instance, 'Claude 3.7 Sonnet' shows a notable improvement with CoT on 'Sealing (P9)' (from 13.2% direct to 69.8% CoT) and 'Stickiness (P10)' (from 79.1% direct to 100.0% CoT). However, for the same model, CoT appears to slightly decrease performance on 'Density (P6)' (from 55.7% direct to 41.5% CoT). Its '(T)' variant in Table 8 (which is its CoT run) also shows improvements in some areas like 'Complexity (P3)'.

## D.3   Affordance Evaluations

Following Table 12 shows Accuracy (%) of VLMs on recognizing **atleast one affordances** for objects using **Single-Category Mapping** in PAC Bench. For the object classes 'Adhesive tape', 'Backpack', 'Band-aid', 'Bathroom accessory', 'Bathroom cabinet', 'Bathtub', 'Blender', 'Book', 'Bookcase', 'Bottle', 'Bowl', 'Box', 'Cabinetry', 'Can opener', 'Cart', 'Chair', 'Chest of drawers', 'Closet', 'Clothing', 'Coffeemaker', 'Container', 'Cooking spray', 'Countertop', 'Cupboard', 'Cutting board', 'Desk', 'Diaper', 'Dishwasher', 'Door', 'Door handle', 'Drawer', 'Drill (Tool)', 'Egg (Food)', 'Filing cabinet', 'Flashlight', 'Flowerpot', 'Food processor', 'Fork', 'Frying pan', 'Furniture', 'Gas stove', 'Glove', 'Grinder', 'Hammer', 'Home appliance', 'Infant bed', 'Jug', 'Kettle', 'Kitchen & dining room table', 'Kitchen appliance', 'Kitchen knife', 'Kitchen utensil', 'Knife', 'Ladder', 'Ladle', 'Laptop', 'Lavender (Plant)', 'Light bulb', 'Light switch', 'Measuring cup', 'Microwave oven', 'Milk', 'Mirror', 'Mixer', 'Mixing bowl', 'Mobile phone', 'Mug', 'Organ (Musical Instrument)', 'Oven', 'Paper towel', 'Pen', 'Pitcher (Container)', 'Plant', 'Plastic bag', 'Plate', 'Plumbing fixture', 'Power plugs and sockets', 'Pressure cooker', 'Refrigerator', 'Remote control', 'Scissors', 'Screwdriver', 'Serving tray', 'Shelf', 'Shower', 'Sink', 'Slow cooker', 'Soap dispenser', 'Spatula', 'Spice rack', 'Spoon', 'Stairs', 'Stool', 'Table', 'Tablet computer', 'Tableware', 'Tap', 'Toaster', 'Toilet', 'Toilet

Table 7: Properties Accuracy for Humanoid dataset

| Model | P1 | P2 | P3 | P4 | P5 | P6 | P7 | P8 | P9 | P10 | P11 | P12 |
|---|---|---|---|---|---|---|---|---|---|---|---|---|
| Claude 3.7 Sonnet | 74.6 | 47.8 | 47.3 | 93.0 | 30.3 | 55.7 | 55.7 | 59.7 | 13.2 | 79.1 | 39.3 | 48.3 |
| Claude 3.5 Sonnet | 83.6 | 50.2 | 48.8 | 89.6 | 28.9 | 52.7 | 55.2 | 58.7 | 19.4 | 83.6 | 42.8 | 50.7 |
| Gemini 2.0 Flash 001 | 76.6 | 55.2 | 49.3 | 63.2 | 39.8 | 46.8 | 54.7 | 41.3 | 38.2 | 66.7 | 53.2 | 40.3 |
| Gemini 2.5 Flash P | 71.6 | 53.2 | 56.2 | 74.1 | 27.9 | 40.3 | 63.2 | 65.2 | 37.5 | 41.8 | 42.3 | 33.8 |
| GPT-4.1* | 76.1 | 51.2 | 52.7 | 66.7 | 55.7 | 58.2 | 64.2 | 60.7 | 43.8 | 81.6 | 41.8 | 43.3 |
| GPT-4.1 Mini | 55.2 | 36.3 | 47.3 | 75.1 | 36.3 | 40.3 | 60.2 | 58.7 | 15.3 | 49.3 | 38.8 | 26.9 |
| Llama 4 Maverick | 82.1 | 43.8 | 46.3 | 82.6 | 77.1 | 57.7 | 54.2 | 47.8 | 40.3 | 62.7 | 40.8 | 59.2 |
| Llama 4 Scout | 81.6 | 51.2 | 45.8 | 62.2 | 60.2 | 43.3 | 51.2 | 54.2 | 36.1 | 73.6 | 44.3 | 37.8 |
| Llama 3.2 90B VI* | 59.7 | 37.3 | 36.3 | 39.3 | 51.2 | 44.8 | 37.3 | 39.8 | 27.1 | 56.7 | 16.9 | 31.3 |
| Qwen2.5 VL | 31.3 | 47.8 | 46.3 | 27.9 | 22.9 | 4.5 | 35.8 | 34.3 | 2.8 | 5.0 | 15.4 | 24.9 |
| Qwen VL Plus* | 25.4 | 15.4 | 28.4 | 22.9 | 20.4 | 39.3 | 34.3 | 29.4 | 31.3 | 12.4 | 14.4 | 5.5 |
| Grok 2 Vision | 69.7 | 49.3 | 45.8 | 53.7 | 82.6 | 40.3 | 56.7 | 55.2 | 11.1 | 78.6 | 37.3 | 31.8 |
| Grok Vision Beta* | 7.5 | 4.5 | 4.5 | 8.0 | 1.5 | 5.0 | 4.5 | 3.0 | 1.4 | 7.0 | 3.5 | 1.0 |

Table 8: Properties accuracy using **chain-of-thought (COT) prompting**. (**) Subset of properties evaluated.

| Model | P1 | P2 | P3 | P4 | P5 | P6 | P7 | P8 | P9 | P10 | P11 | P12 |
|---|---|---|---|---|---|---|---|---|---|---|---|---|
| Claude 3.5 Sonnet | 21.5 | 2.6 | 4.4 | 4.2 | 35.2 | 1.2 | 43.2 | 20.0 | 6.4 | 3.5 | 52.2 | 4.9 |
| Claude 3.7 Sonnet | 89.9 | 21.3 | 36.8 | 94.6 | 24.6 | 41.5 | 41.7 | 50.2 | 69.8 | 100.0 | 63.3 | 35.1 |
| Claude 3.7 Sonnet (T) | 84.5 | 9.9 | 41.0 | 94.2 | 25.9 | 10.9 | 28.2 | 24.8 | 54.4 | 78.3 | 48.0 | 16.6 |
| Gemini 2.0 Flash 001 | 62.5 | 20.6 | 86.0 | 7.8 | 36.5 | 62.5 | 48.5 | 60.3 | 57.9 | 39.9 | 24.9 | 44.8 |
| Gemini 2.5 Flash P | 57.6 | 30.3 | 47.8 | 15.5 | 30.4 | 43.2 | 32.9 | 45.8 | 60.3 | 76.4 | 31.8 | 28.4 |
| Gemini 2.5 Pro P** | – | 28.9 | – | – | – | 20.7 | – | – | – | – | 35.0 | – |
| Llama 3.2 90B Vision I | 36.1 | 17.8 | 37.8 | 4.4 | 18.5 | 27.6 | 12.9 | 51.0 | 34.3 | 23.5 | 27.9 | 6.4 |
| Llama 4 Maverick | 57.2 | 36.8 | 57.0 | 69.7 | 38.1 | 49.9 | 19.2 | 56.7 | 94.1 | 96.5 | 41.0 | 38.0 |
| Llama 4 Scout | 44.9 | 32.1 | 15.7 | 0.7 | 4.8 | 53.6 | 20.7 | 32.2 | 89.2 | 9.6 | 28.7 | 36.5 |
| GPT-4.1 Mini | 71.0 | 27.1 | 86.7 | 63.2 | 31.6 | 44.1 | 22.9 | 48.7 | 68.6 | 94.2 | 52.5 | 28.9 |
| GPT-4.1 | 11.1 | 18.4 | 39.8 | 5.4 | 31.8 | 29.5 | 32.7 | 47.2 | 93.5 | 38.0 | 39.6 | 8.6 |
| o4-mini-high | 4.1 | 21.5 | 66.3 | 16.4 | 0.6 | 27.8 | 30.5 | 35.9 | 75.2 | 62.0 | 26.3 | 8.1 |
| Qwen VL Plus | 53.4 | 26.3 | 71.2 | 2.7 | 1.8 | 50.1 | 1.5 | 2.6 | 54.0 | 3.0 | 3.2 | 68.7 |
| Qwen2.5 VL | 53.3 | 25.2 | 38.6 | 12.5 | 22.0 | 12.8 | 44.2 | 61.4 | 65.1 | 70.9 | 70.2 | 19.5 |

paper', 'Tool', 'Toothbrush', 'Torch', 'Towel', 'Toy', 'Waffle iron', 'Wardrobe', 'Washing machine', 'Waste container', 'Whisk', 'Window blind', 'Wok', 'Wood-burning stove', 'Wrench', 'Zucchini'.

| Object Class | Claude 3.5 S. | Claude 3.7 S. (T) | Claude 3.7 S. | Gemini 2.0 F001 | Gemini 2.5 FP | Gemini 2.5 PP | Llama 3.2 11B (1) | Llama 3.2 11B (2) | Llama 3.2 90B VI | Llama 4 Mav. | Llama 4 Sct. | GPT 4.1 Mini | GPT 4.1 | o4-mini-high (T) | Qwen VP | Qwen 2.5 VL | Grok 2 Vis. | Grok V. Beta |
|---|---|---|---|---|---|---|---|---|---|---|---|---|---|---|---|---|---|---|
| Adhesive Tape | 0.0 | 0.0 | 100.0 | 0.0 | 0.0 | 0.0 | 0.0 | 100.0 | 100.0 | 0.0 | 0.0 | 0.0 | 0.0 | 0.0 | 0.0 | 0.0 | 0.0 | 0.0 |
| Backpack | 0.0 | 0.0 | 0.0 | 0.0 | 0.0 | 100.0 | 100.0 | 0.0 | 0.0 | 0.0 | 100.0 | 0.0 | 0.0 | 0.0 | 0.0 | 0.0 | 100.0 | 0.0 |
| Band-Aid | 0.0 | 0.0 | 0.0 | 0.0 | 0.0 | 0.0 | 0.0 | 0.0 | 0.0 | 0.0 | 0.0 | 0.0 | 0.0 | 0.0 | 0.0 | 0.0 | 0.0 | 0.0 |
| Bathroom Accessory | 0.0 | 0.0 | 0.0 | 0.0 | 100.0 | 0.0 | 100.0 | 100.0 | 0.0 | 0.0 | 0.0 | 0.0 | 0.0 | 0.0 | 0.0 | 0.0 | 100.0 | 100.0 |
| Bathroom Cabinet | 0.0 | 0.0 | 0.0 | 0.0 | 0.0 | 0.0 | 0.0 | 0.0 | 0.0 | 0.0 | 100.0 | 0.0 | 0.0 | 0.0 | 0.0 | 0.0 | 0.0 | 0.0 |
| Bathtub | 0.0 | 0.0 | 0.0 | 0.0 | 0.0 | 0.0 | 100.0 | 0.0 | 0.0 | 100.0 | 0.0 | 0.0 | 0.0 | 0.0 | 0.0 | 0.0 | 0.0 | 0.0 |
| Blender | 0.0 | 0.0 | 0.0 | 0.0 | 0.0 | 0.0 | 0.0 | 0.0 | 0.0 | 0.0 | 0.0 | 0.0 | 0.0 | 0.0 | 0.0 | 0.0 | 0.0 | 0.0 |
| Book | 0.0 | 0.0 | 0.0 | 0.0 | 0.0 | 0.0 | 100.0 | 100.0 | 0.0 | 0.0 | 100.0 | 0.0 | 0.0 | 0.0 | 0.0 | 0.0 | 0.0 | 0.0 |
| Bookcase | 0.0 | 0.0 | 0.0 | 0.0 | 0.0 | 0.0 | 0.0 | 0.0 | 0.0 | 0.0 | 0.0 | 0.0 | 0.0 | 0.0 | 0.0 | 0.0 | 0.0 | 0.0 |
| Bottle | 0.0 | 0.0 | 0.0 | 0.0 | 0.0 | 0.0 | 0.0 | 0.0 | 0.0 | 100.0 | 0.0 | 0.0 | 0.0 | 0.0 | 0.0 | 0.0 | 0.0 | 0.0 |
| Bowl | 0.0 | 0.0 | 0.0 | 0.0 | 100.0 | 0.0 | 100.0 | 100.0 | 100.0 | 100.0 | 100.0 | 100.0 | 100.0 | 0.0 | 0.0 | 100.0 | 100.0 | 0.0 |
| Box | 0.0 | 0.0 | 0.0 | 0.0 | 0.0 | 0.0 | 100.0 | 100.0 | 0.0 | 0.0 | 100.0 | 0.0 | 100.0 | 0.0 | 0.0 | 0.0 | 0.0 | 0.0 |

| Object Class | Claude 3.5 S. | Claude 3.7 S. (T) | Claude 3.7 S. | Gemini 2.0 F001 | Gemini 2.5 FP | Gemini 2.5 PP | Llama 3.2 11B (1) | Llama 3.2 11B (2) | Llama 3.2 90B VI | Llama 4 Mav. | Llama 4 Sct. | GPT 4.1 Mini | GPT 4.1 | o4-mini-high (T) | Qwen VP | Qwen 2.5 VL | Grok 2 Vis. | Grok V. Beta |
|---|---|---|---|---|---|---|---|---|---|---|---|---|---|---|---|---|---|---|
| Cabinetry | 0.0 | 0.0 | 0.0 | 0.0 | 100.0 | 100.0 | 0.0 | 0.0 | 0.0 | 0.0 | 100.0 | 0.0 | 0.0 | 0.0 | 0.0 | 0.0 | 0.0 | 0.0 |
| Can Opener | 0.0 | 0.0 | 0.0 | 0.0 | 0.0 | 100.0 | 0.0 | 0.0 | 0.0 | 0.0 | 100.0 | 0.0 | 0.0 | 0.0 | 0.0 | 0.0 | 0.0 | 0.0 |
| Cart | 0.0 | 0.0 | 0.0 | 0.0 | 0.0 | 100.0 | 100.0 | 0.0 | 100.0 | 100.0 | 100.0 | 0.0 | 0.0 | 0.0 | 0.0 | 0.0 | 0.0 | 0.0 |
| Chair | 0.0 | 100.0 | 0.0 | 0.0 | 0.0 | 0.0 | 100.0 | 0.0 | 0.0 | 0.0 | 100.0 | 0.0 | 0.0 | 0.0 | 0.0 | 0.0 | 100.0 | 0.0 |
| Chest Of Drawers | 0.0 | 0.0 | 0.0 | 0.0 | 0.0 | 0.0 | 100.0 | 0.0 | 0.0 | 0.0 | 0.0 | 0.0 | 0.0 | 100.0 | 0.0 | 0.0 | 0.0 | 0.0 |
| Closet | 0.0 | 0.0 | 0.0 | 0.0 | 0.0 | 100.0 | 0.0 | 0.0 | 0.0 | 100.0 | 100.0 | 100.0 | 0.0 | 100.0 | 0.0 | 0.0 | 0.0 | 0.0 |
| Clothing | 0.0 | 0.0 | 0.0 | 0.0 | 0.0 | 0.0 | 100.0 | 100.0 | 100.0 | 100.0 | 0.0 | 0.0 | 0.0 | 0.0 | 0.0 | 0.0 | 0.0 | 0.0 |
| Coffeemaker | 0.0 | 0.0 | 0.0 | 0.0 | 0.0 | 0.0 | 0.0 | 0.0 | 0.0 | 0.0 | 0.0 | 0.0 | 0.0 | 0.0 | 0.0 | 0.0 | 0.0 | 0.0 |
| Container | 100.0 | 0.0 | 0.0 | 100.0 | 0.0 | 100.0 | 0.0 | 0.0 | 100.0 | 0.0 | 0.0 | 100.0 | 100.0 | 0.0 | 0.0 | 0.0 | 0.0 | 0.0 |
| Cooking Spray | 0.0 | 0.0 | 0.0 | 0.0 | 0.0 | 0.0 | 0.0 | 0.0 | 0.0 | 0.0 | 0.0 | 0.0 | 0.0 | 0.0 | 0.0 | 0.0 | 0.0 | 0.0 |
| Countertop | 0.0 | 0.0 | 0.0 | 0.0 | 100.0 | 0.0 | 0.0 | 0.0 | 0.0 | 0.0 | 0.0 | 0.0 | 0.0 | 0.0 | 0.0 | 0.0 | 0.0 | 0.0 |
| Cupboard | 0.0 | 0.0 | 0.0 | 0.0 | 0.0 | 0.0 | 0.0 | 0.0 | 0.0 | 100.0 | 100.0 | 100.0 | 0.0 | 0.0 | 0.0 | 0.0 | 0.0 | 0.0 |
| Cutting Board | 0.0 | 0.0 | 100.0 | 0.0 | 0.0 | 0.0 | 0.0 | 0.0 | 0.0 | 0.0 | 0.0 | 0.0 | 0.0 | 0.0 | 0.0 | 0.0 | 0.0 | 0.0 |
| Desk | 0.0 | 0.0 | 0.0 | 0.0 | 0.0 | 100.0 | 100.0 | 0.0 | 100.0 | 100.0 | 100.0 | 100.0 | 100.0 | 0.0 | 0.0 | 100.0 | 0.0 | 100.0 |
| Diaper | 0.0 | 0.0 | 0.0 | 0.0 | 0.0 | 100.0 | 0.0 | 0.0 | 0.0 | 0.0 | 0.0 | 0.0 | 0.0 | 0.0 | 0.0 | 0.0 | 0.0 | 0.0 |
| Dishwasher | 0.0 | 0.0 | 0.0 | 0.0 | 0.0 | 100.0 | 100.0 | 0.0 | 0.0 | 100.0 | 0.0 | 0.0 | 0.0 | 100.0 | 0.0 | 0.0 | 0.0 | 0.0 |
| Door | 100.0 | 100.0 | 100.0 | 0.0 | 0.0 | 100.0 | 100.0 | 0.0 | 100.0 | 100.0 | 100.0 | 0.0 | 100.0 | 0.0 | 0.0 | 100.0 | 0.0 | 0.0 |
| Door Handle | 100.0 | 0.0 | 0.0 | 100.0 | 100.0 | 0.0 | 0.0 | 0.0 | 100.0 | 100.0 | 100.0 | 100.0 | 0.0 | 0.0 | 0.0 | 100.0 | 100.0 | 0.0 |
| Drawer | 100.0 | 0.0 | 0.0 | 0.0 | 0.0 | 0.0 | 100.0 | 0.0 | 100.0 | 0.0 | 100.0 | 0.0 | 0.0 | 0.0 | 0.0 | 0.0 | 0.0 | 0.0 |
| Drill (Tool) | 0.0 | 0.0 | 0.0 | 0.0 | 0.0 | 0.0 | 0.0 | 0.0 | 0.0 | 0.0 | 0.0 | 0.0 | 0.0 | 0.0 | 0.0 | 0.0 | 0.0 | 0.0 |
| Egg (Food) | 100.0 | 0.0 | 100.0 | 100.0 | 100.0 | 100.0 | 100.0 | 100.0 | 100.0 | 0.0 | 0.0 | 100.0 | 0.0 | 0.0 | 0.0 | 100.0 | 100.0 | 0.0 |
| Filing Cabinet | 0.0 | 0.0 | 0.0 | 0.0 | 0.0 | 0.0 | 0.0 | 0.0 | 0.0 | 0.0 | 0.0 | 0.0 | 0.0 | 0.0 | 0.0 | 0.0 | 0.0 | 0.0 |
| Flashlight | 0.0 | 0.0 | 0.0 | 0.0 | 0.0 | 100.0 | 0.0 | 0.0 | 0.0 | 0.0 | 0.0 | 0.0 | 0.0 | 0.0 | 0.0 | 0.0 | 0.0 | 0.0 |
| Flowerpot | 0.0 | 0.0 | 0.0 | 0.0 | 0.0 | 0.0 | 0.0 | 0.0 | 0.0 | 0.0 | 0.0 | 0.0 | 0.0 | 0.0 | 0.0 | 0.0 | 0.0 | 0.0 |
| Food Processor | 0.0 | 0.0 | 0.0 | 0.0 | 0.0 | 0.0 | 100.0 | 100.0 | 100.0 | 0.0 | 0.0 | 0.0 | 0.0 | 0.0 | 0.0 | 0.0 | 0.0 | 0.0 |
| Fork | 0.0 | 0.0 | 0.0 | 0.0 | 0.0 | 0.0 | 0.0 | 0.0 | 0.0 | 0.0 | 0.0 | 0.0 | 0.0 | 0.0 | 0.0 | 0.0 | 0.0 | 0.0 |
| Frying Pan | 0.0 | 0.0 | 0.0 | 0.0 | 0.0 | 100.0 | 0.0 | 0.0 | 0.0 | 100.0 | 100.0 | 0.0 | 0.0 | 0.0 | 0.0 | 0.0 | 0.0 | 0.0 |
| Furniture | 100.0 | 0.0 | 0.0 | 0.0 | 0.0 | 0.0 | 100.0 | 0.0 | 0.0 | 100.0 | 0.0 | 0.0 | 0.0 | 100.0 | 0.0 | 0.0 | 0.0 | 0.0 |
| Gas Stove | 0.0 | 0.0 | 0.0 | 0.0 | 0.0 | 0.0 | 100.0 | 100.0 | 0.0 | 0.0 | 100.0 | 0.0 | 0.0 | 100.0 | 0.0 | 0.0 | 0.0 | 0.0 |
| Glove | 0.0 | 0.0 | 0.0 | 0.0 | 100.0 | 0.0 | 100.0 | 0.0 | 0.0 | 100.0 | 100.0 | 0.0 | 0.0 | 0.0 | 0.0 | 0.0 | 0.0 | 0.0 |
| Grinder | 0.0 | 0.0 | 0.0 | 0.0 | 100.0 | 0.0 | 100.0 | 0.0 | 0.0 | 0.0 | 0.0 | 0.0 | 0.0 | 0.0 | 0.0 | 0.0 | 0.0 | 100.0 |
| Hammer | 100.0 | 0.0 | 0.0 | 0.0 | 0.0 | 0.0 | 0.0 | 0.0 | 0.0 | 0.0 | 0.0 | 0.0 | 0.0 | 0.0 | 0.0 | 0.0 | 0.0 | 0.0 |
| Home Appliance | 0.0 | 100.0 | 0.0 | 0.0 | 0.0 | 0.0 | 100.0 | 0.0 | 0.0 | 100.0 | 0.0 | 0.0 | 0.0 | 0.0 | 0.0 | 0.0 | 0.0 | 0.0 |
| Infant Bed | 0.0 | 0.0 | 0.0 | 0.0 | 0.0 | 0.0 | 100.0 | 0.0 | 0.0 | 0.0 | 100.0 | 0.0 | 0.0 | 0.0 | 0.0 | 0.0 | 0.0 | 0.0 |
| Jug | 0.0 | 0.0 | 0.0 | 0.0 | 0.0 | 0.0 | 100.0 | 0.0 | 0.0 | 100.0 | 0.0 | 0.0 | 0.0 | 0.0 | 0.0 | 100.0 | 0.0 | 0.0 |
| Kettle | 0.0 | 0.0 | 0.0 | 0.0 | 0.0 | 0.0 | 0.0 | 0.0 | 0.0 | 0.0 | 0.0 | 0.0 | 0.0 | 0.0 | 0.0 | 0.0 | 0.0 | 0.0 |
| Kitchen | 0.0 | 0.0 | 0.0 | 0.0 | 0.0 | 0.0 | 0.0 | 0.0 | 0.0 | 0.0 | 0.0 | 0.0 | 0.0 | 100.0 | 0.0 | 0.0 | 0.0 | 0.0 |
| Kitchen Appliance | 0.0 | 0.0 | 0.0 | 0.0 | 0.0 | 0.0 | 0.0 | 0.0 | 0.0 | 0.0 | 0.0 | 0.0 | 0.0 | 0.0 | 0.0 | 0.0 | 0.0 | 0.0 |
| Kitchen Knife | 0.0 | 100.0 | 100.0 | 100.0 | 0.0 | 100.0 | 0.0 | 0.0 | 0.0 | 0.0 | 0.0 | 100.0 | 100.0 | 100.0 | 0.0 | 0.0 | 0.0 | 0.0 |
| Kitchen Utensil | 0.0 | 100.0 | 0.0 | 0.0 | 0.0 | 0.0 | 100.0 | 0.0 | 0.0 | 0.0 | 0.0 | 0.0 | 0.0 | 100.0 | 0.0 | 0.0 | 0.0 | 100.0 |
| Knife | 100.0 | 0.0 | 100.0 | 0.0 | 0.0 | 0.0 | 100.0 | 0.0 | 0.0 | 0.0 | 0.0 | 100.0 | 0.0 | 0.0 | 0.0 | 0.0 | 0.0 | 0.0 |
| Ladder | 100.0 | 0.0 | 100.0 | 0.0 | 0.0 | 0.0 | 0.0 | 100.0 | 0.0 | 0.0 | 0.0 | 100.0 | 0.0 | 0.0 | 0.0 | 100.0 | 0.0 | 0.0 |
| Ladle | 0.0 | 0.0 | 100.0 | 100.0 | 0.0 | 100.0 | 0.0 | 0.0 | 0.0 | 0.0 | 100.0 | 0.0 | 0.0 | 0.0 | 0.0 | 0.0 | 0.0 | 0.0 |
| Laptop | 0.0 | 0.0 | 0.0 | 0.0 | 0.0 | 0.0 | 100.0 | 0.0 | 0.0 | 0.0 | 0.0 | 0.0 | 0.0 | 0.0 | 0.0 | 0.0 | 0.0 | 0.0 |
| Lavender (Plant) | 0.0 | 0.0 | 0.0 | 0.0 | 0.0 | 0.0 | 0.0 | 0.0 | 0.0 | 0.0 | 100.0 | 0.0 | 0.0 | 100.0 | 0.0 | 0.0 | 0.0 | 0.0 |
| Light Bulb | 0.0 | 0.0 | 0.0 | 0.0 | 0.0 | 0.0 | 0.0 | 0.0 | 100.0 | 100.0 | 100.0 | 0.0 | 0.0 | 0.0 | 0.0 | 0.0 | 0.0 | 0.0 |
| Light Switch | 0.0 | 0.0 | 0.0 | 0.0 | 0.0 | 100.0 | 100.0 | 0.0 | 0.0 | 0.0 | 100.0 | 0.0 | 0.0 | 100.0 | 0.0 | 0.0 | 100.0 | 100.0 |
| Measuring Cup | 0.0 | 0.0 | 0.0 | 0.0 | 0.0 | 0.0 | 0.0 | 0.0 | 0.0 | 0.0 | 0.0 | 0.0 | 0.0 | 0.0 | 0.0 | 0.0 | 0.0 | 0.0 |
| Microwave Oven | 0.0 | 0.0 | 0.0 | 0.0 | 100.0 | 0.0 | 0.0 | 100.0 | 0.0 | 0.0 | 0.0 | 0.0 | 0.0 | 0.0 | 0.0 | 0.0 | 0.0 | 100.0 |
| Milk | 100.0 | 0.0 | 0.0 | 0.0 | 100.0 | 0.0 | 100.0 | 100.0 | 100.0 | 100.0 | 100.0 | 100.0 | 0.0 | 0.0 | 0.0 | 100.0 | 100.0 | 0.0 |
| Mirror | 0.0 | 0.0 | 0.0 | 0.0 | 0.0 | 100.0 | 100.0 | 0.0 | 100.0 | 0.0 | 0.0 | 100.0 | 100.0 | 0.0 | 0.0 | 0.0 | 0.0 | 0.0 |
| Mixer | 0.0 | 0.0 | 0.0 | 0.0 | 0.0 | 100.0 | 0.0 | 0.0 | 0.0 | 0.0 | 0.0 | 0.0 | 0.0 | 0.0 | 0.0 | 0.0 | 0.0 | 0.0 |
| Mixing Bowl | 100.0 | 0.0 | 100.0 | 0.0 | 0.0 | 0.0 | 100.0 | 100.0 | 0.0 | 0.0 | 100.0 | 100.0 | 0.0 | 100.0 | 0.0 | 0.0 | 100.0 | 0.0 |
| Mobile Phone | 0.0 | 0.0 | 0.0 | 0.0 | 0.0 | 0.0 | 0.0 | 0.0 | 0.0 | 100.0 | 0.0 | 0.0 | 0.0 | 0.0 | 0.0 | 0.0 | 0.0 | 0.0 |
| Mug | 0.0 | 0.0 | 0.0 | 0.0 | 0.0 | 0.0 | 0.0 | 0.0 | 100.0 | 0.0 | 0.0 | 0.0 | 0.0 | 0.0 | 0.0 | 0.0 | 0.0 | 0.0 |
| Organ (Musical Instrument) | 0.0 | 0.0 | 100.0 | 0.0 | 0.0 | 0.0 | 0.0 | 0.0 | 0.0 | 0.0 | 100.0 | 0.0 | 0.0 | 0.0 | 0.0 | 0.0 | 100.0 | 0.0 |
| Oven | 0.0 | 0.0 | 0.0 | 0.0 | 0.0 | 0.0 | 0.0 | 0.0 | 0.0 | 0.0 | 0.0 | 0.0 | 0.0 | 0.0 | 0.0 | 0.0 | 0.0 | 0.0 |
| Paper Towel | 0.0 | 0.0 | 0.0 | 0.0 | 0.0 | 0.0 | 100.0 | 0.0 | 0.0 | 0.0 | 0.0 | 0.0 | 0.0 | 0.0 | 0.0 | 0.0 | 0.0 | 0.0 |
| Pen | 0.0 | 0.0 | 0.0 | 0.0 | 100.0 | 0.0 | 100.0 | 0.0 | 0.0 | 0.0 | 0.0 | 0.0 | 0.0 | 0.0 | 0.0 | 0.0 | 100.0 | 0.0 |
| Pitcher (Container) | 0.0 | 0.0 | 0.0 | 0.0 | 0.0 | 0.0 | 0.0 | 100.0 | 100.0 | 100.0 | 100.0 | 100.0 | 0.0 | 0.0 | 0.0 | 100.0 | 0.0 | 0.0 |
| Plant | 100.0 | 0.0 | 0.0 | 0.0 | 0.0 | 0.0 | 0.0 | 0.0 | 0.0 | 0.0 | 0.0 | 0.0 | 0.0 | 0.0 | 0.0 | 0.0 | 0.0 | 0.0 |
| Plastic Bag | 0.0 | 0.0 | 0.0 | 0.0 | 0.0 | 0.0 | 0.0 | 0.0 | 0.0 | 0.0 | 0.0 | 0.0 | 0.0 | 0.0 | 0.0 | 0.0 | 0.0 | 0.0 |
| Plate | 0.0 | 0.0 | 0.0 | 0.0 | 100.0 | 0.0 | 0.0 | 0.0 | 0.0 | 100.0 | 100.0 | 100.0 | 0.0 | 0.0 | 0.0 | 0.0 | 0.0 | 0.0 |
| Plumbing Fixture | 0.0 | 0.0 | 0.0 | 0.0 | 100.0 | 100.0 | 100.0 | 0.0 | 100.0 | 100.0 | 100.0 | 100.0 | 100.0 | 0.0 | 0.0 | 100.0 | 0.0 | 0.0 |
| Power Plugs And Sockets | 0.0 | 0.0 | 0.0 | 0.0 | 0.0 | 0.0 | 100.0 | 0.0 | 0.0 | 0.0 | 100.0 | 0.0 | 0.0 | 0.0 | 0.0 | 0.0 | 0.0 | 0.0 |
| Pressure Cooker | 0.0 | 0.0 | 0.0 | 0.0 | 0.0 | 0.0 | 0.0 | 0.0 | 0.0 | 0.0 | 0.0 | 100.0 | 0.0 | 0.0 | 0.0 | 0.0 | 0.0 | 0.0 |
| Refrigerator | 0.0 | 0.0 | 0.0 | 0.0 | 0.0 | 0.0 | 100.0 | 100.0 | 100.0 | 100.0 | 100.0 | 0.0 | 100.0 | 0.0 | 0.0 | 0.0 | 0.0 | 0.0 |
| Remote Control | 0.0 | 100.0 | 0.0 | 0.0 | 0.0 | 0.0 | 0.0 | 100.0 | 100.0 | 100.0 | 0.0 | 0.0 | 0.0 | 0.0 | 0.0 | 0.0 | 0.0 | 0.0 |
| Scissors | 100.0 | 0.0 | 0.0 | 100.0 | 100.0 | 0.0 | 100.0 | 100.0 | 100.0 | 100.0 | 100.0 | 100.0 | 100.0 | 100.0 | 0.0 | 0.0 | 0.0 | 0.0 |

| Object Class | Claude 3.5 S. | Claude 3.7 S. (T) | Claude 3.7 S. | Gemini 2.0 F001 | Gemini 2.5 FP | Gemini 2.5 PP | Llama 3.2 11B (1) | Llama 3.2 11B (2) | Llama 3.2 90B VI | Llama 4 Mav. | Llama 4 Sct. | GPT 4.1 Mini | GPT 4.1 | o4-mini-high (T) | Qwen VP | Qwen 2.5 VL | Grok 2 Vis. | Grok V. Beta |
|---|---|---|---|---|---|---|---|---|---|---|---|---|---|---|---|---|---|---|
| Screwdriver | 0.0 | 0.0 | 0.0 | 0.0 | 0.0 | 0.0 | 0.0 | 100.0 | 0.0 | 0.0 | 0.0 | 100.0 | 100.0 | 0.0 | 0.0 | 0.0 | 0.0 | 0.0 |
| Serving Tray | 0.0 | 0.0 | 0.0 | 0.0 | 0.0 | 0.0 | 0.0 | 0.0 | 0.0 | 0.0 | 0.0 | 0.0 | 0.0 | 0.0 | 0.0 | 0.0 | 0.0 | 0.0 |
| Shelf | 0.0 | 0.0 | 0.0 | 0.0 | 0.0 | 0.0 | 0.0 | 0.0 | 0.0 | 100.0 | 100.0 | 0.0 | 0.0 | 0.0 | 0.0 | 0.0 | 0.0 | 0.0 |
| Shower | 0.0 | 0.0 | 100.0 | 0.0 | 0.0 | 0.0 | 100.0 | 0.0 | 0.0 | 0.0 | 0.0 | 0.0 | 0.0 | 0.0 | 0.0 | 0.0 | 0.0 | 0.0 |
| Sink | 0.0 | 100.0 | 0.0 | 100.0 | 100.0 | 100.0 | 100.0 | 0.0 | 0.0 | 100.0 | 0.0 | 100.0 | 0.0 | 0.0 | 0.0 | 100.0 | 100.0 | 0.0 |
| Slow Cooker | 0.0 | 0.0 | 0.0 | 0.0 | 0.0 | 0.0 | 0.0 | 0.0 | 0.0 | 0.0 | 0.0 | 0.0 | 0.0 | 0.0 | 0.0 | 0.0 | 0.0 | 0.0 |
| Soap Dispenser | 0.0 | 0.0 | 0.0 | 100.0 | 0.0 | 0.0 | 0.0 | 100.0 | 0.0 | 100.0 | 100.0 | 100.0 | 0.0 | 0.0 | 0.0 | 100.0 | 100.0 | 0.0 |
| Spatula | 100.0 | 100.0 | 100.0 | 0.0 | 0.0 | 0.0 | 100.0 | 0.0 | 0.0 | 100.0 | 100.0 | 100.0 | 100.0 | 100.0 | 0.0 | 0.0 | 100.0 | 0.0 |
| Spice Rack | 0.0 | 0.0 | 0.0 | 0.0 | 0.0 | 0.0 | 100.0 | 0.0 | 0.0 | 0.0 | 0.0 | 0.0 | 0.0 | 0.0 | 0.0 | 100.0 | 0.0 | 0.0 |
| Spoon | 0.0 | 0.0 | 0.0 | 0.0 | 0.0 | 0.0 | 0.0 | 0.0 | 0.0 | 0.0 | 0.0 | 0.0 | 0.0 | 0.0 | 0.0 | 0.0 | 0.0 | 0.0 |
| Stairs | 0.0 | 0.0 | 0.0 | 100.0 | 0.0 | 0.0 | 100.0 | 0.0 | 0.0 | 100.0 | 0.0 | 100.0 | 100.0 | 100.0 | 0.0 | 100.0 | 0.0 | 100.0 |
| Stool | 0.0 | 0.0 | 0.0 | 0.0 | 100.0 | 0.0 | 0.0 | 0.0 | 0.0 | 0.0 | 0.0 | 0.0 | 0.0 | 0.0 | 0.0 | 0.0 | 0.0 | 100.0 |
| Table | 0.0 | 0.0 | 0.0 | 0.0 | 0.0 | 0.0 | 0.0 | 0.0 | 100.0 | 0.0 | 100.0 | 0.0 | 0.0 | 100.0 | 0.0 | 0.0 | 0.0 | 0.0 |
| Tablet Computer | 0.0 | 0.0 | 100.0 | 0.0 | 0.0 | 0.0 | 0.0 | 100.0 | 0.0 | 0.0 | 0.0 | 0.0 | 0.0 | 0.0 | 0.0 | 0.0 | 0.0 | 0.0 |
| Tableware | 0.0 | 100.0 | 0.0 | 0.0 | 100.0 | 100.0 | 100.0 | 0.0 | 0.0 | 100.0 | 100.0 | 0.0 | 0.0 | 0.0 | 0.0 | 0.0 | 0.0 | 0.0 |
| Tap | 0.0 | 0.0 | 0.0 | 0.0 | 0.0 | 0.0 | 0.0 | 0.0 | 0.0 | 0.0 | 0.0 | 0.0 | 0.0 | 100.0 | 0.0 | 0.0 | 0.0 | 0.0 |
| Toaster | 0.0 | 0.0 | 0.0 | 0.0 | 0.0 | 0.0 | 0.0 | 0.0 | 0.0 | 0.0 | 0.0 | 0.0 | 0.0 | 0.0 | 0.0 | 0.0 | 0.0 | 0.0 |
| Toilet | 0.0 | 100.0 | 0.0 | 100.0 | 0.0 | 0.0 | 100.0 | 0.0 | 0.0 | 100.0 | 100.0 | 0.0 | 0.0 | 100.0 | 0.0 | 0.0 | 0.0 | 0.0 |
| Toilet Paper | 0.0 | 0.0 | 0.0 | 0.0 | 0.0 | 100.0 | 0.0 | 100.0 | 0.0 | 0.0 | 100.0 | 0.0 | 0.0 | 0.0 | 0.0 | 0.0 | 100.0 | 0.0 |
| Tool | 0.0 | 0.0 | 100.0 | 0.0 | 0.0 | 0.0 | 100.0 | 0.0 | 0.0 | 0.0 | 100.0 | 0.0 | 0.0 | 0.0 | 0.0 | 0.0 | 0.0 | 100.0 |
| Toothbrush | 0.0 | 0.0 | 0.0 | 0.0 | 0.0 | 0.0 | 0.0 | 100.0 | 0.0 | 100.0 | 0.0 | 0.0 | 0.0 | 0.0 | 0.0 | 0.0 | 0.0 | 0.0 |
| Torch | 0.0 | 0.0 | 0.0 | 0.0 | 100.0 | 0.0 | 0.0 | 0.0 | 0.0 | 0.0 | 0.0 | 0.0 | 0.0 | 0.0 | 0.0 | 0.0 | 0.0 | 0.0 |
| Towel | 0.0 | 100.0 | 100.0 | 100.0 | 0.0 | 100.0 | 100.0 | 0.0 | 100.0 | 100.0 | 100.0 | 100.0 | 100.0 | 100.0 | 0.0 | 100.0 | 100.0 | 0.0 |
| Toy | 0.0 | 0.0 | 0.0 | 0.0 | 0.0 | 0.0 | 0.0 | 0.0 | 0.0 | 100.0 | 0.0 | 0.0 | 0.0 | 0.0 | 0.0 | 0.0 | 0.0 | 100.0 |
| Waffle Iron | 0.0 | 0.0 | 0.0 | 0.0 | 0.0 | 0.0 | 0.0 | 0.0 | 0.0 | 0.0 | 0.0 | 0.0 | 0.0 | 0.0 | 0.0 | 0.0 | 0.0 | 0.0 |
| Wardrobe | 0.0 | 0.0 | 0.0 | 0.0 | 0.0 | 100.0 | 0.0 | 0.0 | 100.0 | 0.0 | 0.0 | 0.0 | 0.0 | 0.0 | 0.0 | 0.0 | 0.0 | 0.0 |
| Washing Machine | 0.0 | 0.0 | 0.0 | 0.0 | 0.0 | 0.0 | 0.0 | 0.0 | 0.0 | 100.0 | 0.0 | 0.0 | 0.0 | 100.0 | 0.0 | 0.0 | 100.0 | 0.0 |
| Waste Container | 0.0 | 0.0 | 0.0 | 0.0 | 0.0 | 100.0 | 100.0 | 100.0 | 100.0 | 0.0 | 0.0 | 100.0 | 0.0 | 0.0 | 0.0 | 0.0 | 0.0 | 0.0 |
| Whisk | 0.0 | 0.0 | 0.0 | 0.0 | 0.0 | 100.0 | 0.0 | 100.0 | 0.0 | 100.0 | 100.0 | 100.0 | 100.0 | 100.0 | 0.0 | 0.0 | 0.0 | 0.0 |
| Window Blind | 0.0 | 0.0 | 0.0 | 0.0 | 0.0 | 0.0 | 0.0 | 0.0 | 0.0 | 100.0 | 0.0 | 0.0 | 0.0 | 0.0 | 0.0 | 0.0 | 0.0 | 0.0 |
| Wok | 0.0 | 0.0 | 0.0 | 100.0 | 0.0 | 100.0 | 0.0 | 0.0 | 0.0 | 0.0 | 100.0 | 100.0 | 0.0 | 0.0 | 0.0 | 0.0 | 0.0 | 0.0 |
| Wood-Burning Stove | 0.0 | 0.0 | 0.0 | 0.0 | 0.0 | 100.0 | 0.0 | 0.0 | 0.0 | 0.0 | 0.0 | 0.0 | 0.0 | 0.0 | 0.0 | 0.0 | 0.0 | 0.0 |
| Wrench | 0.0 | 0.0 | 0.0 | 0.0 | 0.0 | 100.0 | 100.0 | 0.0 | 100.0 | 0.0 | 100.0 | 100.0 | 100.0 | 0.0 | 0.0 | 0.0 | 0.0 | 100.0 |
| Zucchini | 0.0 | 0.0 | 100.0 | 100.0 | 0.0 | 100.0 | 100.0 | 0.0 | 0.0 | 100.0 | 100.0 | 100.0 | 0.0 | 100.0 | 0.0 | 100.0 | 100.0 | 0.0 |

Following Table 14 shows Accuracy (%) of VLMs on recognizing **all correct affordances** for objects using **Single-Category Mapping** in PAC Bench.

| Object Class | Claude 3.5 S. | Claude 3.7 S. (T) | Claude 3.7 S. | Gemini 2.0 F001 | Gemini 2.5 FP | Gemini 2.5 PP | Llama 3.2 11B (1) | Llama 3.2 11B (2) | Llama 3.2 90B VI | Llama 4 Mav. | Llama 4 Sct. | GPT 4.1 Mini | GPT 4.1 | o4-mini-high (T) | Qwen VP | Qwen 2.5 VL | Grok 2 Vis. | Grok V. Beta |
|---|---|---|---|---|---|---|---|---|---|---|---|---|---|---|---|---|---|---|
| Adhesive Tape | 0.0 | 0.0 | 0.0 | 0.0 | 0.0 | 0.0 | 0.0 | 0.0 | 0.0 | 0.0 | 0.0 | 0.0 | 0.0 | 0.0 | 0.0 | 0.0 | 0.0 | 0.0 |
| Backpack | 0.0 | 0.0 | 0.0 | 0.0 | 0.0 | 0.0 | 0.0 | 0.0 | 0.0 | 0.0 | 0.0 | 0.0 | 0.0 | 0.0 | 0.0 | 0.0 | 0.0 | 0.0 |
| Band-Aid | 0.0 | 0.0 | 0.0 | 0.0 | 0.0 | 0.0 | 0.0 | 0.0 | 0.0 | 0.0 | 0.0 | 0.0 | 0.0 | 0.0 | 0.0 | 0.0 | 0.0 | 0.0 |
| Bathroom Accessory | 0.0 | 0.0 | 0.0 | 0.0 | 0.0 | 0.0 | 0.0 | 0.0 | 0.0 | 0.0 | 0.0 | 0.0 | 0.0 | 0.0 | 0.0 | 0.0 | 0.0 | 0.0 |
| Bathroom Cabinet | 0.0 | 0.0 | 0.0 | 0.0 | 0.0 | 0.0 | 0.0 | 0.0 | 0.0 | 0.0 | 0.0 | 0.0 | 0.0 | 0.0 | 0.0 | 0.0 | 0.0 | 0.0 |
| Bathtub | 0.0 | 0.0 | 0.0 | 0.0 | 0.0 | 0.0 | 0.0 | 0.0 | 0.0 | 0.0 | 0.0 | 0.0 | 0.0 | 0.0 | 0.0 | 0.0 | 0.0 | 0.0 |
| Blender | 0.0 | 0.0 | 0.0 | 0.0 | 0.0 | 0.0 | 0.0 | 0.0 | 0.0 | 0.0 | 0.0 | 0.0 | 0.0 | 0.0 | 0.0 | 0.0 | 0.0 | 0.0 |
| Book | 0.0 | 0.0 | 0.0 | 0.0 | 0.0 | 0.0 | 0.0 | 0.0 | 0.0 | 0.0 | 0.0 | 0.0 | 0.0 | 0.0 | 0.0 | 0.0 | 0.0 | 0.0 |
| Bookcase | 0.0 | 0.0 | 0.0 | 0.0 | 0.0 | 0.0 | 0.0 | 0.0 | 0.0 | 0.0 | 0.0 | 0.0 | 0.0 | 0.0 | 0.0 | 0.0 | 0.0 | 0.0 |
| Bottle | 0.0 | 0.0 | 0.0 | 0.0 | 0.0 | 0.0 | 0.0 | 0.0 | 0.0 | 0.0 | 0.0 | 0.0 | 0.0 | 0.0 | 0.0 | 0.0 | 0.0 | 0.0 |
| Bowl | 0.0 | 0.0 | 0.0 | 0.0 | 0.0 | 0.0 | 0.0 | 0.0 | 0.0 | 0.0 | 100.0 | 0.0 | 0.0 | 0.0 | 0.0 | 0.0 | 0.0 | 0.0 |
| Box | 0.0 | 0.0 | 0.0 | 0.0 | 0.0 | 0.0 | 0.0 | 0.0 | 0.0 | 0.0 | 0.0 | 0.0 | 0.0 | 0.0 | 0.0 | 0.0 | 0.0 | 0.0 |
| Cabinetry | 0.0 | 0.0 | 0.0 | 0.0 | 0.0 | 100.0 | 0.0 | 0.0 | 0.0 | 0.0 | 0.0 | 0.0 | 0.0 | 0.0 | 0.0 | 0.0 | 0.0 | 0.0 |
| Can Opener | 0.0 | 0.0 | 0.0 | 0.0 | 0.0 | 0.0 | 0.0 | 0.0 | 0.0 | 0.0 | 0.0 | 0.0 | 0.0 | 0.0 | 0.0 | 0.0 | 0.0 | 0.0 |
| Cart | 0.0 | 0.0 | 0.0 | 0.0 | 0.0 | 0.0 | 0.0 | 0.0 | 0.0 | 0.0 | 0.0 | 0.0 | 0.0 | 0.0 | 0.0 | 0.0 | 0.0 | 0.0 |
| Chair | 0.0 | 0.0 | 0.0 | 0.0 | 0.0 | 0.0 | 0.0 | 0.0 | 0.0 | 0.0 | 0.0 | 0.0 | 0.0 | 0.0 | 0.0 | 0.0 | 0.0 | 0.0 |
| Chest Of Drawers | 0.0 | 0.0 | 0.0 | 0.0 | 0.0 | 0.0 | 0.0 | 0.0 | 0.0 | 0.0 | 0.0 | 0.0 | 0.0 | 0.0 | 0.0 | 0.0 | 0.0 | 0.0 |

| Object Class | Claude 3.5 S. | Claude 3.7 S. (T) | Claude 3.7 S. | Gemini 2.0 F001 | Gemini 2.5 FP | Gemini 2.5 PP | Llama 3.2 11B (1) | Llama 3.2 11B (2) | Llama 3.2 90B VI | Llama 4 Mav. | Llama 4 Sct. | GPT 4.1 Mini | GPT 4.1 | o4-mini-high (T) | Qwen VP | Qwen 2.5 VL | Grok 2 Vis. | Grok V. Beta |
|---|---|---|---|---|---|---|---|---|---|---|---|---|---|---|---|---|---|---|
| Closet | 0.0 | 0.0 | 0.0 | 0.0 | 0.0 | 0.0 | 0.0 | 0.0 | 0.0 | 0.0 | 0.0 | 0.0 | 0.0 | 0.0 | 0.0 | 0.0 | 0.0 | 0.0 |
| Clothing | 0.0 | 0.0 | 0.0 | 0.0 | 0.0 | 0.0 | 0.0 | 0.0 | 0.0 | 0.0 | 0.0 | 0.0 | 0.0 | 0.0 | 0.0 | 0.0 | 0.0 | 0.0 |
| Coffeemaker | 0.0 | 0.0 | 0.0 | 0.0 | 0.0 | 0.0 | 0.0 | 0.0 | 0.0 | 0.0 | 0.0 | 0.0 | 0.0 | 0.0 | 0.0 | 0.0 | 0.0 | 0.0 |
| Container | 0.0 | 0.0 | 0.0 | 0.0 | 0.0 | 0.0 | 0.0 | 0.0 | 0.0 | 0.0 | 0.0 | 0.0 | 0.0 | 0.0 | 0.0 | 0.0 | 0.0 | 0.0 |
| Cooking Spray | 0.0 | 0.0 | 0.0 | 0.0 | 0.0 | 0.0 | 0.0 | 0.0 | 0.0 | 0.0 | 0.0 | 0.0 | 0.0 | 0.0 | 0.0 | 0.0 | 0.0 | 0.0 |
| Countertop | 0.0 | 0.0 | 0.0 | 0.0 | 0.0 | 0.0 | 0.0 | 0.0 | 0.0 | 0.0 | 0.0 | 0.0 | 0.0 | 0.0 | 0.0 | 0.0 | 0.0 | 0.0 |
| Cupboard | 0.0 | 0.0 | 0.0 | 0.0 | 0.0 | 0.0 | 0.0 | 0.0 | 0.0 | 0.0 | 0.0 | 0.0 | 0.0 | 0.0 | 0.0 | 0.0 | 0.0 | 0.0 |
| Cutting Board | 0.0 | 0.0 | 0.0 | 0.0 | 0.0 | 0.0 | 0.0 | 0.0 | 0.0 | 0.0 | 0.0 | 0.0 | 0.0 | 0.0 | 0.0 | 0.0 | 0.0 | 0.0 |
| Desk | 0.0 | 0.0 | 0.0 | 0.0 | 0.0 | 0.0 | 0.0 | 0.0 | 0.0 | 0.0 | 0.0 | 0.0 | 0.0 | 0.0 | 0.0 | 0.0 | 0.0 | 0.0 |
| Diaper | 0.0 | 0.0 | 0.0 | 0.0 | 0.0 | 0.0 | 0.0 | 0.0 | 0.0 | 0.0 | 0.0 | 0.0 | 0.0 | 0.0 | 0.0 | 0.0 | 0.0 | 0.0 |
| Dishwasher | 0.0 | 0.0 | 0.0 | 0.0 | 0.0 | 0.0 | 0.0 | 0.0 | 0.0 | 0.0 | 0.0 | 0.0 | 0.0 | 0.0 | 0.0 | 0.0 | 0.0 | 0.0 |
| Door | 0.0 | 0.0 | 0.0 | 0.0 | 0.0 | 0.0 | 0.0 | 0.0 | 0.0 | 0.0 | 0.0 | 0.0 | 0.0 | 0.0 | 0.0 | 0.0 | 0.0 | 0.0 |
| Door Handle | 0.0 | 0.0 | 0.0 | 0.0 | 0.0 | 0.0 | 0.0 | 0.0 | 0.0 | 0.0 | 0.0 | 0.0 | 0.0 | 0.0 | 0.0 | 0.0 | 0.0 | 0.0 |
| Drawer | 0.0 | 0.0 | 0.0 | 0.0 | 0.0 | 0.0 | 0.0 | 0.0 | 0.0 | 0.0 | 0.0 | 0.0 | 0.0 | 0.0 | 0.0 | 0.0 | 0.0 | 0.0 |
| Drill (Tool) | 0.0 | 0.0 | 0.0 | 0.0 | 0.0 | 0.0 | 0.0 | 0.0 | 0.0 | 0.0 | 0.0 | 0.0 | 0.0 | 0.0 | 0.0 | 0.0 | 0.0 | 0.0 |
| Egg (Food) | 0.0 | 0.0 | 0.0 | 0.0 | 0.0 | 0.0 | 0.0 | 0.0 | 0.0 | 0.0 | 0.0 | 0.0 | 0.0 | 0.0 | 0.0 | 0.0 | 0.0 | 0.0 |
| Filing Cabinet | 0.0 | 0.0 | 0.0 | 0.0 | 0.0 | 0.0 | 0.0 | 0.0 | 0.0 | 0.0 | 0.0 | 0.0 | 0.0 | 0.0 | 0.0 | 0.0 | 0.0 | 0.0 |
| Flashlight | 0.0 | 0.0 | 0.0 | 0.0 | 0.0 | 0.0 | 0.0 | 0.0 | 0.0 | 0.0 | 0.0 | 0.0 | 0.0 | 0.0 | 0.0 | 0.0 | 0.0 | 0.0 |
| Flowerpot | 0.0 | 0.0 | 0.0 | 0.0 | 0.0 | 0.0 | 0.0 | 0.0 | 0.0 | 0.0 | 0.0 | 0.0 | 0.0 | 0.0 | 0.0 | 0.0 | 0.0 | 0.0 |
| Food Processor | 0.0 | 0.0 | 0.0 | 0.0 | 0.0 | 0.0 | 0.0 | 0.0 | 0.0 | 0.0 | 0.0 | 0.0 | 0.0 | 0.0 | 0.0 | 0.0 | 0.0 | 0.0 |
| Fork | 0.0 | 0.0 | 0.0 | 0.0 | 0.0 | 0.0 | 0.0 | 0.0 | 0.0 | 0.0 | 0.0 | 0.0 | 0.0 | 0.0 | 0.0 | 0.0 | 0.0 | 0.0 |
| Frying Pan | 0.0 | 0.0 | 0.0 | 0.0 | 0.0 | 0.0 | 0.0 | 0.0 | 0.0 | 0.0 | 0.0 | 0.0 | 0.0 | 0.0 | 0.0 | 0.0 | 0.0 | 0.0 |
| Furniture | 0.0 | 0.0 | 0.0 | 0.0 | 0.0 | 0.0 | 0.0 | 0.0 | 0.0 | 0.0 | 0.0 | 0.0 | 0.0 | 0.0 | 0.0 | 0.0 | 0.0 | 0.0 |
| Gas Stove | 0.0 | 0.0 | 0.0 | 0.0 | 0.0 | 0.0 | 0.0 | 0.0 | 0.0 | 0.0 | 0.0 | 0.0 | 0.0 | 0.0 | 0.0 | 0.0 | 0.0 | 0.0 |
| Glove | 0.0 | 0.0 | 0.0 | 0.0 | 0.0 | 0.0 | 0.0 | 0.0 | 0.0 | 0.0 | 0.0 | 0.0 | 0.0 | 0.0 | 0.0 | 0.0 | 0.0 | 0.0 |
| Grinder | 0.0 | 0.0 | 0.0 | 0.0 | 0.0 | 0.0 | 0.0 | 0.0 | 0.0 | 0.0 | 0.0 | 0.0 | 0.0 | 0.0 | 0.0 | 0.0 | 0.0 | 0.0 |
| Hammer | 0.0 | 0.0 | 0.0 | 0.0 | 0.0 | 0.0 | 0.0 | 0.0 | 0.0 | 0.0 | 0.0 | 0.0 | 0.0 | 0.0 | 0.0 | 0.0 | 0.0 | 0.0 |
| Home Appliance | 0.0 | 0.0 | 0.0 | 0.0 | 0.0 | 0.0 | 0.0 | 0.0 | 0.0 | 0.0 | 0.0 | 0.0 | 0.0 | 0.0 | 0.0 | 0.0 | 0.0 | 0.0 |
| Infant Bed | 0.0 | 0.0 | 0.0 | 0.0 | 0.0 | 0.0 | 0.0 | 0.0 | 0.0 | 0.0 | 0.0 | 0.0 | 0.0 | 0.0 | 0.0 | 0.0 | 0.0 | 0.0 |
| Jug | 0.0 | 0.0 | 0.0 | 0.0 | 0.0 | 0.0 | 0.0 | 0.0 | 0.0 | 0.0 | 0.0 | 0.0 | 0.0 | 0.0 | 0.0 | 0.0 | 0.0 | 0.0 |
| Kettle | 0.0 | 0.0 | 0.0 | 0.0 | 0.0 | 0.0 | 0.0 | 0.0 | 0.0 | 0.0 | 0.0 | 0.0 | 0.0 | 0.0 | 0.0 | 0.0 | 0.0 | 0.0 |
| Kitchen | 0.0 | 0.0 | 0.0 | 0.0 | 0.0 | 0.0 | 0.0 | 0.0 | 0.0 | 0.0 | 0.0 | 0.0 | 0.0 | 0.0 | 0.0 | 0.0 | 0.0 | 0.0 |
| Kitchen Appliance | 0.0 | 0.0 | 0.0 | 0.0 | 0.0 | 0.0 | 0.0 | 0.0 | 0.0 | 0.0 | 0.0 | 0.0 | 0.0 | 0.0 | 0.0 | 0.0 | 0.0 | 0.0 |
| Kitchen Knife | 0.0 | 0.0 | 0.0 | 0.0 | 0.0 | 0.0 | 0.0 | 0.0 | 0.0 | 0.0 | 0.0 | 0.0 | 0.0 | 0.0 | 0.0 | 0.0 | 0.0 | 0.0 |
| Kitchen Utensil | 0.0 | 0.0 | 0.0 | 0.0 | 0.0 | 0.0 | 0.0 | 0.0 | 0.0 | 0.0 | 0.0 | 0.0 | 0.0 | 0.0 | 0.0 | 0.0 | 0.0 | 0.0 |
| Knife | 0.0 | 0.0 | 0.0 | 0.0 | 0.0 | 0.0 | 0.0 | 0.0 | 0.0 | 0.0 | 0.0 | 0.0 | 0.0 | 0.0 | 0.0 | 0.0 | 0.0 | 0.0 |
| Ladder | 0.0 | 0.0 | 0.0 | 0.0 | 0.0 | 0.0 | 0.0 | 0.0 | 0.0 | 0.0 | 0.0 | 0.0 | 0.0 | 0.0 | 0.0 | 0.0 | 0.0 | 0.0 |
| Ladle | 0.0 | 0.0 | 0.0 | 0.0 | 0.0 | 0.0 | 0.0 | 0.0 | 0.0 | 0.0 | 0.0 | 0.0 | 0.0 | 0.0 | 0.0 | 0.0 | 0.0 | 0.0 |
| Laptop | 0.0 | 0.0 | 0.0 | 0.0 | 0.0 | 0.0 | 0.0 | 0.0 | 0.0 | 0.0 | 0.0 | 0.0 | 0.0 | 0.0 | 0.0 | 0.0 | 0.0 | 0.0 |
| Lavender (Plant) | 0.0 | 0.0 | 0.0 | 0.0 | 0.0 | 0.0 | 0.0 | 0.0 | 0.0 | 0.0 | 0.0 | 0.0 | 0.0 | 0.0 | 0.0 | 0.0 | 0.0 | 0.0 |
| Light Bulb | 0.0 | 0.0 | 0.0 | 0.0 | 0.0 | 0.0 | 0.0 | 0.0 | 0.0 | 0.0 | 0.0 | 0.0 | 0.0 | 0.0 | 0.0 | 0.0 | 0.0 | 0.0 |
| Light Switch | 0.0 | 0.0 | 0.0 | 0.0 | 0.0 | 0.0 | 0.0 | 0.0 | 0.0 | 0.0 | 0.0 | 0.0 | 0.0 | 0.0 | 0.0 | 0.0 | 0.0 | 0.0 |
| Measuring Cup | 0.0 | 0.0 | 0.0 | 0.0 | 0.0 | 0.0 | 0.0 | 0.0 | 0.0 | 0.0 | 0.0 | 0.0 | 0.0 | 0.0 | 0.0 | 0.0 | 0.0 | 0.0 |
| Microwave Oven | 0.0 | 0.0 | 0.0 | 0.0 | 0.0 | 0.0 | 0.0 | 0.0 | 0.0 | 0.0 | 0.0 | 0.0 | 0.0 | 0.0 | 0.0 | 0.0 | 0.0 | 0.0 |
| Milk | 0.0 | 0.0 | 0.0 | 0.0 | 0.0 | 0.0 | 0.0 | 0.0 | 0.0 | 0.0 | 0.0 | 0.0 | 0.0 | 0.0 | 0.0 | 0.0 | 0.0 | 0.0 |
| Mirror | 0.0 | 0.0 | 0.0 | 0.0 | 0.0 | 0.0 | 0.0 | 0.0 | 0.0 | 0.0 | 0.0 | 0.0 | 0.0 | 0.0 | 0.0 | 0.0 | 0.0 | 0.0 |
| Mixer | 0.0 | 0.0 | 0.0 | 0.0 | 0.0 | 0.0 | 0.0 | 0.0 | 0.0 | 0.0 | 0.0 | 0.0 | 0.0 | 0.0 | 0.0 | 0.0 | 0.0 | 0.0 |
| Mixing Bowl | 0.0 | 0.0 | 0.0 | 0.0 | 0.0 | 0.0 | 0.0 | 0.0 | 0.0 | 0.0 | 0.0 | 0.0 | 0.0 | 0.0 | 0.0 | 0.0 | 0.0 | 0.0 |
| Mobile Phone | 0.0 | 0.0 | 0.0 | 0.0 | 0.0 | 0.0 | 0.0 | 0.0 | 0.0 | 0.0 | 0.0 | 0.0 | 0.0 | 0.0 | 0.0 | 0.0 | 0.0 | 0.0 |
| Mug | 0.0 | 0.0 | 0.0 | 0.0 | 0.0 | 0.0 | 0.0 | 0.0 | 0.0 | 0.0 | 0.0 | 0.0 | 0.0 | 0.0 | 0.0 | 0.0 | 0.0 | 0.0 |
| Organ (Musical Instrument) | 0.0 | 0.0 | 0.0 | 0.0 | 0.0 | 0.0 | 0.0 | 0.0 | 0.0 | 0.0 | 0.0 | 0.0 | 0.0 | 0.0 | 0.0 | 0.0 | 0.0 | 0.0 |
| Oven | 0.0 | 0.0 | 0.0 | 0.0 | 0.0 | 0.0 | 0.0 | 0.0 | 0.0 | 0.0 | 0.0 | 0.0 | 0.0 | 0.0 | 0.0 | 0.0 | 0.0 | 0.0 |
| Paper Towel | 0.0 | 0.0 | 0.0 | 0.0 | 0.0 | 0.0 | 0.0 | 0.0 | 0.0 | 0.0 | 0.0 | 0.0 | 0.0 | 0.0 | 0.0 | 0.0 | 0.0 | 0.0 |
| Pen | 0.0 | 0.0 | 0.0 | 0.0 | 0.0 | 0.0 | 0.0 | 0.0 | 0.0 | 0.0 | 0.0 | 0.0 | 0.0 | 0.0 | 0.0 | 0.0 | 0.0 | 0.0 |
| Pitcher (Container) | 0.0 | 0.0 | 0.0 | 0.0 | 0.0 | 0.0 | 0.0 | 0.0 | 0.0 | 0.0 | 0.0 | 0.0 | 0.0 | 0.0 | 0.0 | 0.0 | 0.0 | 0.0 |
| Plant | 0.0 | 0.0 | 0.0 | 0.0 | 0.0 | 0.0 | 0.0 | 0.0 | 0.0 | 0.0 | 0.0 | 0.0 | 0.0 | 0.0 | 0.0 | 0.0 | 0.0 | 0.0 |
| Plastic Bag | 0.0 | 0.0 | 0.0 | 0.0 | 0.0 | 0.0 | 0.0 | 0.0 | 0.0 | 0.0 | 0.0 | 0.0 | 0.0 | 0.0 | 0.0 | 0.0 | 0.0 | 0.0 |
| Plate | 0.0 | 0.0 | 0.0 | 0.0 | 0.0 | 0.0 | 0.0 | 0.0 | 0.0 | 0.0 | 0.0 | 0.0 | 0.0 | 0.0 | 0.0 | 0.0 | 0.0 | 0.0 |
| Plumbing Fixture | 0.0 | 0.0 | 0.0 | 0.0 | 0.0 | 0.0 | 0.0 | 0.0 | 0.0 | 0.0 | 0.0 | 0.0 | 0.0 | 0.0 | 0.0 | 0.0 | 0.0 | 0.0 |
| Power Plugs And Sockets | 0.0 | 0.0 | 0.0 | 0.0 | 0.0 | 0.0 | 0.0 | 0.0 | 0.0 | 0.0 | 0.0 | 0.0 | 0.0 | 0.0 | 0.0 | 0.0 | 0.0 | 0.0 |
| Pressure Cooker | 0.0 | 0.0 | 0.0 | 0.0 | 0.0 | 0.0 | 0.0 | 0.0 | 0.0 | 0.0 | 0.0 | 0.0 | 0.0 | 0.0 | 0.0 | 0.0 | 0.0 | 0.0 |
| Refrigerator | 0.0 | 0.0 | 0.0 | 0.0 | 0.0 | 0.0 | 0.0 | 0.0 | 0.0 | 0.0 | 0.0 | 0.0 | 0.0 | 0.0 | 0.0 | 0.0 | 0.0 | 0.0 |
| Remote Control | 0.0 | 0.0 | 0.0 | 0.0 | 0.0 | 0.0 | 0.0 | 0.0 | 0.0 | 0.0 | 0.0 | 0.0 | 0.0 | 0.0 | 0.0 | 0.0 | 0.0 | 0.0 |
| Scissors | 0.0 | 0.0 | 0.0 | 0.0 | 0.0 | 0.0 | 0.0 | 0.0 | 0.0 | 0.0 | 0.0 | 0.0 | 0.0 | 0.0 | 0.0 | 100.0 | 0.0 | 0.0 |
| Screwdriver | 0.0 | 0.0 | 0.0 | 0.0 | 0.0 | 0.0 | 0.0 | 0.0 | 0.0 | 0.0 | 0.0 | 0.0 | 0.0 | 0.0 | 0.0 | 0.0 | 0.0 | 0.0 |
| Serving Tray | 0.0 | 0.0 | 0.0 | 0.0 | 0.0 | 0.0 | 0.0 | 0.0 | 0.0 | 0.0 | 0.0 | 0.0 | 0.0 | 0.0 | 0.0 | 0.0 | 0.0 | 0.0 |
| Shelf | 0.0 | 0.0 | 0.0 | 0.0 | 0.0 | 0.0 | 0.0 | 0.0 | 0.0 | 0.0 | 0.0 | 0.0 | 0.0 | 0.0 | 0.0 | 0.0 | 0.0 | 0.0 |
| Shower | 0.0 | 0.0 | 0.0 | 0.0 | 0.0 | 0.0 | 0.0 | 0.0 | 0.0 | 0.0 | 0.0 | 0.0 | 0.0 | 0.0 | 0.0 | 0.0 | 0.0 | 0.0 |
| Sink | 0.0 | 0.0 | 0.0 | 0.0 | 0.0 | 0.0 | 0.0 | 0.0 | 0.0 | 0.0 | 0.0 | 0.0 | 0.0 | 0.0 | 0.0 | 0.0 | 0.0 | 0.0 |

| Object Class | Claude 3.5 S. | Claude 3.7 S. (T) | Claude 3.7 S. | Gemini 2.0 F001 | Gemini 2.5 FP | Gemini 2.5 PP | Llama 3.2 11B (1) | Llama 3.2 11B (2) | Llama 3.2 90B VI | Llama 4 Mav. | Llama 4 Sct. | GPT 4.1 Mini | GPT 4.1 | o4-mini-high (T) | Qwen VP | Qwen 2.5 VL | Grok 2 Vis. | Grok V. Beta |
|---|---|---|---|---|---|---|---|---|---|---|---|---|---|---|---|---|---|---|
| Slow Cooker | 0.0 | 0.0 | 0.0 | 0.0 | 0.0 | 0.0 | 0.0 | 0.0 | 0.0 | 0.0 | 0.0 | 0.0 | 0.0 | 0.0 | 0.0 | 0.0 | 0.0 | 0.0 |
| Soap Dispenser | 0.0 | 0.0 | 0.0 | 0.0 | 0.0 | 0.0 | 0.0 | 0.0 | 0.0 | 0.0 | 0.0 | 0.0 | 0.0 | 0.0 | 0.0 | 0.0 | 0.0 | 0.0 |
| Spatula | 0.0 | 0.0 | 0.0 | 0.0 | 0.0 | 0.0 | 0.0 | 0.0 | 0.0 | 0.0 | 0.0 | 0.0 | 0.0 | 0.0 | 0.0 | 0.0 | 0.0 | 0.0 |
| Spice Rack | 0.0 | 0.0 | 0.0 | 0.0 | 0.0 | 0.0 | 0.0 | 0.0 | 0.0 | 0.0 | 0.0 | 0.0 | 0.0 | 0.0 | 0.0 | 0.0 | 0.0 | 0.0 |
| Spoon | 0.0 | 0.0 | 0.0 | 0.0 | 0.0 | 0.0 | 0.0 | 0.0 | 0.0 | 0.0 | 0.0 | 0.0 | 0.0 | 0.0 | 0.0 | 0.0 | 0.0 | 0.0 |
| Stairs | 0.0 | 0.0 | 0.0 | 0.0 | 0.0 | 0.0 | 0.0 | 0.0 | 0.0 | 0.0 | 0.0 | 0.0 | 0.0 | 100.0 | 0.0 | 0.0 | 0.0 | 0.0 |
| Stool | 0.0 | 0.0 | 0.0 | 0.0 | 0.0 | 0.0 | 0.0 | 0.0 | 0.0 | 0.0 | 0.0 | 0.0 | 0.0 | 0.0 | 0.0 | 0.0 | 0.0 | 0.0 |
| Table | 0.0 | 0.0 | 0.0 | 0.0 | 0.0 | 0.0 | 0.0 | 0.0 | 0.0 | 0.0 | 0.0 | 0.0 | 0.0 | 0.0 | 0.0 | 0.0 | 0.0 | 0.0 |
| Tablet Computer | 0.0 | 0.0 | 0.0 | 0.0 | 0.0 | 0.0 | 0.0 | 0.0 | 0.0 | 0.0 | 0.0 | 0.0 | 0.0 | 0.0 | 0.0 | 0.0 | 0.0 | 0.0 |
| Tableware | 0.0 | 0.0 | 0.0 | 0.0 | 0.0 | 0.0 | 0.0 | 0.0 | 0.0 | 0.0 | 0.0 | 0.0 | 0.0 | 0.0 | 0.0 | 0.0 | 0.0 | 0.0 |
| Tap | 0.0 | 0.0 | 0.0 | 0.0 | 0.0 | 0.0 | 0.0 | 0.0 | 0.0 | 0.0 | 0.0 | 0.0 | 0.0 | 0.0 | 0.0 | 0.0 | 0.0 | 0.0 |
| Toaster | 0.0 | 0.0 | 0.0 | 0.0 | 0.0 | 0.0 | 0.0 | 0.0 | 0.0 | 0.0 | 0.0 | 0.0 | 0.0 | 0.0 | 0.0 | 0.0 | 0.0 | 0.0 |
| Toilet | 0.0 | 0.0 | 0.0 | 0.0 | 0.0 | 0.0 | 0.0 | 0.0 | 0.0 | 0.0 | 0.0 | 0.0 | 0.0 | 0.0 | 0.0 | 0.0 | 0.0 | 0.0 |
| Toilet Paper | 0.0 | 0.0 | 0.0 | 0.0 | 0.0 | 0.0 | 0.0 | 0.0 | 0.0 | 0.0 | 0.0 | 0.0 | 0.0 | 0.0 | 0.0 | 0.0 | 0.0 | 0.0 |
| Tool | 0.0 | 0.0 | 0.0 | 0.0 | 0.0 | 0.0 | 0.0 | 0.0 | 0.0 | 0.0 | 0.0 | 0.0 | 0.0 | 0.0 | 0.0 | 0.0 | 0.0 | 0.0 |
| Toothbrush | 0.0 | 0.0 | 0.0 | 0.0 | 0.0 | 0.0 | 0.0 | 0.0 | 0.0 | 0.0 | 0.0 | 0.0 | 0.0 | 0.0 | 0.0 | 0.0 | 0.0 | 0.0 |
| Torch | 0.0 | 0.0 | 0.0 | 0.0 | 0.0 | 0.0 | 0.0 | 0.0 | 0.0 | 0.0 | 0.0 | 0.0 | 0.0 | 0.0 | 0.0 | 0.0 | 0.0 | 0.0 |
| Towel | 0.0 | 0.0 | 0.0 | 0.0 | 0.0 | 0.0 | 0.0 | 0.0 | 0.0 | 0.0 | 0.0 | 0.0 | 0.0 | 0.0 | 0.0 | 0.0 | 0.0 | 0.0 |
| Toy | 0.0 | 0.0 | 0.0 | 0.0 | 0.0 | 0.0 | 0.0 | 0.0 | 0.0 | 0.0 | 0.0 | 0.0 | 0.0 | 0.0 | 0.0 | 0.0 | 0.0 | 0.0 |
| Waffle Iron | 0.0 | 0.0 | 0.0 | 0.0 | 0.0 | 0.0 | 0.0 | 0.0 | 0.0 | 0.0 | 0.0 | 0.0 | 0.0 | 0.0 | 0.0 | 0.0 | 0.0 | 0.0 |
| Wardrobe | 0.0 | 0.0 | 0.0 | 0.0 | 0.0 | 0.0 | 0.0 | 0.0 | 0.0 | 0.0 | 0.0 | 0.0 | 0.0 | 0.0 | 0.0 | 0.0 | 0.0 | 0.0 |
| Washing Machine | 0.0 | 0.0 | 0.0 | 0.0 | 0.0 | 0.0 | 0.0 | 0.0 | 0.0 | 0.0 | 0.0 | 0.0 | 0.0 | 0.0 | 0.0 | 0.0 | 0.0 | 0.0 |
| Waste Container | 0.0 | 0.0 | 0.0 | 0.0 | 0.0 | 0.0 | 0.0 | 0.0 | 0.0 | 0.0 | 0.0 | 0.0 | 0.0 | 0.0 | 0.0 | 0.0 | 0.0 | 0.0 |
| Whisk | 0.0 | 0.0 | 0.0 | 0.0 | 0.0 | 0.0 | 0.0 | 0.0 | 0.0 | 0.0 | 0.0 | 0.0 | 0.0 | 0.0 | 0.0 | 0.0 | 0.0 | 0.0 |
| Window Blind | 0.0 | 0.0 | 0.0 | 0.0 | 0.0 | 0.0 | 0.0 | 0.0 | 0.0 | 0.0 | 0.0 | 0.0 | 0.0 | 0.0 | 0.0 | 0.0 | 0.0 | 0.0 |
| Wok | 0.0 | 0.0 | 0.0 | 0.0 | 0.0 | 0.0 | 0.0 | 0.0 | 0.0 | 0.0 | 0.0 | 0.0 | 0.0 | 0.0 | 0.0 | 0.0 | 0.0 | 0.0 |
| Wood-Burning Stove | 0.0 | 0.0 | 0.0 | 0.0 | 0.0 | 0.0 | 0.0 | 0.0 | 0.0 | 0.0 | 0.0 | 0.0 | 0.0 | 0.0 | 0.0 | 0.0 | 0.0 | 0.0 |
| Wrench | 0.0 | 0.0 | 0.0 | 0.0 | 0.0 | 0.0 | 0.0 | 0.0 | 0.0 | 0.0 | 0.0 | 0.0 | 0.0 | 0.0 | 0.0 | 0.0 | 0.0 | 0.0 |
| Zucchini | 0.0 | 0.0 | 0.0 | 0.0 | 0.0 | 0.0 | 0.0 | 0.0 | 0.0 | 0.0 | 0.0 | 0.0 | 0.0 | 0.0 | 0.0 | 0.0 | 0.0 | 0.0 |

## D.4 Constraint Evaluations

# E Human Survey

This section describes how we gathered and filtered the human–annotated labels that accompany our three image collections: (i) a single–image subset of OpenImages, (ii) the *Real-Robo* dual-view humanoid dataset, and (iii) the *RoboCasa* synthetic renders. Across all datasets we collected categorical judgements for **15 physical-property ontologies** (e.g. *Weight*, *Hardness*, *Capacity*) together with free-form affordances and, where relevant, environment constraints. The same label set, category order, and keyboard shortcuts were used everywhere to ensure a uniform annotation experience (see Figures 11–15).

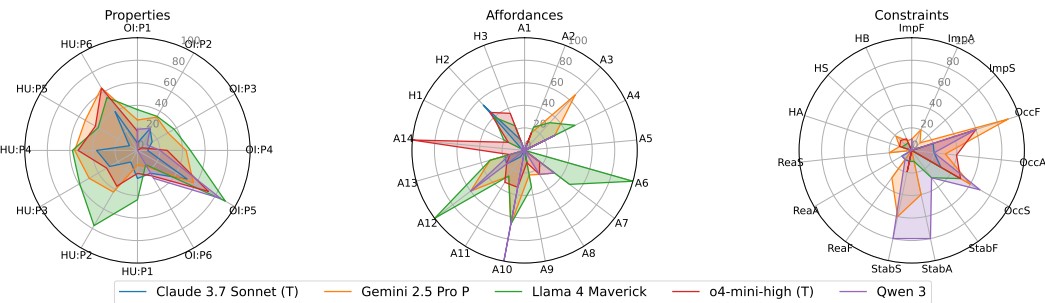

Figure 10: Performance of best models from each family

Table 9: Affordance Accuracy (%) of VLMs on recognizing **at least one correct affordance** for objects grouped by primary categories (Single-Category Mapping)

| Model | A1 | A2 | A3 | A4 | A5 | A6 | A7 | A8 | A9 | A10 | A11 | A12 | A13 | A14 | A15 | A16 | A17 | A18 | H1 | H2 | H3 |
|---|---|---|---|---|---|---|---|---|---|---|---|---|---|---|---|---|---|---|---|---|---|
| Claude 3.5 Sonnet | 0.0 | 0.0 | 0.0 | 0.0 | 0.0 | 0.0 | 16.7 | 25.0 | 0.0 | 66.7 | 13.3 | 40.0 | 9.1 | 0.0 | 0.0 | 0.0 | 44.4 | 0.0 | 2.9 | 47.1 | 14.7 |
| Claude 3.7 Sonnet (T) | 0.0 | 5.6 | 0.0 | 30.0 | 0.0 | 0.0 | 0.0 | 0.0 | 11.1 | 0.0 | 6.7 | 20.0 | 18.2 | 0.0 | 0.0 | 0.0 | 0.0 | 0.0 | 2.9 | 54.4 | 10.3 |
| Claude 3.7 Sonnet | 100.0 | 0.0 | 0.0 | 20.0 | 0.0 | 0.0 | 0.0 | 0.0 | 11.1 | 66.7 | 0.0 | 20.0 | 22.7 | 100.0 | 0.0 | 0.0 | 33.3 | 0.0 | 2.9 | 58.8 | 11.8 |
| Gemini 2.0 Flash 001 | 0.0 | 0.0 | 0.0 | 40.0 | 0.0 | 0.0 | 16.7 | 0.0 | 0.0 | 66.7 | 0.0 | 40.0 | 13.6 | 0.0 | 0.0 | 0.0 | 11.1 | 0.0 | 54.4 | 66.2 | 64.7 |
| Gemini 2.5 Flash P | 0.0 | 5.6 | 0.0 | 20.0 | 0.0 | 50.0 | 0.0 | 0.0 | 11.1 | 66.7 | 13.3 | 40.0 | 18.2 | 0.0 | 50.0 | 0.0 | 22.2 | 0.0 | 52.9 | 55.9 | 57.4 |
| Gemini 2.5 Pro P | 0.0 | 16.7 | 66.7 | 30.0 | 0.0 | 0.0 | 33.3 | 25.0 | 22.2 | 66.7 | 26.7 | 60.0 | 31.8 | 0.0 | 0.0 | 33.3 | 11.1 | 0.0 | 0.0 | 0.0 | 0.0 |
| Llama 3.2 11B Vision I | 100.0 | 22.2 | 0.0 | 30.0 | 0.0 | 50.0 | 33.3 | 0.0 | 22.2 | 66.7 | 0.0 | 0.0 | 13.6 | 0.0 | 50.0 | 33.3 | 33.3 | 0.0 | 20.5 | 27.9 | 25.0 |
| Llama 3.2 90B Vision I | 100.0 | 11.1 | 33.3 | 10.0 | 0.0 | 50.0 | 50.0 | 25.0 | 22.2 | 66.7 | 26.7 | 60.0 | 9.1 | 0.0 | 0.0 | 0.0 | 22.2 | 0.0 | 22.1 | 44.1 | 0.0 |
| Llama 4 Maverick | 0.0 | 22.2 | 33.3 | 50.0 | 0.0 | 100.0 | 50.0 | 0.0 | 33.3 | 66.7 | 26.7 | 60.0 | 31.8 | 0.0 | 0.0 | 33.3 | 11.1 | 100.0 | 20.6 | 39.7 | 23.5 |
| Llama 4 Scout | 0.0 | 11.1 | 66.7 | 50.0 | 0.0 | 50.0 | 50.0 | 25.0 | 33.3 | 66.7 | 53.3 | 60.0 | 54.6 | 100.0 | 50.0 | 0.0 | 33.3 | 0.0 | 20.6 | 27.9 | 26.5 |
| GPT 4.1 Mini | 0.0 | 5.6 | 0.0 | 30.0 | 0.0 | 0.0 | 50.0 | 25.0 | 0.0 | 100.0 | 13.3 | 60.0 | 36.4 | 0.0 | 0.0 | 0.0 | 55.6 | 0.0 | 20.6 | 57.4 | 25.0 |
| GPT 4.1 | 0.0 | 5.6 | 0.0 | 20.0 | 0.0 | 0.0 | 16.7 | 25.0 | 0.0 | 0.0 | 6.7 | 60.0 | 18.2 | 0.0 | 0.0 | 0.0 | 33.3 | 0.0 | 48.5 | 67.6 | 45.6 |
| o4-mini-high (T) | 0.0 | 16.7 | 0.0 | 20.0 | 0.0 | 0.0 | 16.7 | 25.0 | 11.1 | 33.3 | 33.3 | 20.0 | 22.7 | 0.0 | 0.0 | 0.0 | 11.1 | 0.0 | 16.2 | 45.6 | 35.3 |
| Qwen 2.5 VL | 0.0 | 0.0 | 0.0 | 30.0 | 0.0 | 0.0 | 33.3 | 0.0 | 0.0 | 100.0 | 6.7 | 80.0 | 9.1 | 0.0 | 0.0 | 0.0 | 11.1 | 0.0 | 14.7 | 48.5 | 20.6 |
| Qwen 3 | 0.0 | 5.5 | 0.0 | 30.0 | 0.0 | 0.0 | 33.3 | 25.0 | 0.0 | 100.0 | 0.0 | 60.0 | 13.6 | 0.0 | 0.0 | 0.0 | 44.4 | 0.0 | 4.4 | 1.4 | 8.8 |
| Grok 2 Vision | 0.0 | 5.6 | 33.3 | 50.0 | 0.0 | 0.0 | 0.0 | 0.0 | 11.1 | 100.0 | 6.7 | 20.0 | 13.6 | 100.0 | 50.0 | 0.0 | 0.0 | 0.0 | 44.1 | 47.1 | 41.2 |
| Grok 2 Beta | 0.0 | 5.6 | 0.0 | 10.0 | 0.0 | 0.0 | 0.0 | 0.0 | 11.1 | 0.0 | 13.3 | 20.0 | 4.6 | 0.0 | 0.0 | 0.0 | 33.3 | 100.0 | 8.8 | 8.8 | 7.4 |

Table 10: Accuracy (%) of VLMs on recognizing **all correct affordances** for objects grouped by primary categories (Single-Category Mapping) in PAC Bench. Categories C1-C18 are as defined in Table 3

| Model | A1 | A2 | A3 | A4 | A5 | A6 | A7 | A8 | A9 | A10 | A11 | A12 | A13 | A14 | A15 | A16 | A17 | A18 |
|---|---|---|---|---|---|---|---|---|---|---|---|---|---|---|---|---|---|---|
| Claude 3.5 Sonnet | 0.0 | 0.0 | 0.0 | 0.0 | 0.0 | 0.0 | 0.0 | 0.0 | 0.0 | 0.0 | 0.0 | 0.0 | 0.0 | 0.0 | 0.0 | 0.0 | 0.0 | 0.0 |
| Claude 3.7 Sonnet (T) | 0.0 | 0.0 | 0.0 | 0.0 | 0.0 | 0.0 | 0.0 | 0.0 | 0.0 | 0.0 | 0.0 | 0.0 | 0.0 | 0.0 | 0.0 | 0.0 | 0.0 | 0.0 |
| Claude 3.7 Sonnet | 0.0 | 0.0 | 0.0 | 0.0 | 0.0 | 0.0 | 0.0 | 0.0 | 0.0 | 0.0 | 0.0 | 0.0 | 0.0 | 0.0 | 0.0 | 0.0 | 0.0 | 0.0 |
| Gemini 2.0 Flash 001 | 0.0 | 0.0 | 0.0 | 0.0 | 0.0 | 0.0 | 0.0 | 0.0 | 0.0 | 0.0 | 0.0 | 0.0 | 0.0 | 0.0 | 0.0 | 0.0 | 0.0 | 0.0 |
| Gemini 2.5 Flash P | 0.0 | 0.0 | 0.0 | 0.0 | 0.0 | 0.0 | 0.0 | 0.0 | 0.0 | 0.0 | 0.0 | 0.0 | 0.0 | 0.0 | 0.0 | 0.0 | 0.0 | 0.0 |
| Gemini 2.5 Pro P | 0.0 | 0.0 | 0.0 | 0.0 | 0.0 | 0.0 | 0.0 | 0.0 | 0.0 | 0.0 | 6.7 | 0.0 | 0.0 | 0.0 | 0.0 | 0.0 | 0.0 | 0.0 |
| Llama 3.2 11B Vision I | 0.0 | 0.0 | 0.0 | 0.0 | 0.0 | 0.0 | 0.0 | 0.0 | 0.0 | 0.0 | 0.0 | 0.0 | 0.0 | 0.0 | 0.0 | 0.0 | 0.0 | 0.0 |
| Llama 3.2 90B Vision I | 0.0 | 0.0 | 0.0 | 0.0 | 0.0 | 0.0 | 0.0 | 0.0 | 0.0 | 0.0 | 0.0 | 0.0 | 0.0 | 0.0 | 0.0 | 0.0 | 0.0 | 0.0 |
| Llama 4 Maverick | 0.0 | 0.0 | 0.0 | 0.0 | 0.0 | 0.0 | 0.0 | 0.0 | 0.0 | 0.0 | 0.0 | 0.0 | 0.0 | 0.0 | 0.0 | 0.0 | 0.0 | 0.0 |
| Llama 4 Scout | 0.0 | 0.0 | 0.0 | 0.0 | 0.0 | 0.0 | 0.0 | 0.0 | 0.0 | 0.0 | 0.0 | 0.0 | 4.5 | 0.0 | 0.0 | 0.0 | 0.0 | 0.0 |
| GPT 4.1 Mini | 0.0 | 0.0 | 0.0 | 0.0 | 0.0 | 0.0 | 0.0 | 0.0 | 0.0 | 0.0 | 0.0 | 0.0 | 0.0 | 0.0 | 0.0 | 0.0 | 0.0 | 0.0 |
| GPT 4.1 | 0.0 | 0.0 | 0.0 | 0.0 | 0.0 | 0.0 | 0.0 | 0.0 | 0.0 | 0.0 | 0.0 | 20.0 | 0.0 | 0.0 | 0.0 | 0.0 | 0.0 | 0.0 |
| o4-mini-high (T) | 0.0 | 0.0 | 0.0 | 0.0 | 0.0 | 0.0 | 0.0 | 0.0 | 0.0 | 0.0 | 0.0 | 0.0 | 0.0 | 0.0 | 0.0 | 0.0 | 0.0 | 0.0 |
| Qwen VP | 0.0 | 0.0 | 0.0 | 0.0 | 0.0 | 0.0 | 0.0 | 0.0 | 0.0 | 0.0 | 0.0 | 0.0 | 0.0 | 0.0 | 0.0 | 0.0 | 0.0 | 0.0 |
| Qwen 2.5 VL | 0.0 | 0.0 | 0.0 | 0.0 | 0.0 | 0.0 | 0.0 | 0.0 | 0.0 | 0.0 | 0.0 | 0.0 | 0.0 | 0.0 | 0.0 | 0.0 | 11.1 | 0.0 |
| Grok 2 Vision | 0.0 | 0.0 | 0.0 | 0.0 | 0.0 | 0.0 | 0.0 | 0.0 | 0.0 | 0.0 | 0.0 | 0.0 | 0.0 | 0.0 | 0.0 | 0.0 | 0.0 | 0.0 |
| Grok 2 Beta | 0.0 | 0.0 | 0.0 | 0.0 | 0.0 | 0.0 | 0.0 | 0.0 | 0.0 | 0.0 | 0.0 | 0.0 | 0.0 | 0.0 | 0.0 | 0.0 | 0.0 | 0.0 |

Table 11: Accuracy (%) of VLMs on recognizing **at least one correct affordance** for objects using **Multi-Category Mapping** in PAC Bench.

| Model | A1 | A2 | A3 | A4 | A5 | C6 | A7 | A8 | A9 | A10 | A11 | A12 | A13 | A14 | A15 | A16 | A17 | A18 |
|---|---|---|---|---|---|---|---|---|---|---|---|---|---|---|---|---|---|---|
| Claude 3.5 Sonnet | 0.0 | 0.0 | 0.0 | 0.0 | 0.0 | 0.0 | 18.2 | 20.0 | 0.0 | 50.0 | 18.8 | 14.3 | 5.3 | 0.0 | 14.3 | 0.0 | 35.7 | 0.0 |
| Claude 3.7 Sonnet (T) | 0.0 | 5.6 | 0.0 | 27.3 | 0.0 | 0.0 | 0.0 | 0.0 | 9.1 | 0.0 | 12.5 | 14.3 | 10.5 | 0.0 | 0.0 | 12.5 | 7.1 | 0.0 |
| Claude 3.7 Sonnet | 100.0 | 0.0 | 0.0 | 18.2 | 0.0 | 0.0 | 0.0 | 0.0 | 9.1 | 50.0 | 6.2 | 14.3 | 13.2 | 100.0 | 14.3 | 12.5 | 35.7 | 0.0 |
| Gemini 2.0 Flash 001 | 0.0 | 0.0 | 0.0 | 36.4 | 0.0 | 0.0 | 9.1 | 0.0 | 0.0 | 50.0 | 0.0 | 21.4 | 7.9 | 0.0 | 14.3 | 25.0 | 7.1 | 0.0 |
| Gemini 2.5 Flash P | 0.0 | 5.6 | 0.0 | 27.3 | 0.0 | 33.3 | 9.1 | 0.0 | 9.1 | 50.0 | 12.5 | 21.4 | 13.2 | 0.0 | 28.6 | 12.5 | 14.3 | 0.0 |
| Gemini 2.5 Pro P | 0.0 | 16.7 | 66.7 | 36.4 | 0.0 | 33.3 | 36.4 | 20.0 | 27.3 | 50.0 | 31.2 | 57.1 | 26.3 | 0.0 | 14.3 | 37.5 | 28.6 | 0.0 |
| Llama 3.2 11B VI | 100.0 | 22.2 | 0.0 | 27.3 | 0.0 | 33.3 | 18.2 | 20.0 | 18.2 | 50.0 | 0.0 | 7.1 | 18.4 | 0.0 | 42.9 | 50.0 | 28.6 | 0.0 |
| Llama 3.2 90B VI | 100.0 | 11.1 | 33.3 | 18.2 | 0.0 | 33.3 | 45.5 | 20.0 | 18.2 | 50.0 | 31.2 | 42.9 | 10.5 | 0.0 | 28.6 | 25.0 | 14.3 | 0.0 |
| Llama 4 Maverick | 0.0 | 22.2 | 33.3 | 54.5 | 0.0 | 66.7 | 36.4 | 0.0 | 36.4 | 50.0 | 31.2 | 57.1 | 23.7 | 0.0 | 28.6 | 62.5 | 21.4 | 100.0 |
| Llama 4 Scout | 0.0 | 11.1 | 66.7 | 54.5 | 0.0 | 33.3 | 54.5 | 40.0 | 27.3 | 50.0 | 56.2 | 35.7 | 36.8 | 100.0 | 42.9 | 50.0 | 42.9 | 0.0 |
| GPT 4.1 Mini | 0.0 | 5.6 | 0.0 | 36.4 | 0.0 | 0.0 | 27.3 | 20.0 | 0.0 | 75.0 | 12.5 | 42.9 | 23.7 | 0.0 | 28.6 | 25.0 | 50.0 | 0.0 |
| GPT 4.1 | 0.0 | 5.6 | 0.0 | 27.3 | 0.0 | 0.0 | 9.1 | 20.0 | 0.0 | 0.0 | 12.5 | 35.7 | 13.2 | 0.0 | 28.6 | 12.5 | 35.7 | 0.0 |
| o4-mini-high (T) | 0.0 | 16.7 | 0.0 | 18.2 | 0.0 | 0.0 | 9.1 | 20.0 | 18.2 | 25.0 | 31.2 | 21.4 | 18.4 | 0.0 | 14.3 | 12.5 | 21.4 | 0.0 |
| Qwen VP | 0.0 | 0.0 | 0.0 | 0.0 | 0.0 | 0.0 | 0.0 | 0.0 | 0.0 | 0.0 | 0.0 | 0.0 | 0.0 | 0.0 | 0.0 | 0.0 | 0.0 | 0.0 |
| Qwen 2.5 VL | 0.0 | 0.0 | 0.0 | 36.4 | 0.0 | 0.0 | 18.2 | 0.0 | 0.0 | 75.0 | 12.5 | 35.7 | 5.3 | 0.0 | 14.3 | 25.0 | 7.1 | 0.0 |
| Grok 2 Vision | 0.0 | 5.6 | 33.3 | 45.5 | 0.0 | 0.0 | 0.0 | 0.0 | 18.2 | 75.0 | 6.2 | 28.6 | 7.9 | 100.0 | 14.3 | 37.5 | 7.1 | 0.0 |
| Grok Vision Beta | 0.0 | 5.6 | 0.0 | 9.1 | 0.0 | 0.0 | 0.0 | 0.0 | 9.1 | 0.0 | 12.5 | 14.3 | 5.3 | 0.0 | 14.3 | 0.0 | 21.4 | 100.0 |

Table 12: Accuracy (%) of VLMs on recognizing **all correct affordances** for objects using **Multi-Category Mapping** in PAC Bench.

| Model | C1 | C2 | C3 | C4 | C5 | C6 | C7 | C8 | C9 | C10 | C11 | C12 | C13 | C14 | C15 | C16 | C17 | C18 |
|---|---|---|---|---|---|---|---|---|---|---|---|---|---|---|---|---|---|---|
| Claude 3.5 Sonnet | 0.0 | 0.0 | 0.0 | 0.0 | 0.0 | 0.0 | 0.0 | 0.0 | 0.0 | 0.0 | 0.0 | 0.0 | 0.0 | 0.0 | 0.0 | 0.0 | 0.0 | 0.0 |
| Claude 3.7 Sonnet (T) | 0.0 | 0.0 | 0.0 | 0.0 | 0.0 | 0.0 | 0.0 | 0.0 | 0.0 | 0.0 | 0.0 | 0.0 | 0.0 | 0.0 | 0.0 | 0.0 | 0.0 | 0.0 |
| Claude 3.7 Sonnet | 0.0 | 0.0 | 0.0 | 0.0 | 0.0 | 0.0 | 0.0 | 0.0 | 0.0 | 0.0 | 0.0 | 0.0 | 0.0 | 0.0 | 0.0 | 0.0 | 0.0 | 0.0 |
| Gemini 2.0 Flash 001 | 0.0 | 0.0 | 0.0 | 0.0 | 0.0 | 0.0 | 0.0 | 0.0 | 0.0 | 0.0 | 0.0 | 0.0 | 0.0 | 0.0 | 0.0 | 0.0 | 0.0 | 0.0 |
| Gemini 2.5 Flash P | 0.0 | 0.0 | 0.0 | 0.0 | 0.0 | 0.0 | 0.0 | 0.0 | 0.0 | 0.0 | 0.0 | 0.0 | 0.0 | 0.0 | 0.0 | 0.0 | 0.0 | 0.0 |
| Gemini 2.5 Pro P | 0.0 | 0.0 | 0.0 | 0.0 | 0.0 | 0.0 | 9.1 | 0.0 | 0.0 | 0.0 | 6.2 | 0.0 | 0.0 | 0.0 | 0.0 | 0.0 | 0.0 | 0.0 |
| Llama 3.2 11B VI | 0.0 | 0.0 | 0.0 | 0.0 | 0.0 | 0.0 | 0.0 | 0.0 | 0.0 | 0.0 | 0.0 | 0.0 | 0.0 | 0.0 | 0.0 | 0.0 | 0.0 | 0.0 |
| Llama 3.2 90B VI | 0.0 | 0.0 | 0.0 | 0.0 | 0.0 | 0.0 | 0.0 | 0.0 | 0.0 | 0.0 | 0.0 | 0.0 | 0.0 | 0.0 | 0.0 | 0.0 | 0.0 | 0.0 |
| Llama 4 Maverick | 0.0 | 0.0 | 0.0 | 0.0 | 0.0 | 0.0 | 0.0 | 0.0 | 0.0 | 0.0 | 0.0 | 0.0 | 0.0 | 0.0 | 0.0 | 0.0 | 0.0 | 0.0 |
| Llama 4 Scout | 0.0 | 0.0 | 0.0 | 0.0 | 0.0 | 0.0 | 0.0 | 0.0 | 0.0 | 0.0 | 0.0 | 0.0 | 2.6 | 0.0 | 0.0 | 0.0 | 0.0 | 0.0 |
| GPT 4.1 Mini | 0.0 | 0.0 | 0.0 | 0.0 | 0.0 | 0.0 | 0.0 | 0.0 | 0.0 | 0.0 | 0.0 | 0.0 | 0.0 | 0.0 | 0.0 | 0.0 | 0.0 | 0.0 |
| GPT 4.1 | 0.0 | 0.0 | 0.0 | 0.0 | 0.0 | 0.0 | 0.0 | 0.0 | 0.0 | 0.0 | 0.0 | 7.1 | 0.0 | 0.0 | 0.0 | 0.0 | 0.0 | 0.0 |
| o4-mini-high (T) | 0.0 | 0.0 | 0.0 | 0.0 | 0.0 | 0.0 | 0.0 | 0.0 | 0.0 | 0.0 | 0.0 | 0.0 | 0.0 | 0.0 | 0.0 | 0.0 | 0.0 | 0.0 |
| Qwen VP | 0.0 | 0.0 | 0.0 | 0.0 | 0.0 | 0.0 | 0.0 | 0.0 | 0.0 | 0.0 | 0.0 | 0.0 | 0.0 | 0.0 | 0.0 | 0.0 | 0.0 | 0.0 |
| Qwen 2.5 VL | 0.0 | 0.0 | 0.0 | 0.0 | 0.0 | 0.0 | 0.0 | 0.0 | 0.0 | 0.0 | 0.0 | 0.0 | 0.0 | 0.0 | 14.3 | 0.0 | 7.1 | 0.0 |
| Grok 2 Vision | 0.0 | 0.0 | 0.0 | 0.0 | 0.0 | 0.0 | 0.0 | 0.0 | 0.0 | 0.0 | 0.0 | 0.0 | 0.0 | 0.0 | 0.0 | 0.0 | 0.0 | 0.0 |
| Grok Vision Beta | 0.0 | 0.0 | 0.0 | 0.0 | 0.0 | 0.0 | 0.0 | 0.0 | 0.0 | 0.0 | 0.0 | 0.0 | 0.0 | 0.0 | 0.0 | 0.0 | 0.0 | 0.0 |

Table 15: Examples of Real-World Humanoid Constraint Scenarios from PAC Bench. Each scenario includes a question posed about a potential action and the ground-truth constraint explanation. Scenarios are captured using synchronized Agent View (from robot's perspective) and Side View cameras.

| Views Provided | Question Posed | Ground-Truth Constraint Explanation |
|---|---|---|
| Agent View (cam_0) Side View (cam_1) | Can the robot stack the object near the right hand on the object near the left hand? | No the cube won't balance on the pyramid. |
| Agent View (cam_0) Side View (cam_1) | Can we keep the ball inside the penstand? | No the the penstand is inverted. |
| Agent View (cam_0) Side View (cam_1) | Can we keep the ball inside the penstand? | No the the opening of the penstand is covered by the hand. |
| Agent View (cam_0) Side View (cam_1) | Can you keep the food on the plate? | No the box is closed. |
| Agent View (cam_0) Side View (cam_1) | Can you write on the notepad using the marker? | No the marker is closed. |
| Agent View (cam_0) Side View (cam_1) | Can you keep the food on the plate? | No the box is on the plate. |

## E.1 Annotation Pipelines

**Single-image (OpenImages).** We created one Label Studio[5] project per property. Each task presents a pre-cropped object (bounding box supplied) and radio-button choices covering the ontology plus *Don't Apply* and *Don't Know*. Annotators select exactly one option that best reflects the object's *current visual state* (e.g. a sauce-coated spoon is marked *Sticky*); an example interface is shown in Figure 15. Hot-keys (1–4 to pick a category, `Ctrl/Cmd+\Enter` to advance) support rapid, fatigue-free labelling. The per-property job dashboard is illustrated in Figure 14. Open-vocabulary affordances could not be captured with fixed radio buttons, so they were instead filled into a shared Google Sheet (≤3 verbs per image ID).

**Dual-view (*Real-Robo* & *RoboCasa*).** Label Studio does not support paired views, so we developed a lightweight Python/Tkinter GUI that shows the left/right camera frames side-by-side (Figures 12 and 13). The GUI mirrors the exact ontologies, category ordering, and hot-keys of the single-image pipeline and appends three affordance text boxes plus a drop-down for task-level constraints. For

---

[5] https://labelstud.io

completeness, the corresponding single-image TkInter variant used for synthetic objects is depicted in Figure 11.

## E.2 Annotation Schedule and Effort

Each property job comprises ~680 items and takes ≈40 minutes per annotator after a brief tutorial. All properties were labelled by at least two annotators to enable later consensus filtering (see below); several critical properties were triple-annotated when calendar time allowed. The total annotation effort is roughly 15 properties × 2.2 annotators × 40 min ≈ 22 person-hours for OpenImages and 7 person-hours for the dual-view collections.

## E.3 Quality Control

We employ a strict *unanimity filter*: for every image (or view-pair) the final label is retained only if *all* assigned annotators agreed. Disagreements are discarded from the main release (and provided as a separate "disagreement split") to guarantee that the benchmark set reflects high-confidence, noise-free supervision.

## E.4 Annotators

All annotations were performed by members of the LENS Lab (2024).[6] We thank the lab for their contributions and support.

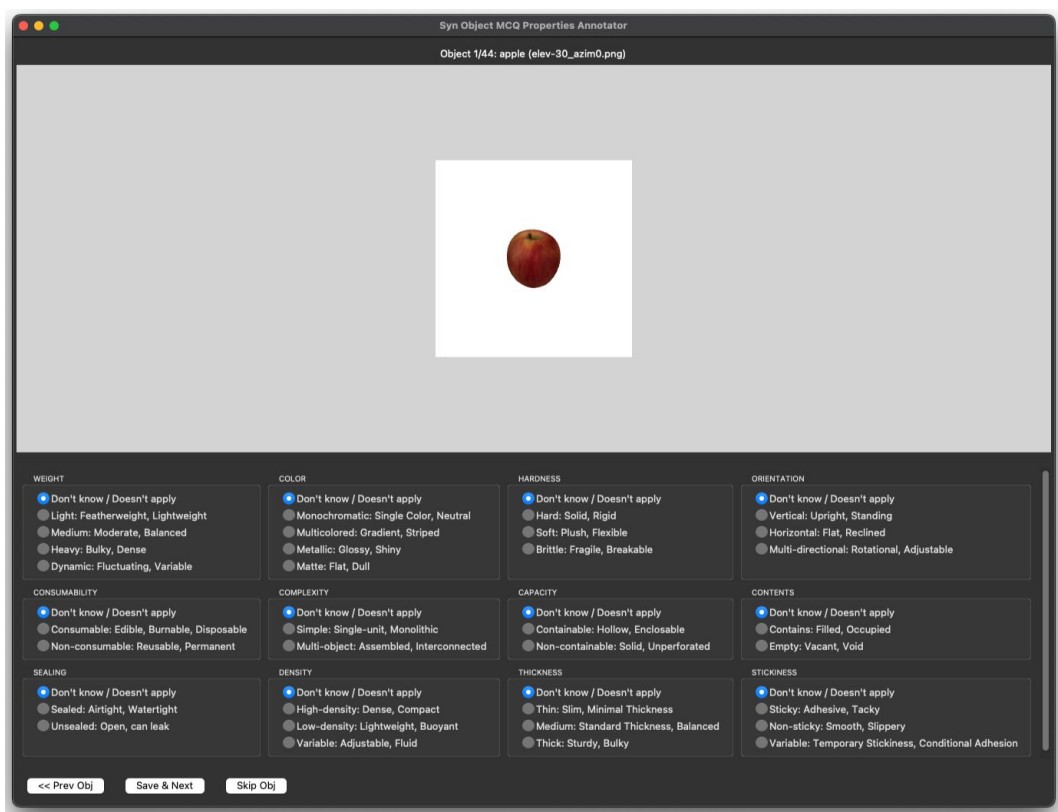

Figure 11: TkInter single-image property annotator (synthetic objects).

---

[6]https://ransml.github.io/lens-lab/

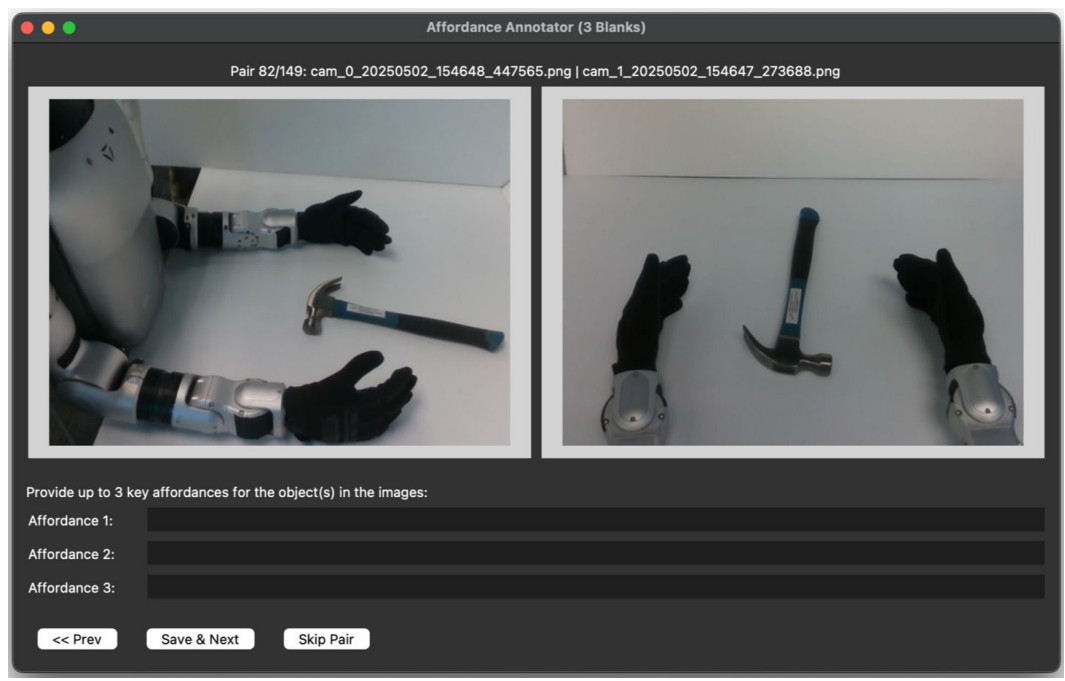

Figure 12: TkInter dual-view affordance annotator.

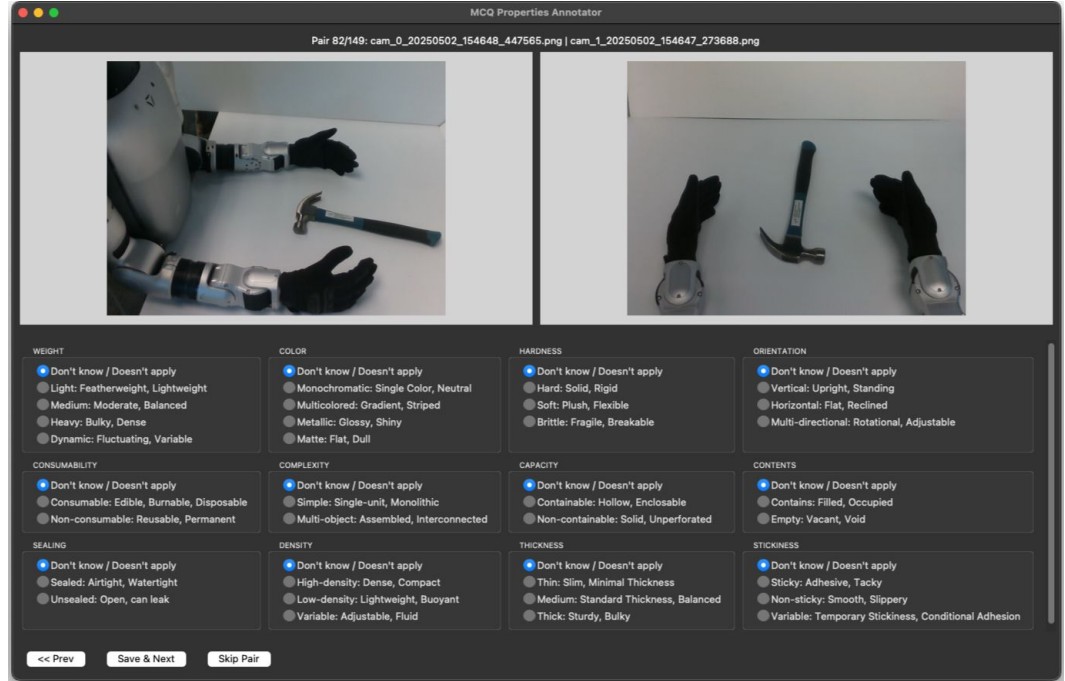

Figure 13: TkInter dual-view property annotator (Real-Robo / RoboCasa).

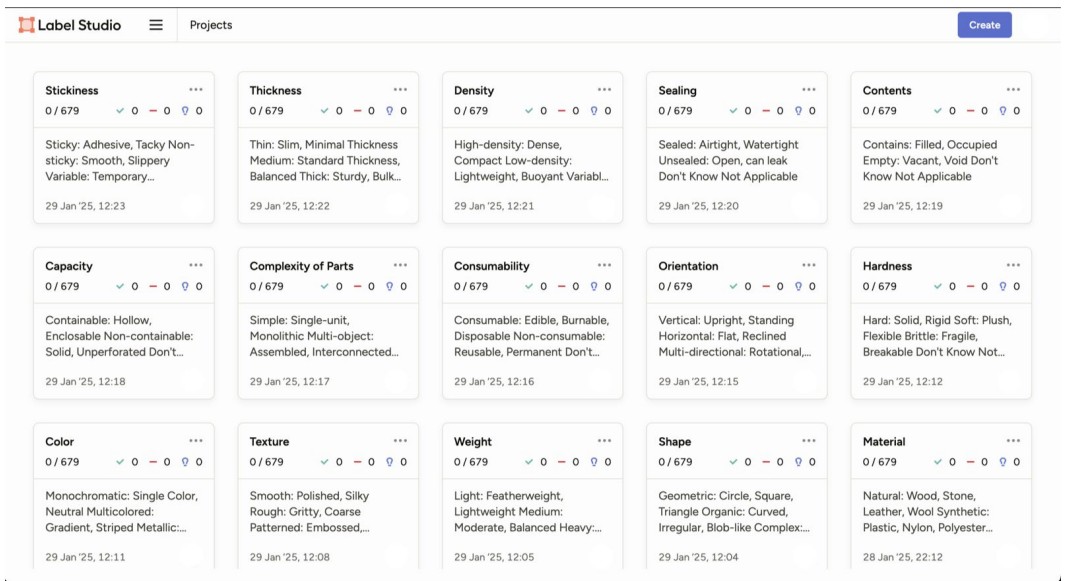

Figure 14: Label Studio project dashboard with 15 property jobs.

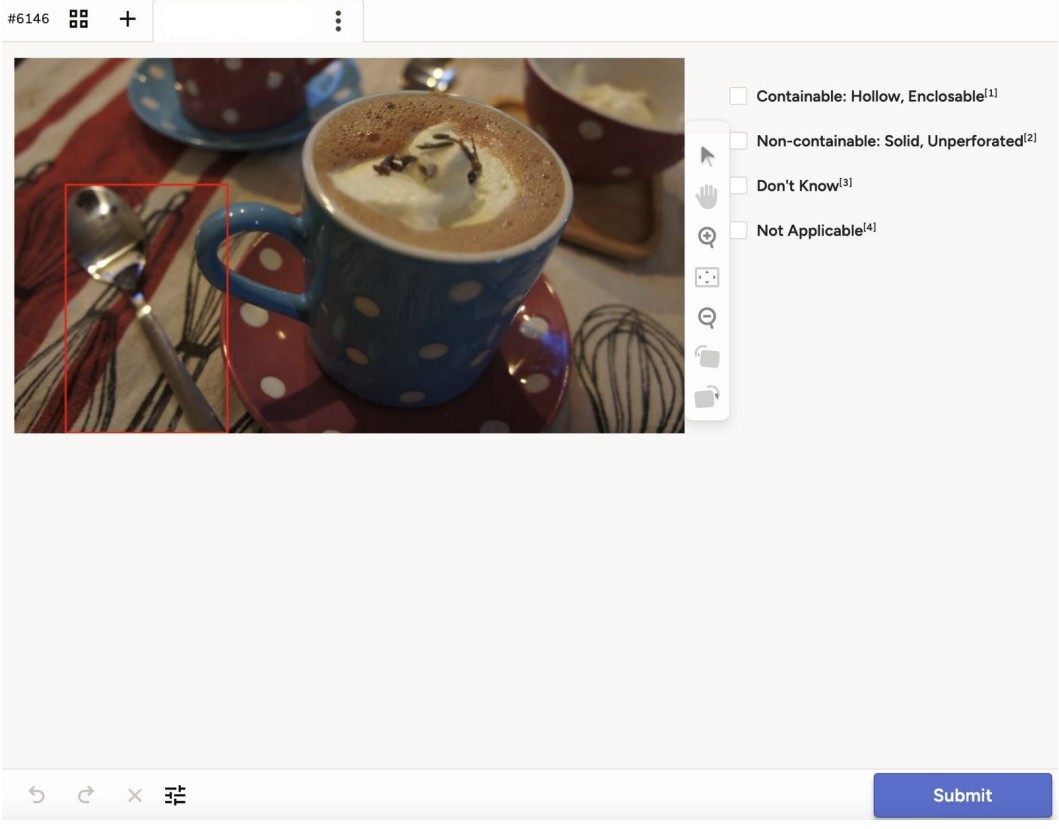

Figure 15: Label Studio image view with bounding box and radio-button options.

