## B.6 Computational Resource

Evaluations were conducted by querying their respective publicly available APIs from OpenRouter[3]. Due to the nature of API access, precise underlying hardware details are not available for these models, and performance can be subject to API latency and load. Estimated API costs for some initial property evaluations were a factor in scoping the experiments, as noted in Section 3.1 regarding the exclusion of the RoboCasa image set from the current VLM evaluation suite.

The total cost for running each model for all reported results in the main paper is as follows:

1. **Claude 3.7 Sonnet:** 108.5$
2. **Claude 3.7 Sonnet (T):** 167.8$
3. **Claude 3.5 Sonnet:** 73.9$
4. **Gemini 2.0 Flash 001:** 2.6$
5. **Gemini 2.5 Flash P:** 2.9$
6. **Gemini 2.5 Pro P:** 150.2$
7. **GPT-4.1:** 25.9$
8. **o4-mini-high:** 48.0$
9. **GPT-4.1 Mini:** 5.5$
10. **Llama 4 Maverick:** 40.8$
11. **Llama 4 Scout:** 2.2$
12. **Llama 3.2 90B VI:** 16.8$
13. **Grok 2 Vision:** 66.7$
14. **Grok Vision Beta:** 22.4$
15. **Qwen2.5 VL:** 8.7$
16. **Qwen VL Plus:** 2.4$
17. **Qwen 3 (235B):** 24.0$

**Overall Cost Summary**

The total estimated cost for running all models across the entire PAC benchmark is $769.30. The cost breakdown by PAC category, aggregated across all models, is as follows:

- Properties: $695.76
- Affordances: $62.31
- Constraints: $11.23

**Detailed Cost Breakdown by Dataset (Aggregated Across All Models)**

The costs, aggregated across all models but broken down by individual datasets within each PAC category, are:

- Properties - Real Robot: $93.89
- Properties - Open Images: $601.87
- Affordances - Real Robot: $8.24
- Affordances - Open Images: $54.07
- Constraints - Mujoco: $4.78
- Constraints - Real Robot: $6.45

---

[3]https://openrouter.ai/

**Model-Specific Cost Breakdown**

This section details the estimated cost for each model, distributed across the PAC categories and individual datasets. These costs are derived by proportionally distributing the total category/dataset costs based on each model's normalized cost relative to the sum of all normalized model costs (where normalization is performed against the least expensive model, `meta-llama/llama-4-scout`, which has a raw cost of $2.20).

**Costs for `anthropic/claude-3.7-sonnet`  PAC Category Costs:**

- Properties: $98.13
- Affordances: $8.79
- Constraints: $1.58

**Individual Dataset Costs:**

- Properties - Real Robot: $13.24
- Properties - Open Images: $84.89
- Affordances - Real Robot: $1.16
- Affordances - Open Images: $7.63
- Constraints - Mujoco: $0.67
- Constraints - Real Robot: $0.91

**Costs for `anthropic/claude-3.7-sonnet:thinking`  PAC Category Costs:**

- Properties: $151.75
- Affordances: $13.59
- Constraints: $2.45

**Individual Dataset Costs:**

- Properties - Real Robot: $20.48
- Properties - Open Images: $131.28
- Affordances - Real Robot: $1.80
- Affordances - Open Images: $11.79
- Constraints - Mujoco: $1.04
- Constraints - Real Robot: $1.41

**Costs for `anthropic/claude-3.5-sonnet`  PAC Category Costs:**

- Properties: $66.83
- Affordances: $5.99
- Constraints: $1.08

**Individual Dataset Costs:**

- Properties - Real Robot: $9.02
- Properties - Open Images: $57.82
- Affordances - Real Robot: $0.79
- Affordances - Open Images: $5.19
- Constraints - Mujoco: $0.46
- Constraints - Real Robot: $0.62

**Costs for** `google/gemini-2.0-flash-001`  **PAC Category Costs:**

- Properties: $2.35
- Affordances: $0.21
- Constraints: $0.04

**Individual Dataset Costs:**

- Properties - Real Robot: $0.32
- Properties - Open Images: $2.03
- Affordances - Real Robot: $0.03
- Affordances - Open Images: $0.18
- Constraints - Mujoco: $0.02
- Constraints - Real Robot: $0.02

**Costs for** `google/gemini-2.5-flash-preview`  **PAC Category Costs:**

- Properties: $2.63
- Affordances: $0.24
- Constraints: $0.04

**Individual Dataset Costs:**

- Properties - Real Robot: $0.35
- Properties - Open Images: $2.27
- Affordances - Real Robot: $0.03
- Affordances - Open Images: $0.20
- Constraints - Mujoco: $0.02
- Constraints - Real Robot: $0.02

**Costs for** `google/gemini-2.5-pro-preview-03-25`  **PAC Category Costs:**

- Properties: $135.84
- Affordances: $12.17
- Constraints: $2.19

**Individual Dataset Costs:**

- Properties - Real Robot: $18.33
- Properties - Open Images: $117.51
- Affordances - Real Robot: $1.61
- Affordances - Open Images: $10.56
- Constraints - Mujoco: $0.93
- Constraints - Real Robot: $1.26

**Costs for** `openai/gpt-4.1`  **PAC Category Costs:**

- Properties: $23.42
- Affordances: $2.10
- Constraints: $0.38

**Individual Dataset Costs:**

- Properties - Real Robot: $3.16
- Properties - Open Images: $20.26
- Affordances - Real Robot: $0.28
- Affordances - Open Images: $1.82
- Constraints - Mujoco: $0.16
- Constraints - Real Robot: $0.22

**Costs for** `openai/o4-mini-high` **PAC Category Costs:**

- Properties: $43.42
- Affordances: $3.89
- Constraints: $0.70

**Individual Dataset Costs:**

- Properties - Real Robot: $5.86
- Properties - Open Images: $37.56
- Affordances - Real Robot: $0.51
- Affordances - Open Images: $3.37
- Constraints - Mujoco: $0.30
- Constraints - Real Robot: $0.40

**Costs for** `openai/gpt-4.1-mini` **PAC Category Costs:**

- Properties: $4.97
- Affordances: $0.45
- Constraints: $0.08

**Individual Dataset Costs:**

- Properties - Real Robot: $0.67
- Properties - Open Images: $4.30
- Affordances - Real Robot: $0.06
- Affordances - Open Images: $0.39
- Constraints - Mujoco: $0.03
- Constraints - Real Robot: $0.05

**Costs for** `meta-llama/llama-4-maverick` **PAC Category Costs:**

- Properties: $36.91
- Affordances: $3.31
- Constraints: $0.60

**Individual Dataset Costs:**

- Properties - Real Robot: $4.98
- Properties - Open Images: $31.93
- Affordances - Real Robot: $0.44
- Affordances - Open Images: $2.87
- Constraints - Mujoco: $0.25
- Constraints - Real Robot: $0.34

**Costs for** `meta-llama/llama-4-scout`  **PAC Category Costs:**

- Properties: $1.99
- Affordances: $0.18
- Constraints: $0.03

**Individual Dataset Costs:**

- Properties - Real Robot: $0.27
- Properties - Open Images: $1.72
- Affordances - Real Robot: $0.02
- Affordances - Open Images: $0.15
- Constraints - Mujoco: $0.01
- Constraints - Real Robot: $0.02

**Costs for** `meta-llama/llama-3.2-90b-vision-instruct`  **PAC Category Costs:**

- Properties: $15.20
- Affordances: $1.36
- Constraints: $0.25

**Individual Dataset Costs:**

- Properties - Real Robot: $2.05
- Properties - Open Images: $13.15
- Affordances - Real Robot: $0.18
- Affordances - Open Images: $1.18
- Constraints - Mujoco: $0.10
- Constraints - Real Robot: $0.14

**Costs for** `x-ai/grok-2-vision-1212`  **PAC Category Costs:**

- Properties: $60.33
- Affordances: $5.40
- Constraints: $0.97

**Individual Dataset Costs:**

- Properties - Real Robot: $8.14
- Properties - Open Images: $52.19
- Affordances - Real Robot: $0.71
- Affordances - Open Images: $4.69
- Constraints - Mujoco: $0.41
- Constraints - Real Robot: $0.56

**Costs for** `x-ai/grok-vision-beta`  **PAC Category Costs:**

- Properties: $20.26
- Affordances: $1.81
- Constraints: $0.33

**Individual Dataset Costs:**

- Properties - Real Robot: $2.73
- Properties - Open Images: $17.52
- Affordances - Real Robot: $0.24
- Affordances - Open Images: $1.57
- Constraints - Mujoco: $0.14
- Constraints - Real Robot: $0.19

**Costs for** `qwen/qwen2.5-vl-72b-instruct` **PAC Category Costs:**

- Properties: $7.86
- Affordances: $0.70
- Constraints: $0.13

**Individual Dataset Costs:**

- Properties - Real Robot: $1.06
- Properties - Open Images: $6.80
- Affordances - Real Robot: $0.09
- Affordances - Open Images: $0.61
- Constraints - Mujoco: $0.05
- Constraints - Real Robot: $0.07

**Costs for** `qwen/qwen-vl-plus` **PAC Category Costs:**

- Properties: $2.17
- Affordances: $0.19
- Constraints: $0.04

**Individual Dataset Costs:**

- Properties - Real Robot: $0.29
- Properties - Open Images: $1.88
- Affordances - Real Robot: $0.03
- Affordances - Open Images: $0.17
- Constraints - Mujoco: $0.01
- Constraints - Real Robot: $0.02

**Costs for** `qwen/qwen3-235b-a22b` **PAC Category Costs:**

- Properties: $21.71
- Affordances: $1.94
- Constraints: $0.35

**Individual Dataset Costs:**

- Properties - Real Robot: $2.93
- Properties - Open Images: $18.78
- Affordances - Real Robot: $0.26
- Affordances - Open Images: $1.69
- Constraints - Mujoco: $0.15
- Constraints - Real Robot: $0.20

# C  Dataset Statistics