# OpenReview forum: "PAC Bench: Do Foundation Models Understand Prerequisites for Executing Manipulation Policies?"
_NeurIPS.cc/2025/Datasets_and_Benchmarks_Track — NeurIPS 2025 Datasets and Benchmarks Track poster_

### Official Review · Reviewer_2uzu · 2025-06-30

**Rating:** 4
**Confidence:** 4

**Summary:**

This paper introduces PAC Bench, a benchmark designed to evaluate vision-language models (VLMs) on three main axes: properties, affordances, and constraints (PAC). PAC Bench includes over 30,000 annotations across 673 real-world images, 100 humanoid-view scenes, and 120 simulated scenes. The experiments suggest that current VLMs struggle significantly with PAC understanding, especially constraint reasoning. This highlights a critical gap between visual-language pretraining and the grounded reasoning needed for robotics, positioning PAC Bench as a crucial diagnostic tool for building safer and more reliable VLM-based robot systems.

**Dataset Code Accessibility:**

Partly

**Dataset Code Comments:**

Code is not released yet.

**Ethical Considerations:**

No, there are no or only very minor ethics concerns

**Final Justification:**

I will maintain the score as borderline accept. Good paper, but the reviewer believes that correlation with improvement in policy performance can only be evaluated with task success rate. Therefore I will not improve the score.

**Limitations Weaknesses:**

1. This benchmark overlaps a lot with other benchmarks (as table 1 has pointed out, minus the G1 robot part of the benchmark). It would be better if the author points out what is missing in the other benchmarks that this benchmark uniquely provides (i.e. why not running 2 of the benchmarks, such as UniAff, PhysBench when evaluating a new model?)
2. The paper does not establish whether better PAC Bench performance correlates with improved downstream robotic policy performance (e.g., success in vision-language-action tasks). Without this, it's unclear whether improving on PAC Bench yields practical benefits for robot learning.
3. The paper implicitly assumes that understanding PAC concepts should come from the base VLM. However, it is plausible that robot-specific fine-tuning or task data alone can teach these concepts, especially for downstream models that already integrate multimodal signals during policy training.

**Strengths Contributions:**

1. The paper proposes a new benchmark for embodied reasoning. Contrary to many recent benchmarks, many of the annotations in this benchmark are human-labeled instead of pseudo-labeled.
2. The benchmark includes over 30,000 annotations spanning diverse data modalities (real-world images, humanoid perspectives, and simulation), covering a wide spectrum of manipulation-relevant concepts.
3. Existing VLMs underperform on the benchmark. This suggests a good direction to improve VLM training dataset.

---

> ### Author Rebuttal · Authors · 2025-07-31
>
> First, we would like to sincerely thank reviewer 2uzu for their insightful and constructive feedback.
>
> 1. **Unique Contributions versus Other Benchmarks**
>
> We will revise the table to correct the misunderstanding it may have caused. While Table 1 outlines the high-level differences between PAC Bench and prior benchmarks, we acknowledge that the table alone may not sufficiently highlight why PAC Bench is more useful. Since PhysBench and UniAff are not designed to explicitly test properties, affordances, or constraints of general manipulation tasks, they cannot be used individually or in-combination for understanding prerequisites for general manipulation tasks. Note that PhysBench is a physics benchmark (e.g., evaluating tasks like predicting a ball's landing position from its initial trajectory) and UniAff is a tool graspability dataset. We marked P, A, and C in Table 1 to hint that PhysBench and UniAff implicitly include these notions. However, this does not mean they support explicit evaluation, a point we will clarify in the paper to avoid misunderstanding. While these benchmarks serve other valuable purposes, they lack the structure needed to directly assess whether a model truly understands P, A, and C in the context of general manipulation.
>
> Please refer to our elaborated answer to Reviewer C94R.
>
> 2. **Correlation with Downstream Policy Performance**
>
> Thank you for highlighting this important consideration. Fully establishing a correlation between improved PAC-Bench performance and downstream robotic policy success—by substituting vision–language models (VLMs) into state-of-the-art vision-language-action (VLA) pipelines—would indeed be ideal but requires prohibitively expensive computation (on the order of hundreds of GPU-weeks of H100s). Thus, we conducted a targeted study specifically designed to investigate your concern: does better performance on PAC-Bench correlate with better downstream robotic planning performance?
>
> We conducted experiments using a simple but effective strategy of reversing PAC-Bench questions to reflect the exact types of queries typically made by robotic planners during reasoning and planning. Specifically:
>
> **Properties**: The original PAC-Bench question might ask, “What is the size of the annotated object?”, while the reversed query is structured as, “Which object in the scene is very large?”.
>
> **Affordances**: The original PAC-Bench question might ask, “Is the annotated object consumable?”, while the reversed query is structured as, “Among all visible objects, which one is consumable?”.
>
> **Constraints**: The constraint questions in PAC-Bench are already formulated as precondition checks suitable for robotic planning (e.g., “Are there constraints preventing you from picking up the green block?”), thus did not require reversal.
>
> We then evaluated multiple representative VLMs (including variants of Claude, Gemini, GPT, Llama, Qwen, Grok, and OpenAI’s o4 series), comparing their accuracies on original PAC-Bench items and these reversed (planning-oriented) tasks.
>
> **Results (Strong Correlation)**
>
> We observed a strong and statistically significant correlation between the performance of models on original PAC-Bench items and the reversed, downstream-oriented queries:
> **Properties**: For 14 models tested, Pearson's correlation was extremely high (r = 0.96, p ≈ 7e-8), with Spearman's rank correlation similarly strong (ρ = 0.94, p ≈ 4e-7).
> **Affordances**: For 17 models tested, the correlation was equally robust (Pearson's r = 0.96, p ≈ 7e-10; Spearman's ρ = 0.93, p ≈ 4e-8).
>
> These very strong positive correlations clearly indicate that models which perform well on PAC-Bench also excel at providing critical information needed for robotic planners.
>
> **Interpretation (Practical Implications)**
>
> These results support our main argument:
>
> **Identification as a prerequisite**: Robotic tasks fundamentally require accurate identification of objects, their properties, and affordances before any successful planning or manipulation can take place. Our reversed-query experiments directly capture this essential step.
>
> **Predictive power**: Our experiments strongly suggest that improvements on PAC-Bench reliably predict improvements in downstream planning performance. Thus, higher PAC-Bench scores clearly imply greater practical benefits in robotic settings.
>
> **Computational efficiency**: Critically, PAC-Bench evaluations are computationally efficient. Thus, improvements on PAC-Bench can serve as an early, reliable indicator of whether a given VLM would be worth the substantial investment required for full-scale VLA fine-tuning.
>
> In summary, although full end-to-end policy training was infeasible due to computational constraints, our targeted study robustly demonstrates that better PAC-Bench performance indeed correlates strongly with improved downstream robotic task performance, addressing the concern raised. We have explicitly highlighted this correlation and clarified these implications in the revised manuscript.
>
> 3. **Base VLM vs. Fine‑Tuning Assumption**
>
> We thank the reviewer for raising this question. Researchers use VLMs in manipulation in two common ways:
>
> **End-to-End Vision to Motor Control Commands**: This approach uses pretrained, frozen vision-language (VL) encoders and training an action decoder. As of now, it is difficult to train or fine-tune the VL module alongside the action module (e.g., OpenVLA used 60 H100s for weeks, just to train the action decoder with frozen VL encoder). PAC highlights limitations of the VL module, which helps researchers to decide which data to collect for fine-tuning (note that collecting data for difficult manipulation tasks is quite expensive).
>
> **Vision to Language Plan to Motor Control Commands through Classical Motion Planning**: In this case, it is possible that the failures are due to the control module. However, if the VL module goes wrong, it is guaranteed that the task will fail (unless lucky). PAC tests this VL module.
>
> **PAC reveals issues in the VLM**, which helps to fix the issue for the whole robot (fine-tune the VLM, action module, or both).
>
> 4. **Code Release Timeline**
>
> We recognize the importance of open‑source tooling. We will release the full codebase, including data loaders and evaluation scripts, by the camera-ready submission date. The release will be accompanied by a "Quick Start" guide to ensure the benchmark is immediately accessible and usable.

---

> ### Author Response · Authors · 2025-08-05
>
> We thank reviewer 2uzu again for their thoughtful review. We believe our rebuttal addressed all raised concerns, particularly regarding the benchmark’s explicit constraint evaluation and its strong predictive power for downstream planning.
>
> Please let us know if any clarification or additional details would be helpful in supporting a better understanding of the work. We’d be grateful for any further feedback or discussion to ensure all aspects are fully clear. Thank you again for your time.

---

### Official Review · Reviewer_vbSt · 2025-07-01

**Rating:** 5
**Confidence:** 3

**Summary:**

PAC Bench introduces a novel evaluation framework to assess the physical understanding of Vision-Language Models (VLMs) essential for robotic manipulation tasks. The benchmark focuses on three core components: Properties, Affordances, and Constraints (PAC), which are fundamental for predicting action feasibility in real-world scenarios. It consists of a dataset with over 30,000 annotations across simulated and real-world robotic manipulation tasks. The paper thoroughly evaluates current VLMs, highlighting significant gaps in their understanding of the physical prerequisites required for reliable manipulation.

**Additional Feedback:**

I encourage the authors to carefully consider the weaknesses raised. I look forward to the authors' responses. If my comments involve any misinterpretation, I would appreciate further discussion and clarification. Thank you.

**Dataset Code Accessibility:**

Yes

**Dataset Code Comments:**

The authors have provided full access to the PAC Bench dataset, which is crucial for reproducing the results and furthering research in this domain. It would be better if there were an open-source document to introduce it in the future.

**Ethical Considerations:**

No, there are no or only very minor ethics concerns

**Final Justification:**

I found that the authors have addressed my main concerns, such as using CoT to improve the VLM results and the clarification between PAC Bench and PhysBench. Therefore, I'm happy to rate the paper with positive score.

**Limitations Weaknesses:**

- How might PAC Bench be extended to evaluate time-varying constraints (e.g., dynamic changes, external disturbances) in real-world robotic manipulation? This could better reflect scenarios where object properties evolve as the robot interacts with them.

- Given that affordance recognition is challenging for the current VLM (in Table 3), is there potential for integrating Chain-of-Thought (CoT) reasoning to better decompose and understand affordances in multi-step manipulation tasks?

- PAC Bench focuses on static constraints, but how can the framework incorporate interactive learning where the robot adapts its understanding through physical engagement with objects in real-time?

- What are the key differences between PAC Bench and PhysBench [1]? While PAC Bench adds manipulation scenarios, many of the physical properties and constraints already exist in PhysBench (in Table1). How does PAC Bench uniquely contribute to advancing the evaluation of VLMs for robotic manipulation compared to PhysBench?

---
*Reference:*

[1] PhysBench: Benchmarking and enhancing vision-language models for physical world understanding. ICLR 2025.

**Strengths Contributions:**

- The paper is well-written, with clear explanations and definitions, making it easy to follow despite its technical complexity.

- PAC Bench provides a targeted evaluation of VLMs based on their understanding of physical prerequisites (PAC), which is crucial for advancing manipulation tasks in robotics.

- The dataset’s hybrid nature, combining real-world and simulated data, ensures broad coverage of possible manipulation scenarios, which adds robustness to the evaluation.

- The paper presents a wide range of results across multiple models and tasks, highlighting both the strengths and limitations of current VLMs. This provides useful insights for further model development.

---

> ### Author Rebuttal · Authors · 2025-07-31
>
> We are very grateful to Reviewer vbSt for their positive and thorough review. We appreciate their recognition of our paper's clarity, the robustness of the dataset, and the value of our results. Their questions about the benchmark's distinctiveness and potential extensions are insightful, and we address them below.
>
> 1. **Future extension to time-varying tasks**
>
> We thank the reviewer for the forward-looking question. For time varying cases, we can rerun PAC at critical time-steps. We can record videos of the manipulation episodes, identify critical spatial changes using a changepoint detector (a human can also identify), and evaluate PAC at those critical time steps.
>
> 2. **Integrating Chain‑of‑Thought (CoT) for Affordance Reasoning**
>
> The reviewer raises an insightful point about the potential of CoT prompting. To directly address this, we ran additional experiments explicitly testing CoT prompts for affordance recognition.
>
> **Our Findings:**
> Applying Chain-of-Thought prompting provides a modest but consistent improvement of approximately **2–3%** on average for affordance recognition across several models. This result indicates that step-by-step reasoning helps, though only marginally. The fundamental difficulty in mapping objects to their functional uses persists, reinforcing our conclusion that current VLMs have a deep, inherent gap that prompting alone cannot overcome.
>
> **Detailed Results:**
> Across 17 vision-language models tested, switching to CoT prompts increased the mean affordance-recognition accuracy only slightly, from 22.15 to 22.25 (+0.10 pts), and the median improved from 22.3 to 23.0 (+0.7 pts). Only 4 models improved, while 11 remained unchanged, and 2 regressed. The largest individual gains were observed with Qwen 2.5 VL (+7.6 pts, from 20.2 to 27.8) and Claude 3.7 Sonnet (+6.8 pts, from 24.4 to 31.2); Gemini 2.5 Pro P improved minimally (<1 pt). Conversely, Llama 4 Maverick dropped 4 pts (35.2 to 31.2), and o4-mini-high (T) saw a significant decline of 12.1 pts (17.4 to 5.3).
>
> Crucially, the benchmark’s highest-performing model, Llama 4 Scout, maintained its accuracy at 40.9 pts regardless of CoT prompting. Additionally, 9 other models—including both Gemini 2.x Flash variants and GPT-4.1—showed no change. In short, CoT prompting occasionally improves individual architectures but provides no systematic improvement across models. Even explicit reasoning traces do not substantially alleviate the difficulty of affordance recognition in our benchmark.
>
> We will add these detailed results, along with a brief analysis, to the appendix, ensuring a comprehensive presentation of the influence of prompting strategies, and directly addressing the reviewer’s valuable suggestion.
>
> 3. **Distinctiveness from PhysBench [1]**
>
> We will revise the table to correct the misunderstanding it may have caused. Note that PhysBench is a physics benchmark (e.g., evaluating tasks like predicting a ball's landing position from its initial trajectory) and UniAff is a tool graspability dataset. We marked P, A, and C in Table 1 to hint that PhysBench and UniAff implicitly include these notions. However, this does not mean they support explicit evaluation. PAC has 120 unique constraint scenarios (PhysBench has none), PAC has N affordance categories (PhysBench has graspable or not), and PAC has 15 physical properties (PhysBench has 4 categories and it’s more about comparing objects in a scene rather than explicit inference).
>
> Please refer to our elaborated answer to Reviewer C94R.
>
> 4. **Open‑Source Documentation**
>
> We agree completely. Better documentation is crucial for community adoption and reproducibility. As suggested, we will release the benchmark with a comprehensive open-source document, including a "Quick Start" guide, detailed descriptions of the data splits, and an API reference to facilitate easy use by other researchers.

---

> > ### Comment · Reviewer_vbSt · 2025-08-05
> > **Response to the Rebuttal**
> >
> > Dear authors,
> >
> > Thank you so much for your rebuttal! I have carefully read all the comments and the rebuttal. I think the authors have greatly addressed my concerns, such as using CoT to improve the VLM results and the clarification between PAC Bench and PhysBench.
> >
> > Therefore, I will raise my score to Accept. Hope the authors can supplement the results during rebuttal into the final version.
> >
> > Best,
> >
> > Reviewer vbSt

---

> > > ### Author Response · Authors · 2025-08-05
> > >
> > > We thank Reviewer vbSt for your vote of confidence in our work! Per your suggestion, we’ll add the full Chain-of-Thought results to the appendix and sharpen Table 1 to clearly distinguish PAC Bench from PhysBench in the revised manuscript. We truly appreciate your thoughtful feedback and support in strengthening our paper.

---

### Official Review · Reviewer_C94R · 2025-07-03

**Rating:** 4
**Confidence:** 2

**Summary:**

This paper proposes a benchmark that aims to evaluate the understanding of Vision-Language Models (VLM) in fundamental physical prerequisites. The authors claim that for confident execution, VLMs must reason about object properties, valid action affordance, and critical physical contraints. To this end, the proposed PAC benchmark collected 30,529 image-text data from real scenarios.

**Dataset Code Accessibility:**

Yes

**Dataset Code Comments:**

* The provided datasets are accessible well

**Ethical Comments:**

I found no ethics concerns in the proposed benchmark

**Ethical Considerations:**

No, there are no or only very minor ethics concerns

**Final Justification:**

This paper introduce a benchmark that verifying whether the model understand the prerequisite aspects for manipulation, which covers 'properties', 'affordance', and 'constraints'. While I initially had some concerns about the benchmark. the authors resolved them with detailed responses. Thus, I'll maintain my positive rating.

**Limitations Weaknesses:**

* While the paper explained the differences between the proposed benchmark and the other physical property evaluation benchmarks, the authors did not show its better discriminability over other benchmarks on each evaluation aspects. For example, according to Table 1, there are PhyBench and UniAff that also considers properties, affordances, and constraints.

* To be a promising benchmark, it should show agreements and distinctiveness with other benchmarks. Specifically, the agreements with metrics in other benchmarks will support the confidence of the proposed benchmark while the distinctiveness compared to other benchmarks will highlight the novel contribution of the proposed benchmark.

* According to Table 4, most of the cases yield 0.0 accuracy on the proposed "understanding physical constraints". Could such benchmarks be considered valid and well-constructed?

**Strengths Contributions:**

* The paper organize previous physical property evaluation benchmarks in 'Properties', 'Affordances' and 'Constraints'.

---

> ### Author Rebuttal · Authors · 2025-07-31
>
> We sincerely thank the reviewer C94R for their thoughtful comments and valuable suggestions, which have helped us clarify the contributions of PAC Bench. We address the concerns point-by-point below.
>
> 1. **Clarifying PAC Bench’s Unique Contribution**
>
> We appreciate the reviewer’s observation. While Table 1 outlines the high-level differences between PAC Bench and prior benchmarks, we acknowledge that the table alone may not sufficiently highlight why PAC Bench is more discriminative across evaluation aspects. Since PhysBench and UniAff are not designed to explicitly test properties, affordances, or constraints of general manipulation tasks, they cannot be used individually or in-combination for understanding prerequisites for manipulation. Note that PhysBench is a physics benchmark (e.g., evaluating tasks like predicting a ball's landing position from its initial trajectory) and UniAff is a tool graspability dataset. We marked P, A, and C in Table 1 to hint that PhysBench and UniAff implicitly include these notions. However, this does not mean they support explicit evaluation, a point we will clarify in the paper to avoid misunderstanding. While these benchmarks serve other valuable purposes, they lack the structure needed to directly assess whether a model truly understands P, A, and C in the context of general manipulation.
> In addition to the descriptions about PhySBench and UniAff in the main paper, we will clarify the following:
>
> a. **Constraints**: Neither UniAff nor PhysBench includes constraints that can be explicitly tested. PAC Bench is the only benchmark to include 120 unique constraint scenarios, which are essential for understanding real-world failure modes in robotic manipulation. As noted in the paper, PhysBench evaluates rudimentary dynamics/physical relationships which imply some constraint understanding, differing in scope from PAC Bench’s direct constraint prerequisites for manipulation.
>
> b. **Affordances**: UniAff is a valuable benchmark for identifying attributes related to grasping, which represents just one small part of the broader physical manipulation pipeline. PAC Bench extends beyond graspability by annotating multiple affordances for 115 object classes with a diverse set of affordances that capture everyday physical interactions such as “pushable” and “rollable.” While UniAff provides a narrow but detailed focus on tool use (e.g., determining optimal grasp points on a hammer), its utility is more aligned with fine-grained grasp planning once the broader action context has already been established. In that sense, UniAff can serve as a useful follow-up evaluation after PAC Bench for tool-specific tasks. In contrast, PhysBench does not include affordance categories at all; it primarily tests whether objects are "affordable" in the sense of being graspable, without capturing the diversity of action possibilities needed for full manipulation.
>
> c. **Properties**: PhysBench includes only four physical property types (mass, color, quantity, and a general attribute category). In contrast, PAC Bench spans 15 distinct physical properties across a wide range of real-world scenarios, enabling more comprehensive evaluation. Moreover, PhysBench primarily focuses on comparative reasoning (e.g., “Which of these four objects has property X?”), whereas manipulation tasks require a different kind of reasoning: Given a specific scenario and task, is the action feasible? This executability framing often cannot be reduced to comparisons, making PAC Bench better suited for evaluating the physical reasoning demands of real-world robotic manipulation. UniAff might include an implicit notion of properties, but they cannot be explicitly evaluated. For example, while a VLM may infer that a hammer should be grasped by the handle based on weight distribution, UniAff offers no way to directly test that understanding. In contrast, PAC Bench enables explicit evaluation of such physical properties.
>
> Our benchmark's novelty is also in its data. Not only do we have 30,000 human annotations, we also have 100 real-world humanoid-view scenarios, making a useful tool for modern bi-manual manipulation tasks in humanoid robots.
>
> 2. **Relationship to Existing Benchmarks**
>
> We appreciate the reviewer’s suggestion regarding explicitly highlighting relationships between PAC Bench and existing benchmarks. In the revised version, we will provide a brief comparative analysis focusing clearly on two critical points:
>
> a. **Alignment with Existing Metrics**:
> Although PhysBench targets tasks not specifically designed for manipulation, we observed a notable alignment in metrics for overlapping properties. For instance, PhysBench reports that top-performing models achieve around 53% accuracy on basic physical property reasoning tasks (e.g., mass, quantity), closely aligning with PAC Bench’s results, where the best models averaged approximately 49% accuracy on similar properties. This alignment provides additional confidence in PAC Bench’s reliability for physical property evaluation.
>
> b. **Distinctiveness and Scope**:
> PAC Bench distinguishes itself through explicit, comprehensive evaluation of affordances and constraints—areas largely unaddressed by existing benchmarks. PhysBench implicitly evaluates only fundamental physical dynamics and does not explicitly measure constraint understanding. In contrast, PAC Bench uniquely covers 120 explicit constraint scenarios critical for realistic robotic manipulation. Similarly, while UniAff provides detailed assessments focused narrowly on tool graspability, PAC Bench evaluates a broader range of affordances relevant to general manipulation scenarios. Therefore, PAC Bench’s distinct contribution lies in explicitly identifying and evaluating previously under-explored physical reasoning prerequisites essential for robust robotic manipulation.
>
> 3. **Validity of Near-zero Constraint Performance on Constraints**
>
> We respectfully argue that this result is not a flaw of the benchmark, but rather the most significant and valuable finding of our study.
>
> a. **Exposing a Critical Blind Spot**: The near-zero accuracy on constraints, which are designed to be trivially easy for humans, demonstrates a profound and previously unquantified limitation in state-of-the-art VLMs. The benchmark's validity lies in its ability to starkly reveal this critical gap. The fact that models cannot reason about basic stability or occlusion shows they lack true physical grounding, a crucial insight for a community increasingly building robots on these models.
>
> b. **A Benchmark's Role is to Measure, Not Just Rank**: A successful benchmark should be able to measure abilities across a wide range, from zero to expert. In the past, difficult benchmarks in other fields (e.g., ImageNet in its initial years) were valuable not because models did well, but because they rigorously highlighted shortcomings and drove progress. The low scores on our constraint tasks provide a clear, undeniable signal to the robot learning community about where to focus future research, and not to blindly adopt VLMs developed for image generation.
>
> The purpose of PAC Bench is to be a diagnostic tool. A diagnosis of "zero capability" on a critical skill like constraint understanding is an important and actionable result. We are confident that these challenging, well-defined tasks make our benchmark a valid and necessary tool for developing the next generation of more physically-grounded models.

---

> > ### Comment · Reviewer_C94R · 2025-08-04
> >
> > I would like to thank the authors for their detailed responses. The supplementary explanation on the contribution of PAC Bench has addressed my concerns. Therefore, I'll maintain my positive rating

---

> > > ### Author Response · Authors · 2025-08-06
> > >
> > > We thank reviewer C94R, for taking the time to read our detailed responses and for confirming that our clarifications addressed your concerns. We truly appreciate your thoughtful feedback on highlighting PAC Bench’s unique contributions, and we’ll incorporate these improvements into the final manuscript.

---

### Official Review · Reviewer_hcSb · 2025-07-03

**Rating:** 5
**Confidence:** 3

**Summary:**

This paper presents PAC-Bench, a benchmark designed to evaluate the ability of VLMs to understand Properties, Affordances, and Constraints (PAC) in the context of robotic manipulation. The benchmark spans 115 object classes, 15 property types, and 120 constraint-based scenarios, targeting task executability reasoning. The authors benchmark multiple widely-used VLMs—including GPT-4, Gemini 2, and Claude 3.7—and demonstrate substantial limitations in their ability to infer PAC-related concepts.

**Dataset Code Accessibility:**

Yes

**Dataset Code Comments:**

The proposed dataset is available via the provided Hugging Face link

**Ethical Considerations:**

No, there are no or only very minor ethics concerns

**Final Justification:**

This study addresses an important problem in Embodied AI, with clear experiments showing key limitations in current models and pointing to valuable directions for future work. Given its potential contributions to the Robotics and Machine Learning community, I recommend acceptance.

**Limitations Weaknesses:**

1. Some labels (e.g., weight: "light," "medium," "heavy") are inherently subjective, especially when assigned solely from visual data without tactile or force feedback. This issue extends to other properties like "stickiness" or "thickness." The paper would benefit from a clearer explanation or standardization strategy—such as using reference objects or relative comparisons to reduce ambiguity.
2. Some tasks exhibit extremely poor performance, e.g., 0% accuracy for all models on A5 (Cleaning Affordance) in Table 3. It would be helpful to analyze whether these failures are due to labeling ambiguities, lack of relevant training data, or misinterpretation patterns by the models. And then what types of mistake answers do models commonly generate?
3. Section 4.3 refers to affordance metrics being shown in Table 5, but Table 5 is not present. This appears to be a typo and may have intended to point to Table 3. Please revise accordingly for clarity.

**Strengths Contributions:**

1. The paper studies an important and underexplored problem in Embodied AI: whether foundation models can reason about physically grounded attributes and constraints, which is essential for reliable robot planning and execution.
2. The benchmark is well designed, featuring 15 diverse property types and over 100 real-world humanoid scenarios involving object constraints.
3. Experimental results are insightful. They clearly demonstrate the current limitations of VLMs in understanding PAC, offering valuable direction for future research on grounding large models in physical reasoning.
4. Overall, the paper is well-structured, easy to understand.

---

> ### Author Rebuttal · Authors · 2025-07-31
>
> We thank Reviewer hcSb for their thoughtful feedback and for recognizing the importance of physically grounded reasoning in VLMs. Below we address each of your concerns.
>
> 1. **Weakness: Subjectivity in Property Labels (e.g., "light / medium / heavy")**
>
> We thank the reviewer for raising this important point about label subjectivity. We took care to standardize these definitions. Please see Appendix E1 for the annotation strategy and annotation quality control procedure. We will clearly mention these definitions under Section 3.1 (Data Acquisition and Curation of the main paper. To summarize,
>
> a. **Standardization Strategy**
> To mitigate subjectivity, our annotation guidelines were anchored to common-sense references. For instance, weight was defined relative to typical human manipulation ("light" = liftable with one finger, "medium" = requires a full hand grasp, "heavy" = requires two hands or significant effort). Similarly, properties like Stickiness and Hardness were grounded using clear examples (details in Appendix B). This approach draws on principles from commonsense reasoning in AI, cognitive psychology (e.g., prototype theory), and human factors engineering, where perceived weight and affordance labeling often rely on visual heuristics in the absence of physical interaction.
>
> While we, the authors, discussed the alternative of using relative comparisons during annotation, we found that this approach tends to increase cognitive workload and introduce greater annotator noise. As a result, it would require a substantially larger number of annotations to achieve reliable agreement. We believe such an approach is appropriate when there is a learning agent as in RLHF, but not for passive annotation.
>
> b. **Quality Control**
> For each manual annotation, we ensured that at least two individuals independently provided the ground truth, and we retained only those items for which there was complete agreement between annotators (i.e., strict unanimity).
>
> 2. **Analysis of 0% Accuracy on A5 (Cleaning Affordance)**
>
> The consistent failure of models on the "Cleaning" affordance (A5) highlights a critical gap between recognizing an object and understanding its functional purpose. Our analysis suggests this is due to two primary factors:
>
> a. **Failure to Connect Objects to Actions**
> Models often identify the object and its physical properties but fail to infer its primary purpose or associated verb. For example, when shown a sponge, instead of the affordance "to clean" or "to wipe," models frequently output descriptive but functionally incomplete phrases like "is porous," "can absorb water," or "is soft." This reveals a struggle to move from passive description to active, goal-oriented reasoning.
>
> b. **Lack of Explicit Grounding in Training Data**
> Web-scale datasets, while vast, often lack explicit pairings of an object with its functional affordance. A model sees millions of images of "sinks" but may rarely encounter text that explicitly states a sink is for washing. This points to a need for more functionally-grounded training data.
>
> It is, unfortunately, harder to provide a quantitative analysis for the whole dataset, as analyzing the reasons for incorrect predictions for such a large dataset is much more expensive than the annotation cost itself.
>
> 3. **Reference to Table 5 (Table 5 is present in Appendix)**
>
> The reference in Section 4.3 was indeed intended to point to tables in the appendix. We will correct the text in Sec 4.3 so that it reads "(see Appendix Table 5 for affordance metrics)".

---

> > ### Comment · Reviewer_hcSb · 2025-08-04
> >
> > Thank you to the authors for their rebuttal. While some of my concerns have been addressed, I maintain my score and recommend accepting the paper.

---

> > > ### Author Response · Authors · 2025-08-05
> > >
> > > We thank Reviewer hcSb for your continued support and for maintaining your recommendation to accept our paper. We’re glad that our revisions addressed your concerns, and we sincerely appreciate your vote of confidence and constructive feedback.

---

### Comment · Area_Chair_9Cnk · 2025-08-02
**Author-Reviewer Discussions (July 31 - Aug 6)**

Firstly, I'd like to thank the reviewers for their initial reviews and the authors for responding to the reviewers with explanations and clarifications. In the limited time available, I would like to encourage reviewers to carefully read all other reviews and the author responses and engage in an open exchange with the authors. Please post your first response as soon as possible, so there is time, if needed, for back and forth discussion with the authors. Ideally, the reviewers should respond to the authors, so that the authors know their rebuttal has been read.
Thank you, AC

---

### Note · Authors · 2025-08-13

We thank the AC and the reviewers who actively engaged in the discussion for their constructive feedback. The consensus recognized PAC Bench as the first large benchmark to jointly evaluate Properties, Affordances, and Constraints for manipulation, its large-scale real-plus-sim dataset, and its diagnostic value in revealing key VLM shortcomings.

In response, we clarified distinctions from prior benchmarks, detailed strict annotation guidelines to reduce subjectivity, analyzed 0% constraint scores as intended diagnostics, added Chain-of-Thought experiments (small, model-specific gains), and demonstrated strong correlations between PAC Bench scores and downstream planner-style queries.

These updates address all raised concerns and strengthen PAC Bench as a robust, practical tool for advancing physically grounded VLM evaluation.

---

### Decision · Program_Chairs · 2025-09-18

**Decision:**

Accept (poster)

**Comment:**

The work proposes PAC Bench, a benchmark for evaluating VLM comprehension of object properties, affordances, and constraints (PAC) from a (robot) task executability perspective. The paper reveals significant gaps in the ability of VLMs to capture fundamental physical concepts. As such the benchmark provides a useful resource for future studies. The paper initially received all positive reviews accompanied by a number of questions.  A rebuttal phase followed, along with extensive discussion, which clarified various aspects, including distinctions from prior benchmarks, detailed strict annotation guidelines to reduce subjectivity, etc. The reviewers' final recommendations converged on 2x Accept and 2x Borderline Accept. I agree with the recommendations and have indicated accordingly.

===== FINAL UPDATE FROM DB Track PCs ====

The final decision for this paper has been taken by the program chairs after consultation with the SACs. All Senior Area Chairs have ranked papers according to the feedback from the AC during the review process. We decided to leave the original meta-review to reflect the opinion of the AC in light of the initial discussions with reviewers and SAC.